# Protracted neuronal recruitment in the temporal lobes of young children

Marcos Assis Nascimento[1,2 ✉], Sean Biagiotti[3], Vicente Herranz-Pérez[4,5], Samara Santiago[3,6,7], Raymund Bueno[8,9], Chun J. Ye[8,9,10,11,12,13], Taylor J. Abel[14], Zhuangzhi Zhang[15], Juan S. Rubio-Moll[16], Arnold R. Kriegstein[2,17], Zhengang Yang[15], Jose Manuel Garcia-Verdugo[4], Eric J. Huang[18], Arturo Alvarez-Buylla[1,2,19 ✉] & Shawn F. Sorrells[3,6,7,19 ✉]

The temporal lobe of the human brain contains the entorhinal cortex (EC). This region of the brain is a highly interconnected integrative hub for sensory and spatial information; it also has a key role in episodic memory formation and is the main source of cortical hippocampal inputs[1–4]. The human EC continues to develop during childhood[5], but neurogenesis and neuronal migration to the EC are widely considered to be complete by birth. Here we show that the human temporal lobe contains many young neurons migrating into the postnatal EC and adjacent regions, with a large tangential stream persisting until the age of around one year and radial dispersal continuing until around two to three years of age. By contrast, we found no equivalent postnatal migration in rhesus macaques (*Macaca mulatta*). Immunostaining and single-nucleus RNA sequencing of ganglionic eminence germinal zones, the EC stream and the postnatal EC revealed that most migrating cells in the EC stream are derived from the caudal ganglionic eminence and become LAMP5[+]RELN[+] inhibitory interneurons. These late-arriving interneurons could continue to shape the processing of sensory and spatial information well into postnatal life, when children are actively interacting with their environment. The EC is one of the first regions of the brain to be affected in Alzheimer's disease, and previous work has linked cognitive decline to the loss of LAMP5[+]RELN[+] cells[6,7]. Our investigation reveals that many of these cells arrive in the EC through a major postnatal migratory stream in early childhood.

The EC is a cortical region located between the hippocampal allocortex and the temporal lobe neocortex. It is interconnected with the late-developing frontal cortex[8,9] and hippocampus[10,11]. The embryonic development of the EC differs from that of the neocortex; one example of this is the early formation of the stellate islands in the EC superficial layers[12,13]. The birth and migration of cortical neurons mainly occur in mid-embryogenesis, but there is evidence suggesting that neurons in the EC continue to mature postnatally in humans[5]. Two molecular markers that are commonly found in immature neurons—the microtubule-binding protein doublecortin (DCX) and the cell-surface protein polysialylated neural cell adhesion molecule (PSA-NCAM)—have been observed in the postnatal human EC[14,15]. This raises the question of whether the EC continues to receive neurons postnatally.

## Neuronal migration in the infant temporal lobe

We examined human temporal lobe samples from fixed post-mortem brains and fresh tissue from epilepsy surgical resections (Supplementary Table 1). Cleared blocks of tissue from the temporal lobe at birth revealed a multilayered lamina that contains densely packed cell nuclei extending from the medial wall of the temporal lobe lateral ventricle (tLV), between the hippocampus and the EC (Fig. 1a, Extended Data Fig. 1a and Supplementary Video 1). Immunostaining for DCX and PSA-NCAM showed that this region contains a large collection of DCX[+]PSA-NCAM[+] cells, which form a network of chains (Fig. 1b,c and Extended Data Fig. 1b,c). We mapped this large medial migratory stream extending towards the EC in coronal sections from the anterior tip of the amygdala to the mid-hippocampus (covering about 2 cm)

[1]Department of Neurological Surgery, University of California, San Francisco, CA, USA. [2]Eli and Edythe Broad Center of Regeneration Medicine and Stem Cell Research, University of California, San Francisco, CA, USA. [3]Department of Neuroscience, University of Pittsburgh, Pittsburgh, PA, USA. [4]Laboratory of Comparative Neurobiology, Institute Cavanilles, University of Valencia, CIBERNED, Valencia, Spain. [5]Department of Cell Biology, Functional Biology and Physical Anthropology, University of Valencia, Burjassot, Spain. [6]Center for Neuroscience Graduate Training Program, University of Pittsburgh, Pittsburgh, PA, USA. [7]Center for the Neural Basis of Cognition at the University of Pittsburgh, Pittsburgh, PA, USA. [8]Institute of Human Genetics, University of California, San Francisco, CA, USA. [9]Division of Rheumatology, Department of Medicine, University of California, San Francisco, CA, USA. [10]Department of Epidemiology and Biostatistics, University of California, San Francisco, CA, USA. [11]Institute of Computational Health Sciences, University of California, San Francisco, CA, USA. [12]Parker Institute for Cancer Immunotherapy, San Francisco, CA, USA. [13]Chan Zuckerberg Biohub, San Francisco, CA, USA. [14]Department of Neurological Surgery, University of Pittsburgh, Pittsburgh, PA, USA. [15]State Key Laboratory of Medical Neurobiology and Institutes of Brain Science, Department of Neurology, Zhongshan Hospital, Fudan University, Shanghai, China. [16]Servicio de Obstetricia, Hospital Universitari i Politècnic La Fe, Valencia, Spain. [17]Department of Neurology, University of California, San Francisco, CA, USA. [18]Department of Pathology, University of California, San Francisco, CA, USA. [19]These authors contributed equally: Arturo Alvarez-Buylla, Shawn F. Sorrells. ✉e-mail: marcos.assisnascimento@ucsf.edu; alvarezbuyllaa@ucsf.edu; shawn.sorrells@pitt.edu

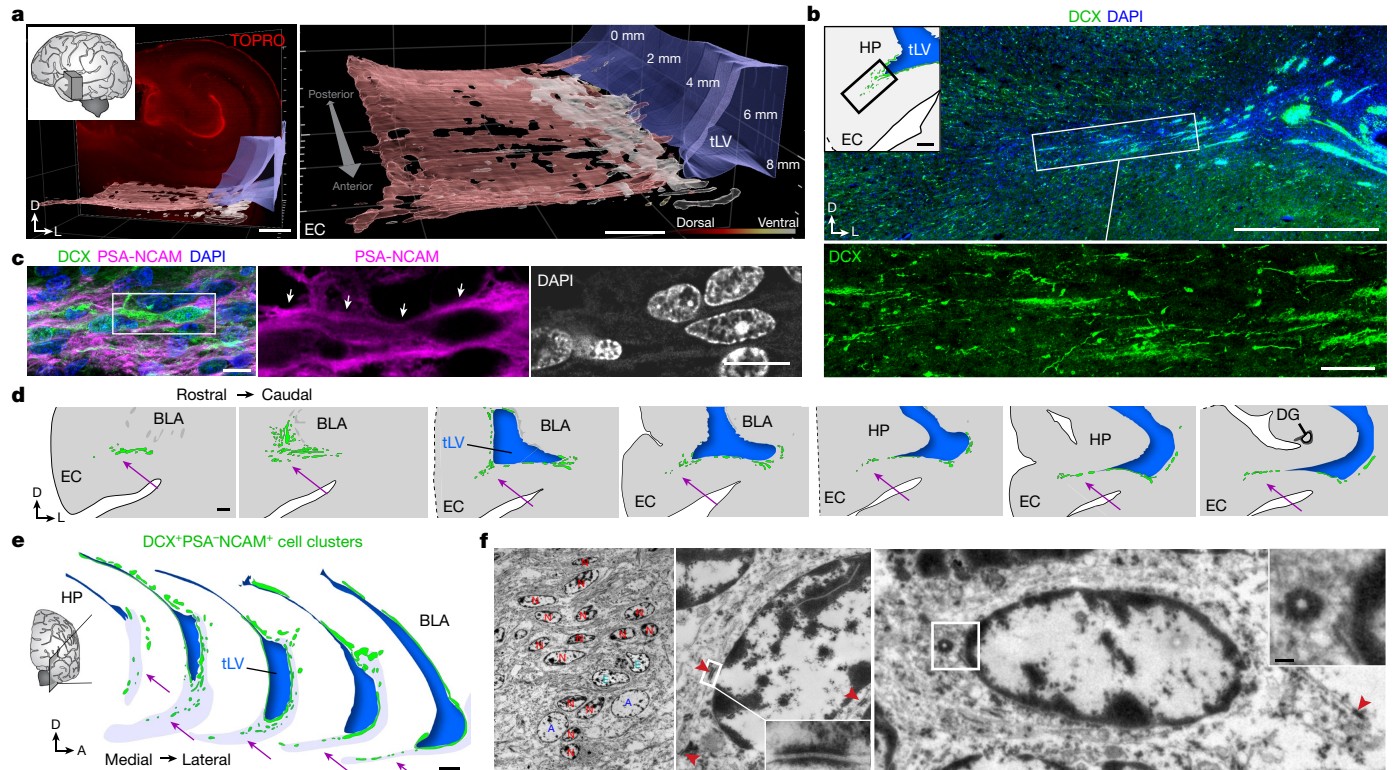

**Fig. 1 | Stream of migratory neurons in the perinatal human temporal lobe.**
**a**, Optical section of a 7.6-mm-thick stack of cleared medial temporal lobe at birth, stained with TOPRO and imaged using light-sheet microscopy. The EC stream can be identified as a multilayered lamina of dense clusters of nuclei extending medially from the tLV. Three-dimensional (3D) reconstruction of these clusters highlights their extension along the anterior–posterior axis (see Supplementary Video 1). **b**, Coronal section at 38 GW, showing DCX+ immunostained cells in the EC stream. HP, hippocampus. **c**, Deconvolution image at birth, showing a dense cluster of DCX+ cells co-expressing PSA-NCAM (arrowheads) with elongated nuclei (DAPI). **d**, Coronal maps of the EC stream (purple arrows) at 38 GW; mapped sections spaced by around 1 mm from the anterior tip of the basolateral amygdala (BLA). **e**, Sagittal maps of the EC stream (purple arrows) at 10 postnatal days; sections spaced by around 1 mm from the medial end of the tLV. **f**, Left, transmission electron microscopy (TEM) image of immature neurons in the human EC stream at birth, showing densely packed neurons (red, N) with compacted chromatin and fusiform morphology, surrounded by astrocytes (blue, A); note the presence of ependymal cells (cyan, E; see also Fig. 3). Middle, higher magnification of a neuron with adherens junctions (arrowheads, inset). Right, many cells in the EC stream had features of migratory neurons: a centrosome (inset) opposite a trailing process with an adhesion point (arrowhead). Scale bars, 1 mm (**a**,**b**,**d**,**e**); 100 μm (**b** bottom); 10 μm (**c**, **f** left); 1 μm (**f** middle and right); 200 nm (**f** middle and right insets). D, dorsal; L, lateral; A, anterior.

(Fig. 1d and Extended Data Fig. 1d). In sagittal sections, these chains were present cascading over the rostral end of the tLV at medial levels, and ventral to the hippocampus between the ventricle and the EC at lateral levels (Fig. 1e). Because DCX can be expressed in more mature neurons[3,10,14], we examined the ultrastructural features of the cells in the dense clusters at birth. The cells had small, elongated cell bodies with leading and trailing processes, scarce cytosol and an elongated nucleus with compacted chromatin (Fig. 1f and Extended Data Fig. 1e), which indicates that these clusters of DCX+PSA-NCAM+ cells correspond to chains of migrating young neurons[16]. We refer to this expansive, medially oriented network of migratory chains extending towards the entorhinal cortex as the EC stream.

## The EC migratory stream persists after birth

To determine how long the EC stream persists postnatally, we stained sections of the temporal lobe for DCX and PSA-NCAM from birth to three years of age. Dense chains of DCX+PSA-NCAM+ cells were present along the temporal lateral ventricle and in the EC stream up to the age of 11 months, but were no longer observed at 14 months (Fig. 2a). In addition, dispersed individually migrating DCX+ cells were present throughout the temporal lobe at birth, including in the developing white matter and the cortical plate. We mapped the location and the

orientation of DCX+ cells with migratory morphology from birth to the age of three years (Fig. 2b,c and Extended Data Fig. 2a–e). The number of individually migrating DCX+ cells in the medial temporal lobe decreased between birth and the age of seven months, but migrating DCX+ cells were still present between seven months and two years of age. At three years of age we could still observe a few individual migratory DCX+ cells in the EC, where persistent immature neurons have been previously described[14,15].

We next evaluated the orientation of DCX+ cells that had a single leading process and elongated nucleus, suggestive of active migration (Fig. 2b,c and Extended Data Fig. 2b–e). We classified DCX+ cells as oriented tangentially (parallel) or radially (perpendicular) to the cortical surface. From birth to two years of age, similar numbers of DCX+ cells were oriented radially and tangentially in subcortical areas close to the EC stream (50.1% at birth; 58.2% at seven months; 58.5% at two years). By contrast, the EC had more than twice as many DCX+ cells migrating radially as tangentially (66.9% of all migrating neurons at birth; 87.5% at seven months; 79.9% at two years). This suggests that within the EC stream and surrounding areas, DCX+ young neurons tend to disperse tangentially, and once in the cortex they move mainly within radial columns.

Next, we investigated whether a similar postnatal migratory stream was present in the temporal lobe of rhesus macaques (*Macaca mulatta*).

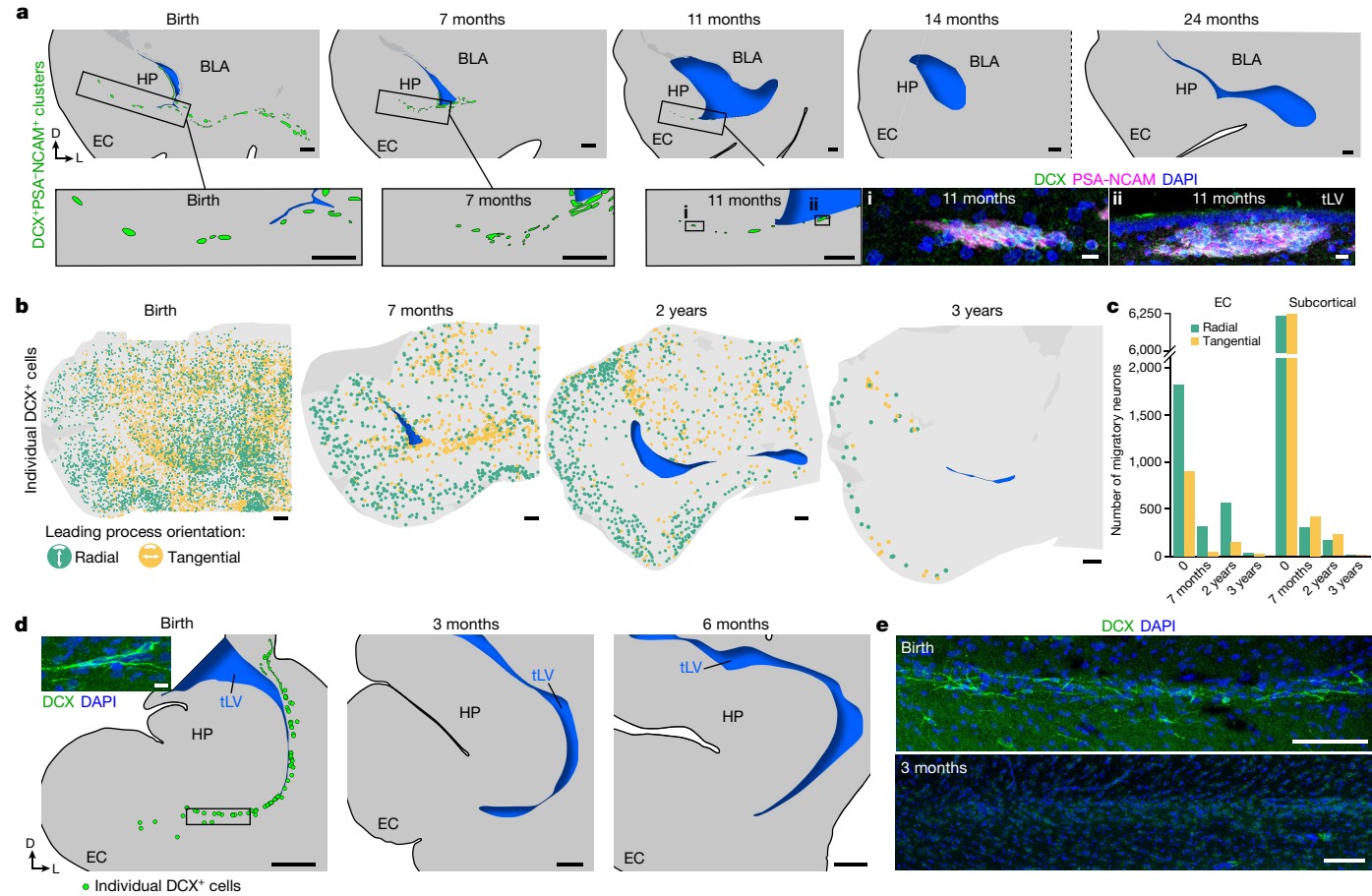

**Fig. 2 | The EC stream supplies migratory neurons until the age of two to three years. a,** Maps of DCX⁺PSA-NCAM⁺ cell clusters in coronal sections of the human medial temporal lobe from birth to 24 months of age. Bottom right box: DCX⁺PSA-NCAM⁺ immunostained cell clusters are shown at 11 months of age. **b,** Maps of the orientation of DCX⁺ neurons in coronal sections from birth to three years of age, indicating the location of radial (orthogonal to the cortical surface; green) and tangentially oriented (yellow) neurons in the EC. **c,** Number of neurons with radial and tangential orientations in subcortical white matter and in the EC between birth and three years of age, counted in 20-μm sections. **d,** Maps of DCX⁺ cells in the macaque medial temporal lobe between birth and six months of age, and (inset) an example of DCX⁺ immunostained cells. **e,** DCX⁺ immunostaining in the macaque EC stream region indicated in **d** (box) in at birth and at three months of age. Scale bars, 1 mm (**a** maps, **b, d** maps); 100 μm (**e**); 10 μm (**a** bottom right stains, **d** top left).

We stained temporal lobe sections for DCX and PSA-NCAM at birth and at 3, 6 and 17 months of age (Fig. 2d,e and Extended Data Fig. 2f–i). At birth, we observed sparse clusters and individual DCX⁺PSA-NCAM⁺ cells in the ventricular–subventricular zone (V-SVZ) of the temporal lobe (Extended Data Fig. 2f,g). In the EC stream region at birth, macaques had a dispersed population of DCX⁺ cells, whereas within the EC, DCX⁺ cells with migratory morphology were infrequent (one or two per section). From 3 to 17 months of age we observed no DCX⁺ neurons in the EC stream or in the EC. These results indicate that the postnatal macaque brain does not have a comparable postnatal migratory stream, but we cannot exclude the possibility that similar neurons arrive earlier during gestation.

## Formation of the EC migratory stream

The prominence of EC stream migration at birth made us wonder how this migratory route forms during gestation. At 18 gestational weeks (GW), the EC was a thin strip of tissue in the ventral temporal lobe adjacent to the ventricle, which at this age was open, with the caudal ganglionic eminence (CGE) germinal zone present more dorsally along the same ventricle wall (Fig. 3a). Between 18 and 22 GW, tissue growth had pressed these ventricular walls close together (Fig. 3b,c and

Extended Data Fig. 3a–d). At 22 GW, the medial ventricle wall was lined with FOXJ1⁺ multiciliated ependymal cells, whereas the lateral wall contained abundant vimentin⁺ radial glia (RG) fibres (Fig. 3c). Between 22 and 29 GW, these ventricle walls completely fused together (Extended Data Fig. 3a–e). Despite the ventricle fusion and the complete absence of an open ventricle, at birth and at seven months this region retained FOXJ1⁺ ependymal cells individually or in clusters (Fig. 3d). This was confirmed ultrastructurally, with clusters of multiciliated ependymal cells embedded within the EC stream at birth (Fig. 3e and Extended Data Fig. 3f). At birth and at seven months, we still observed RG in this region, and their vimentin⁺ processes surrounded the DCX⁺ clusters in the EC stream (Fig. 3d and Extended Data Fig. 3e), resembling the glial tube of the rostral migratory stream (RMS)[9]. These fibres seem to guide each tributary of the EC stream away from the wall of the tLV (Fig. 3d, arrow). The ultrastructure of the EC stream at birth revealed chains of elongated migratory neurons oriented in the direction of surrounding glial fibres (Fig. 3f). By contrast, in the postnatal macaque (at 3 or 17 months), the same ventricular region retained segments that were open postnatally, suggesting that this region does not completely fuse as it does in humans (Extended Data Fig. 3g). Thus, in humans, the fusion of ventricle walls and their V-SVZs creates a path for migrating neurons towards the postnatal EC.

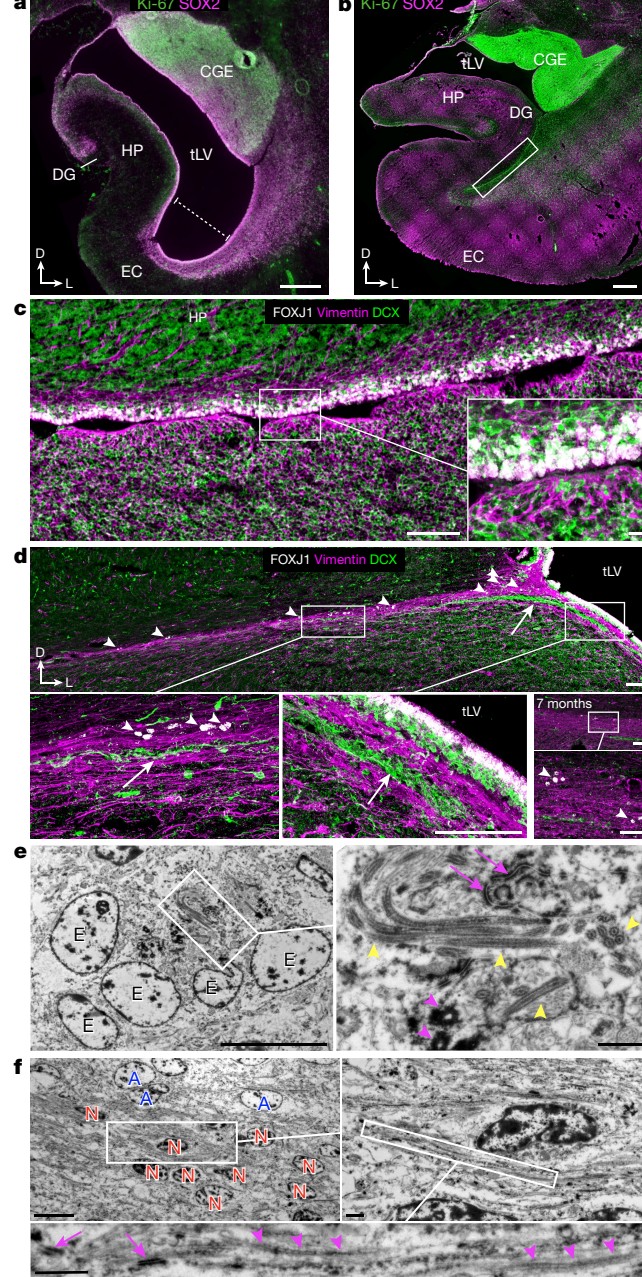

**Fig. 3 | The EC stream extends along a radial scaffold next to a fused extension of the ventricle. a**, At 18 GW, coronal sections show Ki-67⁺SOX2⁺ cells prominent in the CGE and lining the ventricle walls. Ventrally, the medial wall facing the hippocampus and lateral wall facing the cortex are open (dotted line). DG, dentate gyrus. **b**, At 22 GW, the medial and lateral ventricular zone (VZ) are in close proximity. **c**, At 22 GW, in an adjacent section (boxed region in **b**), FOXJ1⁺ cells are present in the medial wall VZ, but not on the opposite wall. Note the network of vimentin⁺ fibres of RG on the lateral wall (inset). **d**, At birth, the ventricular walls in the EC stream region have fused together, with some remaining FOXJ1⁺ ependymal cells (arrowheads), but no open ventricle. Note the multiple layers of medially oriented DCX⁺ cells flanked by vimentin⁺ fibres turning away from the ventricle (arrow). Right, at seven months, clusters of DCX⁺ cells remain in the EC stream, but have decreased in size. Some FOXJ1⁺ ependymal cells (arrowheads) and vimentin⁺ fibres remain. **e**, TEM of five ependymal cells (E) in the EC stream at birth, containing multiple long cilia (yellow arrowheads), ciliary basal bodies (magenta arrowheads) and cell–cell junctions typical of ependymal cells (magenta arrows). **f**, Ultrastructural detail of an immature migratory neuron (red, N) in the EC stream at birth surrounded by glial fibres and cell bodies (blue, A). Adherens junctions (magenta arrows) are visible next to glial fibres (magenta arrowheads). Scale bars, 1 mm (**a**,**b**); 100 μm (**c** left, **d**); 10 μm (**c** inset, **e** left, **f** top left); 1 μm (**e** right, **f** right and bottom).

## The EC stream contains migrating interneurons

To characterize the identity of the migrating neurons in the EC stream, we microdissected this region from a two-week-old sample and performed single-nucleus RNA sequencing (snRNA-seq) (Fig. 4a–d). Most of the captured *DCX*-expressing neurons (72.2%) expressed the GABAergic interneuron markers distal-less homeobox 2 (*DLX2*) (ref. 17) and glutamate decarboxylase 2 (*GAD2*). A smaller fraction expressed the cortical excitatory neuron transcription factor T-brain 1 (*TBR1*) (ref. 18; Fig. 4c,d). The immature inhibitory neurons were present in two clusters in our dataset. The larger cluster (86.5% of cells) expressed nuclear receptor subfamily 2 group F member 2 (*NR2F2*; encoding COUPTFII), consistent with a possible CGE origin. The smaller cluster expressed LIM-homeobox 6 (*LHX6*), consistent with a medial ganglionic eminence (MGE) origin (Fig. 4c,d).

To confirm the identity of the immature neurons in the EC stream, we performed immunostaining on samples between birth and 11 months of age (Extended Data Figs. 4 and 5). From birth to seven months of age, most cells within the temporal lobe V-SVZ and EC stream were DCX⁺DLX2⁺ but not TBR1⁺, corroborating their inhibitory identity (Fig. 4e,f and Extended Data Fig. 5a–e). At birth, 82.4% (1,163/1,412) of the DCX⁺ cells were DLX2⁺ in the EC stream and 1.9% (29/1,553) were TBR1⁺. Furthermore, at birth, most of the migrating neurons in the EC stream were COUPTFII⁺ (88.3%; 339/384), and co-localized with DLX2 (Fig. 4f and Extended Data Figs. 4a,c–e and 5b,c). At this age, a smaller fraction of EC stream cells expressed specificity protein 8 (SP8) (2%; 19/967) or LHX6 (21.0%; 81/423) (Extended Data Fig. 4b,f), consistent with the snRNA-seq data. In contrast to the EC stream, the human RMS at birth contains very few DCX⁺COUPTFII⁺ cells (Extended Data Fig. 4h). The migrating cells in the EC stream also expressed the calcium-binding protein secretagogin (SCGN) (Fig. 4g) which is present in the cytoplasm of immature CGE-derived interneurons[19] (see also Extended Data Fig. 9d).

Within the EC, we next investigated the identity of the individual DCX⁺ cells between birth and two years of age. Similarly to the EC stream, DCX⁺ cells in the EC at birth were 85.5% (868/1,015) DLX2⁺ (Fig. 4h). The EC also contained a subpopulation of DCX⁺TBR1⁺ neurons that were frequently multipolar, suggesting that they had completed their migration (Fig. 4h and Extended Data Fig. 5g,h). Unlike the EC stream, only 21.4% of DCX⁺ cells in the EC were COUPTFII⁺ at birth, and 44.4% were LHX6⁺. With increasing postnatal age, the total population of DCX⁺ cells in the EC decreased and the LHX6⁺ fraction dropped sharply. By contrast, there was a threefold increase in the percentage of DCX⁺COUPTFII⁺ cells, which remained around 65% between three months and two years of age (Fig. 4j,k and Extended Data Fig. 4g). At two years of age, few DCX⁺ cells in the EC expressed SP8, LHX6 or NKX2.1 (12.3%, 5.2% and 1.9%, respectively). Compared to the EC stream, the co-expression of DCX, PSA-NCAM and either COUPTFII or DLX2 in the EC was less frequent, with more cells expressing only DCX or PSA-NCAM (Extended Data Fig. 5i,j). Individually migrating SCGN⁺ cells in the EC were abundant at birth and persisted in large numbers in the seven-month EC, decreasing between eleven months and two years (Fig. 4i). This SCGN immunostaining clearly reveals the morphology and abundance of young migrating neurons in the EC stream and EC. The data show the extent of this massive postnatal migration in the human brain and further support its CGE origin[19].

Unexpectedly, and unlike what is seen in humans, the sparse, individually migrating DCX⁺ cells in the comparable EC stream region in the macaque were rarely COUPTFII⁺ (Fig. 4l and Extended Data Figs. 4i and 5f). Instead, these cells were frequently SP8⁺, which is rare in the human EC stream but common in the macaque RMS[20]. We were able to observe DCX⁺DLX2⁺COUPTFII⁺ cells only within the former CGE region in the macaque. Together, these observations indicate that the sparse individual migratory cells in the macaque at birth have a different molecular identity than do those in the human EC stream.

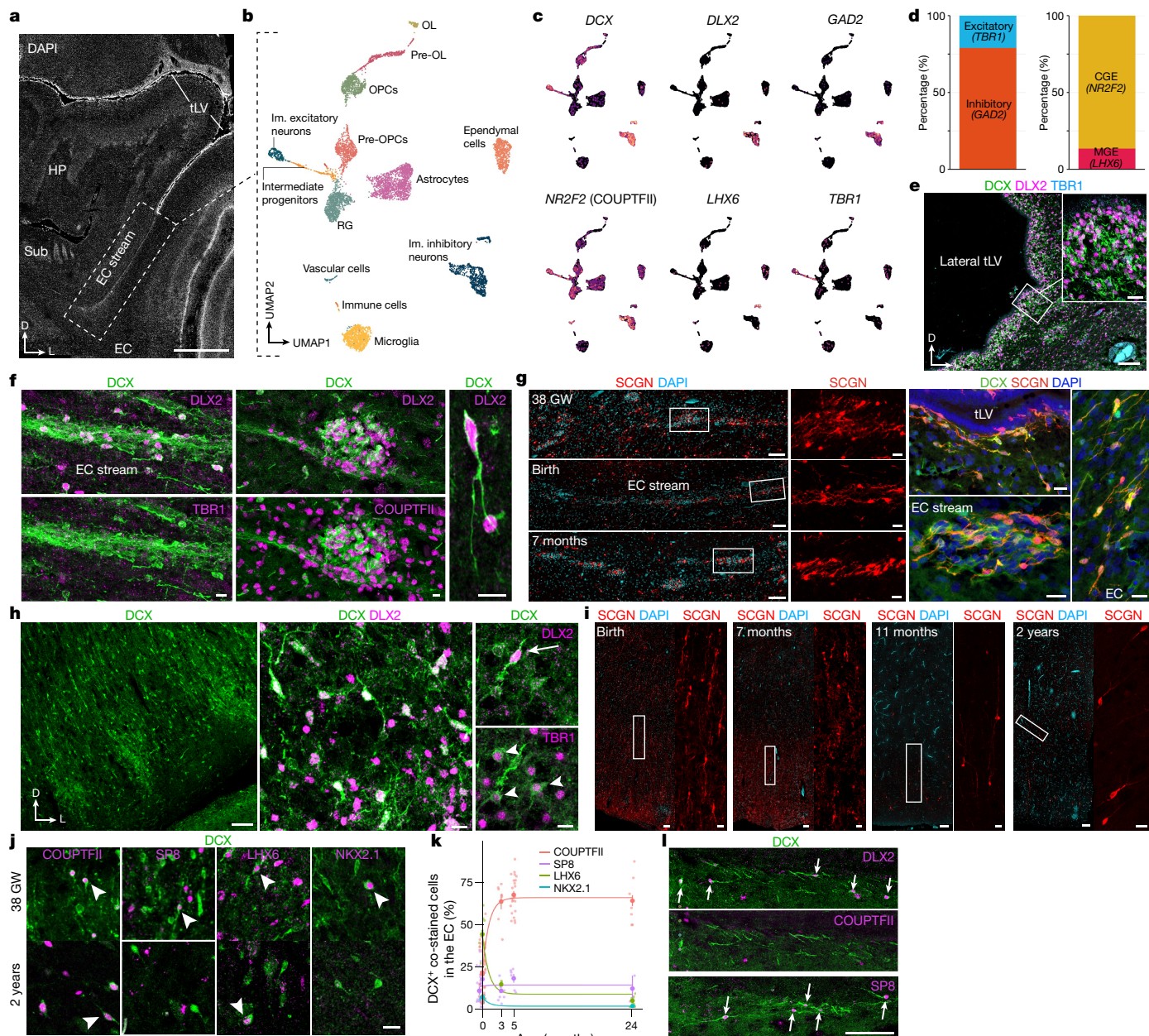

**Fig. 4 | The postnatal EC stream mainly supplies migrating CGE-derived interneurons. a**, Tissue section showing a microdissected region of the EC stream from a two-week-old sample (dotted box). Sub, subiculum. **b**, Unsupervised clustering of nuclei in the EC stream. OL, oligodendrocytes; OPCs, oligodendrocyte precursor cells; Im, immature; pre-OL, pre-oligodendrocytes. **c,d**, Gene-expression feature plots (**c**) and relative abundance (**d**) of immature excitatory (*TBR1*[+]*DCX*[+]) and inhibitory (*DLX2*[+]*GAD2*[+]DCX[+]) neurons and MGE-derived (*LHX6*[+]) or CGE-derived *NR2F2*[+] interneurons. **e**, At birth, the tLV lateral wall contains many DCX[+]DLX2[+] cells that are TBR1[-]. **f**, EC stream DCX[+] cells (at birth) co-stained for DLX2 and TBR1 or DLX2 and COUPTFII. **g**, Left, SCGN[+] cell clusters in the EC stream at 38 GW, birth and 7 postnatal months. Right, DCX[+]SCGN[+] cells at birth in the temporal lobe

V-SVZ, EC stream and extending into the EC. **h**, The human EC at 38 GW immunostained for DCX and DLX2 or TBR1. **i**, The human EC between birth and two years of age, immunostained for SCGN. **j**, The human EC at 38 GW and two years of age, showing DCX[+] cells co-stained with COUPTFII, SP8, LHX6 or NKX2.1 (arrowheads). **k**, Percentage of DCX[+] cells co-stained with COUPTFII, SP8, LHX6 or NKX2.1 in the EC between birth and two years of age. Large dots, mean; small dots, sampled images; bars, s.e.m. of images for each individual; *n* = 5 individuals in three independent experiments. **l**, Top two rows: macaque EC stream region at birth, containing DCX[+] cells co-stained with DLX2 (arrows) that are COUPTFII[-]. Bottom row: this region contains individually migratory DCX[+]SP8[+] cells (arrows). Scale bars, 3 mm (**a**); 100 μm (**e** overview, **g** left, **h** left, **i** left, **l**); 20 μm (**e** inset, **g** right panels, **i** insets, **j**); 10 μm (**f**, **h** right panels).

## The EC stream is produced by CGE progenitors

On the basis of the presence of many migrating young neurons and RG in this collapsed V-SVZ region at birth, we next investigated the nature of the proliferating cells in this area. We began by staining for dividing cells and neural progenitor markers between 18 and 29 GW and at

birth. Cells co-expressing the cell proliferation marker Ki-67 and the neural progenitor marker SRY-box transcription factor 2 (SOX2) were present in a region extending from the ventromedial wall of the ventricle through the EC stream (Extended Data Fig. 6). At birth, a subset of Ki-67[+] cells in this region were vimentin[+]HOPX[+] RG (Fig. 5a–c). Notably, our snRNA-seq of the stream captured a cluster that expressed RG

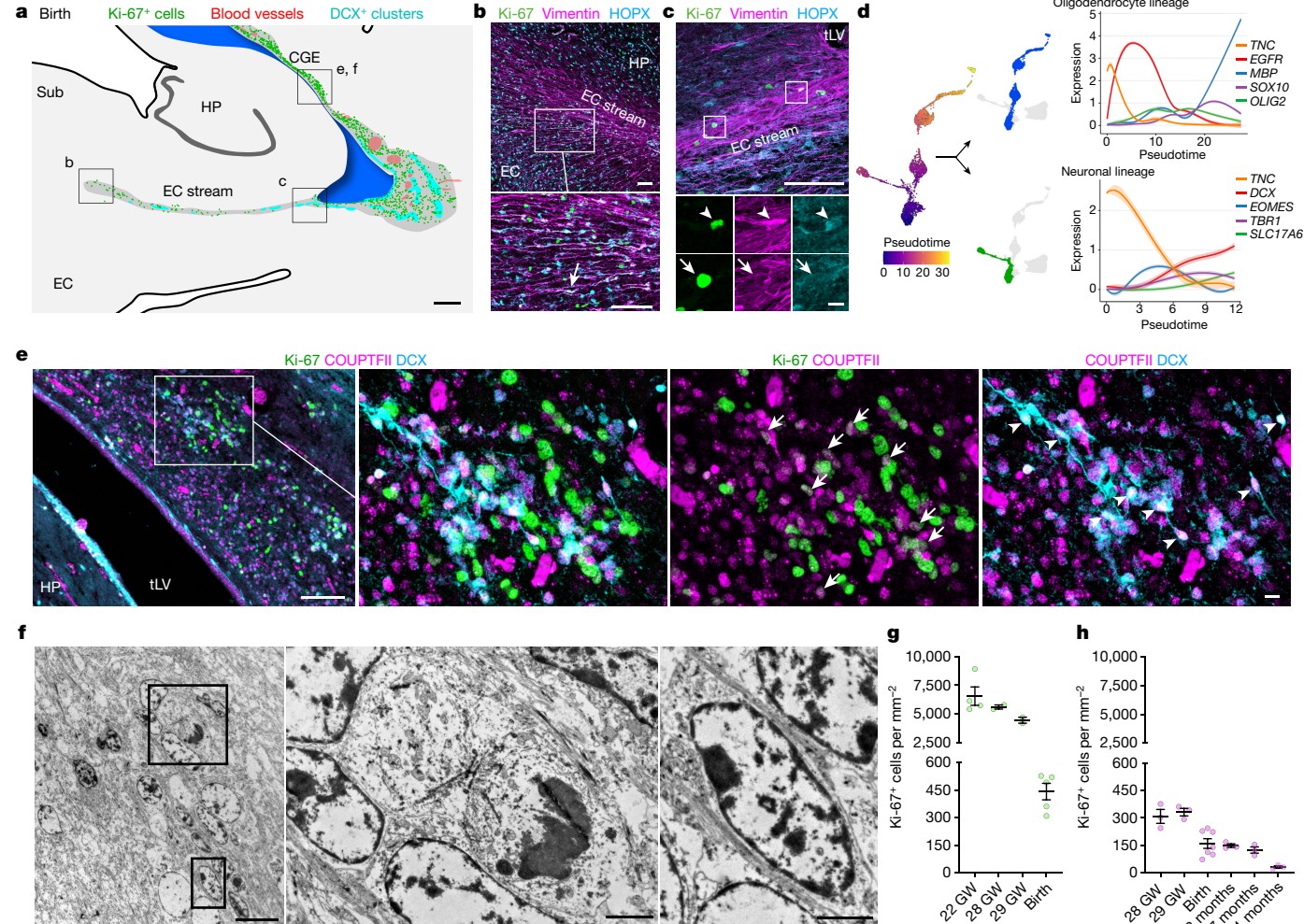

**Fig. 5 | Progenitor cells in the CGE and EC stream region continue dividing at birth. a**, Coronal map (at birth) of Ki-67⁺ cells (green), blood vessels (red) and DCX⁺ cell clusters (cyan) along the tLV and extending along the EC stream. Boxes show the anatomical locations of subsequent panels. **b**, Ki-67⁺vimentin⁺ RG (arrow) at the medial end of the EC stream region at birth. **c**, At birth, Ki-67⁺vimentin⁺ cell bodies located proximal to the tLV are either HOPX⁺ (arrowhead) or HOPX⁻ (arrow). **d**, Lineage trajectory analysis using Monocle reveals a bifurcation emerging from the cluster of RG into an oligodendrocyte lineage (blue, top) and a neurogenic lineage of excitatory cells (green, bottom); on the right, expression of key maturation-associated genes across pseudotime. **e**, Ki-67⁺COUPTFII⁺ cells (arrows) and DCX⁺COUPTFII⁺ cells (arrowheads) located in the CGE region of the temporal lobe at birth. **f**, TEM within the CGE region at birth (near boxed inset in **e**) showing a pair of recently divided cells completing cytokinesis (top boxed region; left of the two magnified panels) near a cell with ultrastructural features of a migratory neuron (bottom boxed region; right of the two magnified panels). **g,h**, Quantification of the density of Ki-67⁺ cells in the CGE (**g**) from 22 GW to birth and in the EC stream (**h**) from 28 GW to 11 postnatal months. Data points are the average section density at each age; bars are mean ± s.e.m., *n* = 4 individuals (**g**) and *n* = 6 individuals (**h**) in three independent experiments. Scale bars: 1 mm (**a**); 100 μm (**b**, **c** top, **e** left); 10 μm (**c** bottom, **e** right, **f** left); 2 μm (**f** right).

markers (*TNC*, *SOX2*, *VIM* and *HOPX* (ref. 21)), distinct from a cluster of differentiated astrocytes (Supplementary Table 4). We investigated possible RG differentiation trajectories using Monocle, and found two main branches departing from the RG cluster: (i) a neurogenic trajectory, beginning with the expression of genes consistent with an intermediate neuronal progenitor identity (for example, *EOMES*) and ending in a population of immature excitatory neurons expressing *DCX*, *TBR1* and *SLC17A6* (Fig. 5d); and (ii) a trajectory following the oligodendrocyte lineage, starting with the expression of the oligodendrocyte progenitor markers *OLIG2* and *SOX10* and ending in a cluster of mature oligodendrocytes expressing *MBP* and *MOG*. In the EC stream microdissection, we did not find intermediate progenitors for interneurons or a differentiation trajectory connecting the local RG to the immature inhibitory neurons.

At birth, we mapped the location of Ki-67⁺ cells across the temporal lobe and identified clusters of proliferating cells next to immature

neurons on the lateral wall of the ventricle in the region of the CGE (Fig. 5e and Extended Data Fig. 7); most were in the outer margins of the V-SVZ, with comparatively few dividing cells in the VZ. Consistent with these being remnants of CGE proliferation continuing at birth, the dense clusters contained Ki-67⁺COUPTFII⁺ cells. These proliferating progenitors were intermingled with migratory DCX⁺COUPTFII⁺ immature neurons (Fig. 5e), and ultrastructural analysis showed migrating neurons next to cells undergoing cytokinesis (Fig. 5f). In our mapped sections, we observed chains of DCX⁺ cells emanating from these proliferation hotspots and migrating tangentially close to the ventricle, along its ventral wall between the lateral tLV and the EC stream (Extended Data Fig. 7a,c–e).

We next investigated proliferation in the CGE and in the EC stream at birth compared with earlier in development. Immunostaining serial sections across the temporal lobe at 18 and 22 GW revealed the CGE on the lateral wall of the tLV, defined by an exceptionally dense collection

of Ki-67[+] progenitors expressing COUPTFII, PROX1 and SP8 (Extended Data Figs. 8 and 9). The CGE at 22 GW also contained a high density of SCGN[+] cells present along the ventricle wall that fuses to form the EC stream, as well as individually migrating SCGN[+] cells oriented towards the EC (Extended Data Fig. 9d). Persisting from 22 GW until 29 GW, the CGE had the highest density of Ki-67[+] cells of any location in the temporal lobe, with tenfold more Ki-67[+] cells than the EC stream region at comparable ages (Fig. 5g,h). At birth, the density of Ki-67[+] cells in the CGE was reduced, but remained around threefold higher than that in the EC stream. Our observations suggest that the human CGE is a hotspot of progenitor proliferation for an extended period of development that extends postnatally, and that it is likely to be the source of the majority of neurons in the postnatal EC stream.

## The postnatal EC stream supplies *LAMP5*[+] neurons

To follow the maturation of neuronal subtypes supplied by the EC stream, we performed further snRNA-seq experiments. We merged the data from the microdissected 14-day-old EC stream (Fig. 4a,b) with nuclei collected from 23-GW MGE, lateral ganglionic eminence (LGE) and CGE; nuclei from the postnatal EC between 14 days and 27 years of age; and published snRNA-seq data from the 50–79-year-old EC (ref. 3; Fig. 6a, Extended Data Fig. 10 and Supplementary Table 5). We reclustered the cortical interneurons from the combined dataset (Fig. 6b–e and Extended Data Fig. 11). Consistent with the progression of donor age (Fig. 6c), we observed a continuum of gene expression associated with different stages of neuronal maturation including proliferating precursors (*TOP2A*[+]) and immature (*DCX*[+]) and mature (*GABRB2*[+]) neurons (Extended Data Fig. 11g,j). Immature interneurons were clustered in five different identities (Fig. 6d). Mature interneurons in the postnatal EC were clustered in nine different identities defined by label transfer from a temporal lobe cortical dataset[22] (Fig. 6e and Supplementary Table 6). Interneurons in the CGE and MGE, but not LGE lineages (Extended Data Fig. 10e), were present in the EC from 23 GW to adulthood. Neurons from the microdissected EC stream were present mainly in only one of the two immature CGE clusters (Im. CGE-2) (Fig. 6b,d,f).

We next used Monocle to investigate the maturation process of EC interneurons. Starting from a cluster of immature inhibitory neurons (Im. Mix; *SOX4*[+]*SOX11*[+] cells) derived from the germinal zone samples, initial branch points separated the CGE- and MGE-derived immature (*DCX*[+]) neurons (Fig. 6d,f,g and Extended Data Fig. 11). Lineage trajectory analysis revealed six maturation trajectories (Fig. 6g). The vast majority of migratory inhibitory neurons in the EC stream were present in CGE-derived maturation trajectories (Fig. 6g, h), mainly in the early stages of the CGE-derived *LAMP5*[+] trajectory. EC stream cells were also present in a smaller fraction of the trajectory for immature *VIP*[+] CGE-derived cells and a very small fraction of MGE-derived LHX6[+] cells (Fig. 6h and Extended Data Fig. 4f). Of note, in the 23-GW EC, very few cells were in the *LAMP5*[+] maturation trajectory, suggesting that these cells have not arrived in the EC at that age (Fig. 6b,f,h and Extended Data Fig. 11a).

To identify gene modules associated with interneuron lineage progression, we performed a weighted gene co-expression network analysis (WGCNA) (Extended Data Fig. 12). This analysis identified five modules of co-expressed genes with correlated expression. Modules 1, 2 and 4 were strongly correlated with pseudotime and donor age (Extended Data Fig. 12b,c). Genes in module 2 were highly expressed in immature neurons and included those associated with neuron maturation (for example, *SOX4* and *SOX11*), cell migration and axonal guidance. Genes in module 4 were associated with protein synthesis (for example, ribosomal genes and *EEF1A1*) and were downregulated with age, consistent with recent findings[23]. Genes in module 1 were highly expressed in mature neurons (for example, genes related to synaptic communication, neuronal adhesion and the maintenance of membrane potential) (Extended Data Fig. 12a and Supplementary

Table 7). Cells in the EC stream highly expressed genes in module 2, but expressed module-1 genes at low levels, consistent with their immature interneuron phenotype (Extended Data Fig. 12e).

To infer the possible destination of EC stream cells in the EC, we used a spatial transcriptomic dataset from the adult human temporal lobe[24] as a reference for label transfer. We identified the cells in our dataset that corresponded to superficial- and deeper-layer CGE-derived *LAMP5*[+] cells (Fig. 6i). Superficial-layer *LAMP5*[+] cells had higher expression of *RELN*, *NDNF* and *NCAM2*, and lower-layer *LAMP5*[+] cells had higher expression of *RASGRF2*, *FBXL7* and *CDH13* (Fig. 6j and Supplementary Table 8). Most of the cells in the EC stream microdissection had an ambiguous identity, probably because of their immaturity, but a small subset expressed genes of upper-layer *LAMP5*[+] neurons (Fig. 6k and Extended Data Fig. 13).

We next validated these findings and identified the location of interneuron subtypes histologically. Consistent with our transcriptomic analysis, the postnatal EC stream contained cells immunoreactive for the interneuron subtype markers LAMP5, RELN and calretinin (CR, co-expressed by VIP cells) (Fig. 6l–n and Extended Data Figs. 11i and 14). At birth, we observed a subpopulation of LAMP5[+]COUPTFII[+] cells in the EC stream and in the EC, consistent with a CGE origin (Fig. 6g,l and Extended Data Fig. 4f). We also observed a subset of LAMP5+ cells that were not COUPTFII[+] and had a layer distribution consistent with MGE-derived LAMP5[+] cells[24]. At birth, the LAMP5[+]COUPTFII[+] cells were preferentially found in the superficial layers of the EC, whereas COUPTFII[+] cells co-expressing RELN, CR and VIP had a wider distribution (Fig. 6n). RELN, which is expressed in the upper-layer subpopulation of LAMP5[+] cells (Fig. 6j and Extended Data Fig. 11i), was present in multiple clusters of migratory neurons along the ventral tLV (Fig. 6m), in the EC stream from 38 GW to 7 months of age (Extended Data Fig. 14a–d) and in the EC at 11 months and 2 years (Extended Data Fig. 14e,f). In addition, there were RELN[+]COUPTFII[+]DCX[+] cells in the EC stream and in the EC. We also observed a population of DCX[+]CR[+] cells in the EC stream at birth and in the EC from birth to the age of two years, likely corresponding to the *VIP*[+] neurons observed in the transcriptomic analysis (Extended Data Fig. 11i and 14g–i). Together, these results indicate that the EC stream supplies LAMP5[+]RELN[+] CGE-derived interneurons that populate mainly the upper layers of the EC.

## Discussion

Our findings show that the postnatal human temporal lobe retains a large migratory stream that carries immature migrating neurons into the EC and neighbouring cortical regions. This migratory stream forms during gestation next to the location of a fused ventricle and is maintained postnatally for at least 11 months. Notably, individual young neurons continue to migrate into the cortex until the age of two to three years, but do not do so at older ages (3–77 years). These findings were corroborated by the donor age-matched snRNA-seq data. Most of these migrating neurons are young *LAMP5*[+] CGE-derived interneurons. We did not observe a similar postnatal migratory stream in rhesus macaques, and the few migratory cells that were present at birth in macaques expressed different transcription factors to those expressed in humans. In the human frontal lobe[8,9], streams of young neurons have also been described in the first few postnatal months. Our study shows that throughout the first year of life, the human temporal lobe contains large streams of migrating young neurons that travel long distances and are recruited into the postnatal circuits of the cerebral cortex.

Cortical interneurons are considered to be key to the brain's evolutionary innovation[25,26]. The contribution of the CGE to the human cortex is nearly twice what is observed in rodents[24,27], and this results in greater numbers and diversity of cortical interneurons[28–32]. Our histological and transcriptomic analyses are consistent with a CGE origin for most of the immature neurons in the postnatal EC stream.

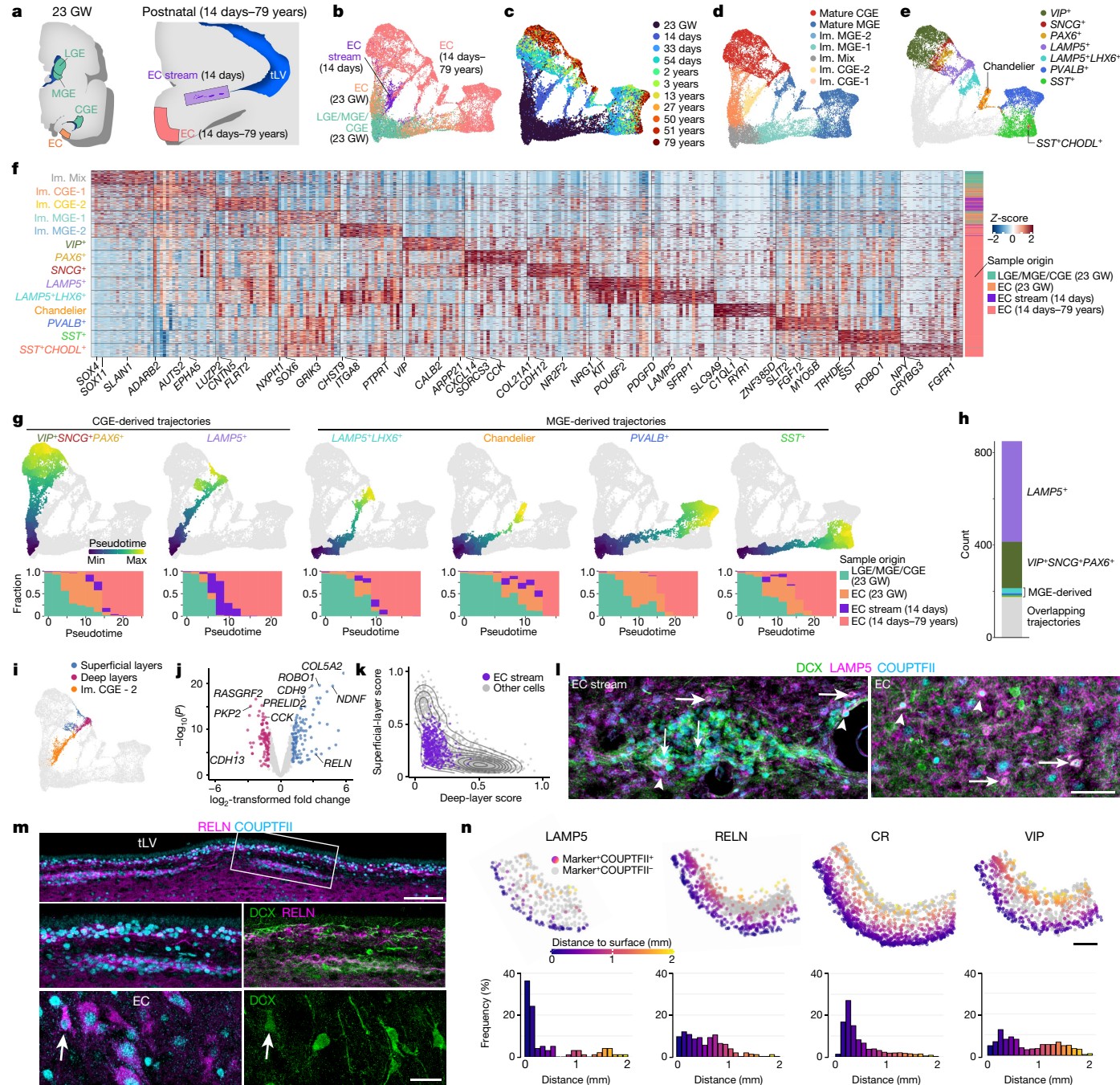

**Fig. 6 | The EC stream mainly supplies CGE-derived *LAMP5*⁺ interneurons.**
**a**, Samples collected for snRNA-seq. **b**–**e**, Uniform manifold approximation and projection (UMAP) plots of maturing interneurons showing the sample origin (**b**), donor age (**c**), subpallial lineages (**d**) and mature neuronal types (**e**). **f**, Heat map of the top differentially expressed (DE) protein-coding genes in each cell identity shown in **d**,**e**. **g**, Top row, main maturation trajectories inferred from Monocle. Bottom row, histograms showing the contributions of each sample origin to each trajectory along pseudotime. **h**, Number of cells from the EC stream in each of the trajectories shown in **g**. **i**, Spatial transcriptomic label transfer[24] identifies superficial- and deep-layer *LAMP5* subpopulations. **j**, Volcano plot of a DE analysis (quasi-likelihood F-test based on pseudobulk counts aggregated by donor) between superficial-layer (blue) and deep-layer

(magenta) *LAMP5*⁺ interneurons. **k**, Module scores for superficial- and deep-layer *LAMP5*⁺ cells. Density lines drawn for non-EC stream cells (grey; see Extended Data Fig. 13). **l**, At birth, LAMP5⁺DCX⁺ (arrows) and LAMP5⁺COUPTFII⁺ (arrowheads) young neurons are present in the EC stream and in the EC. **m**, At birth, RELN⁺DCX⁺COUPTFII⁺ cells are present in streams of migrating neurons close to the ventral temporal lobe VZ. In the EC, many RELN⁺ cells express COUPTFII and a subset are DCX⁺ (arrow). **n**, Top, maps of cells in the EC at birth co-expressing markers of CGE-derived (COUPTFII⁺) interneuron subpopulations: LAMP5, RELN, calretinin (CR) and VIP. Bottom, their frequency distribution across cortical layers. Scale bars, 50 μm (**l**); 1 mm (**n**); 100 μm (**m** top right); 20 μm (**m** bottom right).

In the mouse brain, a migration from the CGE into the hippocampus and EC has been described in embryonic stages[33], and in the macaque a similar migration might also be restricted to embryonic development. This developmental process might be expanded in time in the

larger human brain with increased proliferation and a greatly expanded CGE. Consistently, we observed that the human CGE is very prominent during mid–late gestation and has many more dividing cells as compared with surrounding brain regions (Fig. 5g,h and Extended Data

Figs. 6b,c, 8 and 9). Notably, the CGE was still proliferative at birth, with scattered clusters of dividing cells intermixed with young neurons with migratory morphology (Fig. 5e–h and Extended Data Fig. 7). This suggests that some level of neurogenesis continues postnatally in what is left of the CGE, and might contribute to the postnatal EC stream. Notably, the aberrant late expansion of CGE progenitors in human brain organoids is linked to mutations in the tuberous sclerosis complex genes *TSC1* and *TSC2*, which can result in the formation of periventricular tumours and cortical dysplasia[34].

Histologically and transcriptomically, we observed a subpopulation of RG in the EC stream (Figs. 4b and 5b–d). A subset of these RG are Ki-67+ at birth, and have a transcriptomic trajectory that links them to *EOMES*+ intermediate progenitors and *TBR1*+ immature excitatory neurons. This suggests that excitatory neurogenesis continues postnatally at low levels in the region of the EC stream. It has been suggested that pallial RG can generate inhibitory neurons that are transcriptionally similar to CGE-derived interneurons[35]. However, we did not capture an intermediate progenitor population linking RG to the interneurons in the EC stream. Instead, the majority of the EC stream cells had properties associated with a CGE origin.

In addition to the EC, the prefrontal cortex and the hippocampus continue to recruit new neurons in infants and children, respectively, and the EC is heavily interconnected with both of these regions. In the hippocampus, new neurons are recruited specifically within the dentate gyrus, which receives afferents from the superficial layers of the EC. Our data[10], and those of others[3,36], indicate that dentate neurogenesis rapidly decreases postnatally, as is the case for the EC, with few—if any—new neurons added in adults. However, others have suggested that dentate neurogenesis continues in adults[11,37]. Future studies could investigate whether the postnatal recruitment of new neurons in the EC and in the hippocampus are functionally linked.

The V-SVZ is a common route of passage for tangentially migrating interneurons in the forebrain[8,9,38,39]. The RMS, a major conduit for interneurons, emerges next to the fused olfactory ventricle[16]. In this study, we identify a ventricular fusion in the human temporal lobe that occurs prenatally and provides a scaffold for neuronal migration. Similar to the RMS[16], the migratory neurons in the EC stream migrate in chains and are surrounded by glial cells. These periventricular routes provide a scaffold for tangential migration as brain size and anatomical complexity increase and germinal zones become greatly separated from neuronal destinations[40]. The region of the EC stream is likely to be a route for the tangential migration of multiple neuronal subtypes and not only that of CGE-derived neurons. This possibility is supported by the presence of LHX6+ cells in the EC stream at birth and a sharp decrease in the relative abundance of LHX6+ cells among immature neurons in the EC postnatally.

The EC performs multisensory integration and, as the main source of cortical inputs to the hippocampus, is essential for declarative memory[1,2,41]. This distinctive cortical region contains diverse neuronal subtypes[3], including spatially tuned grid cells[4], and is a brain region in which neuronal loss is first noticed in Alzheimer's disease (AD)[42,43]. The onset of memory symptoms in AD coincides with neuronal loss and non-convulsive seizures in the EC, reflecting an excitation–inhibition imbalance[44]. Notably, projection neurons to the hippocampus in layer 2 of the EC are particularly sensitive to reduced activity[45], which could result from an imbalance in the activity of different subtypes of inhibitory neurons. Although multiple interneuron subtypes are affected in the temporal lobe of patients with AD (ref. 22), recent work suggests that *LAMP5*+ interneurons are particularly vulnerable in patients with AD and in mouse models of the disease[46]. Our results show that the postnatal human EC stream delivers young neurons that are primed to become upper-layer *LAMP5*+*RELN*+ interneurons, a population that has recently been found to be sharply reduced in patients with preclinical AD (ref. 7). Furthermore, another recent study[6] found that the loss of *LAMP5*+*RELN*+ interneurons in the prefrontal cortex is tightly linked to cognitive decline in patients with AD. It will be interesting to determine the functional contribution of these postnatally derived *LAMP5*+*RELN*+ neurons, and to ascertain whether their late arrival in the human brain makes them particularly vulnerable to neurodegeneration.

Our findings highlight how basic cellular processes of brain development, such as neuronal migration and recruitment, are protracted in humans compared with other species. The human EC continues to recruit GABAergic inhibitory neurons postnatally in a cellular mechanism that occurs concurrently with the functional maturation of the EC's capacities for multisensory integration, learning and the development of episodic memory[47]. GABAergic signalling and the maturation of local circuit inhibitory cells are crucial for periods of enhanced plasticity during development[48–50]. The protracted maturation of postnatally recruited inhibitory neurons could provide the EC with a wide time period of plasticity, as these cells integrate complex information from other cortical regions.

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

## Methods

### Collection of human tissue

Fifty-three post-mortem specimens and two post-operative neurosurgical resections were collected for this study (Supplementary Table 1). Tissue was collected with previous consent from the patient or from their next-of-kin in strict observance of the legal and institutional ethical regulations in accordance with each participating institution: (1) The University of California, San Francisco (UCSF) Committee on Human Research. Protocols were approved by the Human Gamete, Embryo and Stem Cell Research Committee (Institutional Review Board) at UCSF (10-02693). (2) The Ethical Committee for Biomedical Investigation, Hospital la Fe (2015/0447) and the University of Valencia Ethical Commission for Human Investigation. (3) In accordance with institutional guidelines and study design approval by the committee for research on decedents (CORID) at the University of Pittsburgh. (4) Specimens collected at the University of Pittsburgh Medical Center had University of Pittsburgh IRB-approved research informed consents along with HIPAA authorizations signed by parents or responsible guardians. We collected tissue blocks from the temporal lobe, anteriorly from the amygdaloid complex to the posterior end of the inferior horn of the lateral ventricle. Samples were either flash-frozen or fixed in 4% paraformaldehyde (PFA) or 10% formalin for more than 24 h (see Supplementary Table 1). Brains were cut into around 1.5-cm blocks, cryoprotected in a series of 10%, 20% and 30% sucrose solutions and then frozen in an embedding medium, OCT. Blocks of the medial temporal lobe were cut into 20-µm sections on a cryostat (Leica CM3050S) and mounted on glass slides for immunohistochemistry.

### Collection of tissue from non-human primates

All experiments were conducted in accordance with Fudan University Shanghai Medical College and University of Pittsburgh guidelines. Macaque monkeys, *M. mulatta*, (Supplementary Table 2), were obtained from the Kunming Primate Research Center of the Chinese Academy of Sciences, Suzhou Xishan Zhongke Laboratory Animal Co. and the University of Pittsburgh. For immunofluorescent staining, macaques were deeply anaesthetized and then transcardially perfused with saline (0.9%) followed by fixation in 4% PFA for 12–48 h before sucrose cryoprotection. Floating sections were prepared at 40 µm on a sliding microtome or cryostat and stored in cryoprotectant (glycerol-ethylene glycol) solution at −20 °C until immunohistochemical processing, which followed the same protocol as used on the human samples.

### Immunohistochemistry

Frozen slides were allowed to equilibrate to room temperature for 3 h. Some antigens required antigen retrieval (Supplementary Table 2), which was conducted at 95 °C in 10 mM Na citrate buffer, pH = 6.0. After antigen retrieval, slides were washed with TNT buffer (0.05% TX100 in phosphate-buffered saline (PBS)) for 10 min, placed in 1% $H_2O_2$ in PBS for 1.5 h and then blocked with TNB solution (0.1 M Tris-HCl, pH 7.5, 0.15 M NaCl and 0.5% blocking reagent from Akoya Biosciences) for 1 h. Slides were incubated in primary antibodies overnight at 4 °C (Supplementary Table 3) and in biotinylated secondary antibodies (Jackson Immunoresearch Laboratories) for 2.5 h at room temperature. All antibodies were diluted in TNB solution. For most antibodies, the conditions of use were validated by the manufacturer (antibody product sheets). When this information was not provided, we performed control experiments, including negative controls with no primary antibody and comparisons to mouse staining patterns.

Sections were then incubated for 30 min in streptavidin-horseradish peroxidase that was diluted (1:200) with TNB. Tyramide signal amplification (Perkin Elmer) was used for some antigens. Sections were incubated in tyramide-conjugated fluorophores for 5 min at the following dilutions: fluorescein, 1:100; Cy3, 1:100; and Cy5, 1:100. After several PBS rinses, sections were mounted in Fluoromount G (SouthernBiotech)

and coverslipped. Staining was conducted in technical triplicates before analysis.

### Fluorescent microscopy, image processing and quantification

Images were acquired on Leica TCS SP8 or SP5 confocal microscopes using 10×/0.45 NA for tilescans, and 20×/0.75 NA or 63×/1.4 NA objective lenses. Imaging files were analysed and quantified in Neurolucida software (MBF Bioscience 2019 version). Linear adjustments to image brightness and contrast were made equivalently across all images using Adobe Photoshop (2023). Cells were counted in z-stack images from sections stained for Ki-67. Quantifications at each age were generated from three to five representative images. Experimental replicates and different co-stains were also analysed. The fluorescence signal for single reactivity and co-localization of immunoreactivity was counted individually using the markers function in the Neurolucida imaging software. The quantification of data was performed with GraphPad Prism (v.9) and R (v.4.1).

### Tissue clearing and staining, light-sheet microscopy and reconstruction of the EC stream

Human samples were cut into sections (around 1 cm thick) containing the medial temporal lobe and were cleared and stained following published protocols[51]. Samples were dehydrated in a graded series of methanol (1 h each) from 20%, 40% and 60% to 100% (×2), followed by overnight incubation in 5% $H_2O_2$ in methanol at 4 °C and rehydration in a graded series of 80%, 60%, 40% and 20% methanol, concluding with PBST solution (PBS with 0.2% Triton X-100) (×2), followed by 36 h in a PBST–DMSO–glycine solution (PBS, 0.2% Triton X-100, 20% dimethyl sulfoxide (DMSO) and 0.3 M glycine) at 37 °C. Immunostaining was performed by blocking in PBS, 0.2% Triton X-100, 10% DMSO and 6% normal donkey serum (NDS) at 37 °C for two days followed by incubation of primary antibodies (1:100) and TOPRO (1:5,000) in PBS-Tween 0.2% with heparin 10 µg ml$^{-1}$ (PTwH), 5% DMSO and 3% NDS at 37 °C for seven days, refreshing solutions every other day. Washes were 5× in PTwH over 24 h followed by secondary antibodies (for example, donkey anti-rabbit-Alexa 555 in PTwH and 3% NDS) at 1:500 at 37 °C for four days and washing 5× in PTwH over 24 h. Next a graded series (1 h) of 20%, 40%, 60%, 80% and 100% (×2) methanol was performed before overnight incubation in 1:2 methanol to dichloromethane (DCM; Sigma 270997). Samples were washed for 20 min (×2) in 100% DCM before placing in dibenzyl ether (Sigma 108014) until clear (30 min). Cleared samples were imaged on a light-sheet microscope (custom-built AZ-100 by Nikon Imaging Center, UCSF) using a sCMOS camera (Andor Neo) and a ×2 (WD = 45 mm, NA = 0.2) or ×5 (WD = 15 mm, NA = 0.5) objective lens. Scans were made at ×2 magnification with a step-size of 16 µm. The stacks were analysed with Imaris v.9.7.1 and images were processed with a Gaussian blur and background subtraction filters before volumes were rendered. The EC stream and the ventricle were drawn with no more than 25 sections of distance and reconstructed in 3D.

### Mapping and quantification of young migrating neurons

To generate maps of the location of the somas of different cell types, two-dimensional tilescans were generated using Neurolucida (10×/0.45 NA or 20×/0.8 NA objective) and contours or markers were placed to indicate the location of each cell type. To generate rose histograms, the angle of the leading process was measured using ImageJ. Leading processes were identified as one single process extending from a DCX$^+$ cell soma. The frequency of DCX$^+$ cell orientations within each segment of the tissue was plotted as a 360-degree histogram using R (v.4.1), and then histograms were overlaid on the map of the tissue within Adobe Illustrator (2022). For vector orientation maps, to determine the angle of migration of each neuron, the angle formed by the shortest path connecting the cell body to the cortical surface and the cell's leading process was measured. If the resulting angle was between −45° and 45°, cells were classified as migrating radially towards the cortical surface

(radial(in)). Cells were classified as migrating tangentially if the angle was between 45° and 135° or −45° and −135°. Cells were classified as radially migrating away from the ventricle if the angle was between −135° and 135° (radial(out)). Plots were done using the ggplot2 package (v.3.3.5) in R (v.4.1).

### Reproducibility
All representative images of immunostainings correspond to findings observed in at least three independent stainings.

### TEM
Temporal lobe tissue fixed with 2.5% glutaraldehyde-2% PFA in 0.1 M phosphate buffer (PB) was transversely sectioned at 200 µm using a Leica VT1200S vibratome (Leica Microsystems). Slices were further post-fixed in 2% osmium tetroxide in 0.1 M PB for 1.5 h at room temperature, washed in deionized water and partially dehydrated in 70% ethanol. Samples were then incubated in 2% uranyl acetate in 70% ethanol in the dark for 2.5 h at 4 °C. Brain slices were further dehydrated in ethanol followed by propylene oxide and infiltrated overnight in Durcupan ACM epoxy resin (Fluka, Sigma-Aldrich). The following day, fresh resin was added and the samples were cured for 72 h at 70 °C. After resin hardening, semithin sections (1.5 µm) were obtained and lightly stained with 1% toluidine blue for light microscopy. Ultrathin sections (70–80 nm) were obtained with a diamond knife using a Ultracut UC7 ultramicrotome (Leica), stained with lead citrate and examined under a FEI Tecnai G2 Spirit transmission electron microscope at 80 kV (FEI Europe) equipped with a Morada CCD digital camera (Olympus Soft Image Solutions).

### Tissue processing for snRNA-seq
Snap-frozen samples of entorhinal cortex, EC stream and germinal zones in gestation were obtained from the University of Maryland Brain Bank or the NIH Neurobiobank (for case list, see Supplementary Table 1) or from UCSF. Samples were sectioned (50 µm) on a cryostat. To collect samples of distinct regions, a ophthalmology microscope (OPMI 6, Zeiss) was positioned over the cryostat and the different regions of interest (germinal zones, EC stream and EC) were dissected using a small stab knife (Sharpoint stab knife 22.5°, Surgical Specialties 72–2201) and collected in a nuclease-free microcentrifuge tube (1.5 ml DNA LoBind Tubes, Eppendorf, 022431048) and stored at −80 °C until RNA quality was assessed and nuclei extracted. For each sample, total RNA was extracted from 40 mg of tissue using a TissueRuptor II (Qiagen) and a RNeasy Mini Kit (Qiagen, 74104). RNA integrity was measured on the Agilent 2100 Bioanalyzer using the RNA Pico Chip Assay. Only samples with a RNA integrity number (RIN) > 7 were used (mean: 7.78).

### Isolation of nuclei and snRNA-seq
Frozen sections were transferred from tubes in dry ice to ice cold lysis buffer (0.01 M Tris-HCl, 0.14 M NaCl, 1 mM $CaCl_2$, 0.02 M $MgCl_2$, 0.03% Tween-20, 0.01% BSA, 10% Nuclei Ez Lysis Buffer (Sigma), and 0.2 U µl$^{-1}$ Protector RNAse inhibitor (Sigma) in diethyl pyrocarbonate (DEPC)-treated water) and the tissue was dissociated using a glass dounce homogenizer (Thomas Scientific, 3431D76). Tissue homogenates were transferred to a separate 30-ml thick-walled polycarbonate ultracentrifuge tube (Beckman Coulter, 355631). Sucrose solution (1.8 M sucrose, 3 mM MgAc2, 1 mM DTT, 10 mM Tris-HCl in DEPC-treated water) was added at the bottom of the tube and homogenates were centrifuged at 107,000$g$ for 2.5 h at 4 °C. The supernatant was discarded, and the nuclei pellet was incubated in 200 µl wash/resuspension buffer (WRB: 1% BSA and 0.2 U µl$^{-1}$ Protector RNAse inhibitor in DEPC-based PBS) for 20 min on ice before resuspending the pellet. For multiplexed runs, resuspended nuclei were filtered twice through a 30-µm strainer. Each one of the five nuclei suspensions was incubated with its own unique CellPlex Cell Multiplexing Oligo (CMO) barcode (10x Genomics) for 5 min followed by a 500$g$ centrifugation at 4 °C for 10 min.

After centrifugation, the supernatant was discarded and nuclei were gently resuspended in WRB using a P200 pipette for a minimum of 2 min, until a clean single-nuclei suspension was obtained. After two more centrifugations and WRB washes to remove unbound CMOs, 10 µl of each nuclei suspension was stained with DAPI and counted on a haemocytometer. Samples were pooled together and diluted to 2,000 nuclei per µl. A target capture of 30,000 nuclei per well in two wells of a G chip (10x Genomics) was used. Gene expression and barcode libraries were prepared in parallel using the Chromium Next GEM Single Cell 3′ Kit v3.1 (10x Genomics) according to the manufacturer's instructions. A detailed step-by-step protocol can be found at https://doi.org/10.17504/protocols.io.kqdg369keg25/v2. For non-multiplexed assays, after centrifugation, each tissue homogenate was gently resuspended in WRB, filtered twice, counted, centrifuged and resuspended at 1,000 nuclei per µl. A total of 16,500 nuclei were loaded in each of the 10x chip G wells, aiming for a 10,000 nuclei recovery per sample. Gene-expression libraries were prepared using the Chromium Next GEM Single Cell 3′ Kit v3.1(10x Genomics). A detailed protocol can be found at: https://doi.org/10.17504/protocols.io.j8nlkk9ydl5r/v2. All libraries were sequenced in a Novaseq 6000 system (Illumina).

### Read alignment, quality control and integration with adult EC dataset
Sequencing reads were aligned using the CellRanger Count v.7 pipeline (10x Genomics) to an optimized reference based on the human (GRCh38) 2020-A reference[52]. FASTQ files generated in this study and those retrieved from the Gene Expression Omnibus (GEO) from a previous study from the adult EC[3] were aligned and pre-processed using the same workflow. For each sample, we used quality control and pre-processing steps that consisted of, first, using SoupX[53] to reduce the effect of ambient RNA in the transcriptional profile in each nucleus. Next, we discarded low-quality nuclei by removing those barcodes with low unique molecular identifier (UMI) counts (bottom 2% of UMI counts or clustered with them) and barcodes with more than 5% of reads belonging to mitochondrial genes. After that, in non-multiplexed libraries, we used DoubletFinder[54] to eliminate likely doublets. In addition, any clusters composed of more than 60% of assigned doublets were removed. In multiplexed libraries, we removed barcodes assigned as 'doublets' or 'ambiguous' by freemuxlet, keeping only barcodes assigned as 'singlets'. At the end, we used DropletQC[55] to calculate the nuclear fraction (the fraction of reads that contained an intronic region) to identify nuclei with low amounts of unprocessed mRNA. For this, we also aligned our data to the standard reference used by 10x in CellRanger. We discarded nuclei with nuclear fraction values < 0.5 (ref. 56).

### Sample demultiplexing and doublet removal
Freemuxlet, a genetic demultiplexing tool in the popscle suite (v.0.1) (https://github.com/statgen/popscle), was used to genetically demultiplex cell donors from Cellplex multiplexed runs, because the CMO labelling was insufficient to fully demultiplex nuclei from snap-frozen tissue. The bam files generated in the CellRanger Count pipeline were used as inputs for freemuxlet and a customized VCF file from the 1000 genomes data filtered for high variant confidence, minor allele frequency (MAF 0.01) and exonic variants as a reference for SNPs. Freemuxlet assigns each droplet barcode to a single donor, or multiple donors in case a droplet contains cells from distinct samples. Donor to sample matching was done by assessing the amount of each CMO in all cells of a designated donor.

### Merging, normalization, dimensionality reduction and clustering
Datasets of each individual sample were merged and normalized using the NormalizeData function in Seurat v.4.2 (ref. 57). In brief, UMI counts were divided by the total UMI counts in each cell, multiplied by a scale factor (10,000) and then log-transformed. Principal component (PC)

analysis was performed in the normalized data and 22 PCs (dataset with all cells) or 8 PCs (interneuron maturation dataset) were used for UMAP dimension reduction. For the dataset containing only the EC stream sample (Figs. 4 and 5) we used SCTransform as the normalization method and 80 PCs for UMAP dimension reduction.

## Label transfer and assignment of interneuron type

Label transfer analyses were performed using Seurat v.4.2 (ref. 57). For label transfer analyses, we subset the interneuron maturation dataset to contain only clusters composed of mature interneurons found in the postnatal cortex and used them as the query dataset. To label the neuronal populations in the postnatal EC, we used a middle temporal gyrus dataset[22] as reference. For the identification of superficial- and deep-layer *LAMP5*+ cells, we used a spatial transcriptomic dataset that identified neuronal populations preferentially located in distinct cortical layers in the temporal lobe[24].

## Pseudotime estimation

For pseudotime estimation we used Monocle 3 (ref. 58) using nn.k = 10, minimal_branch_len = 30, euclidean_distance_ratio = 3 and geodesic_distance_ratio = 0.1 as parameters for the reversed graph embedding function. The node closest to the MGE–CGE branching point was selected as the root node. The six main trajectories were inferred by using the choose_graph_segments function to select cells in the trajectory paths connecting the root node to end-points corresponding to each of the six major classes of adult inhibitory neurons.

## Gene co-expression analysis

To identify modules of co-expressed genes throughout interneuron maturation, we performed a weighted gene co-expression network analysis (WGCNA) using the hdWGCNA package for R[59]. In brief, using the interneuron maturation dataset, we created pseudobulk meta-cells aggregating nuclei that belonged to the same sample using a nearest-neighbour parameter $k$ of 20 and setting the maximum number of shared nuclei between metacells to 10.

## Differential gene expression and module scores

To identify representative genes for each cell identity used in heat maps, we used a Wilcoxon rank sum test in Seurat v.4.2 (ref. 57) and filtered for genes expressed in at least 50% of cells in each identity. For differential expression analysis between superficial- and deep-layer *LAMP5*+ cells, we used a quasi-likelihood F-test on pseudobulk counts aggregated by donor[60] (edgeR package v.3.42). The results of these tests can be found in Supplementary Tables 4–6 and 8. Module scores for superficial- and deep-layer identities were calculated using the AddModuleScore function in Seurat v.4.2. For that, the 10 DE genes with the highest fold change and $P < 0.01$ for superficial- or deep-layer cells were used. Module scores were rescaled to values between 0 and 1.

## Reporting summary

Further information on research design is available in the Nature Portfolio Reporting Summary linked to this article.

## Data availability

The raw data for the newly generated snRNA-seq data are available on Database for Genotypes and Phenotypes (dbGaP, accession phs003509. v1.p1). Count matrices for each sample and metadata were deposited in NCBI's Gene Expression Omnibus (GEO series GSE199762). Processed data used in this study can be browsed and downloaded on Cellxgene. Raw data for additional adult human EC snRNA-seq can be found through GEO series accession number GSE186538 (ref. 3).

## Code availability

The code used for the data analysis is available at https://github.com/massisnascimento/ECstream.

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

**Acknowledgements** We thank the families of tissue donors, whose generosity made this research possible; the University of Maryland Brain Bank and the NIH Neurobiobank for help with collecting human samples; L. Wang, T. Mukhtar, O. Pastor Alonso and M. Paredes for help with additional sample collection; and J. Fudge and J. Cameron for macaque samples and discussions. Sequencing was performed at the UCSF Center for Advanced Technology, supported by UCSF PBBR, RRP IMIA and NIH 1S1OOD028511-01 grants. A.A.-B., M.A.N. and E.J.H. were supported by the National Institute of Neurological Disorders and Stroke (NINDS) (A.A.-B and M.A.N., grant R01NS028478; A.A.-B. and E.J.H., grant P01NS083513; E.J.H., grant R01NS132595-01). A.A.-B. was supported by the UCSF Program for Breakthrough Biomedical Research, funded in part by the Sandler Foundation. S.F.S. and S.B. were supported by R01MH128745, R21MH125367, a Competitive Medical Research Fund award from the University of Pittsburgh Medical Center and startup funds from the University of Pittsburgh. S.S. was supported by T32NS007433 and a Mellon Fellowship. J.M.G.-V. was supported by the Valencian Council for Innovation, Universities, Science and Digital Society (PROMETEO/2019/075) and V.H.-P. was supported by the Spanish Ministry of Science, Innovation and Universities (PCI2018-093062). Z.Y. and Z.Z. were supported by the National Natural Science Foundation of China (NSFC31820103006 and NSFC32200776). C.J.Y. was supported by NIH grants R01HG011239 and R01AI136972, and by the Chan Zuckerberg Initiative. C.J.Y. is an investigator at the Chan Zuckerberg Biohub and is a member of the Parker Institute for Cancer Immunotherapy.

**Author contributions** M.A.N., A.A.-B. and S.F.S. conceived the experiments, conceptualized and wrote the manuscript. M.A.N. conducted the snRNA-seq experiments and analysed the data. M.A.N. and R.B. demultiplexed samples with guidance from C.J.Y. TEM was performed and analysed by V.H.-P. and J.M.G.-V. Acquisition of human entorhinal cortex samples was coordinated by M.A.N., S.F.S., T.J.A., J.S.R.-M., A.R.K. and E.J.H. Macaque immunostaining was performed by Z.Z., Z.Y., S.B., S.S. and S.F.S. Human samples were processed, stained, and confocal or light-sheet imaged by S.F.S., S.B. and S.S. Image analysis, quantification and mapping was performed by M.A.N. S.B., S.S., S.F.S. and M.A.N. performed the 3D reconstruction of the EC stream. All authors approved the final manuscript.

**Competing interests** A.A.-B. is a co-founder and is on the scientific advisory board of Neurona Therapeutics. A.R.K. is a co-founder, consultant and director of Neurona Therapeutics. C.J.Y. is a founder for and holds equity in DropPrint Genomics (now ImmunAI) and Survey Genomics; is a member of the scientific advisory board for and holds equity in Related Sciences and ImmunAI; is a consultant for and holds equity in Maze Therapeutics; is a consultant for TReX Bio, HiBio, ImYoo and Santa Ana; and is also an Innovation Investigator for the Arc Institute. C.J.Y. has received research support from the Chan Zuckerberg Initiative, the Chan Zuckerberg Biohub, Genentech, BioLegend, ScaleBio and Illumina.

**Additional information**
**Correspondence and requests for materials** should be addressed to Marcos Assis Nascimento, Arturo Alvarez-Buylla or Shawn F. Sorrells.

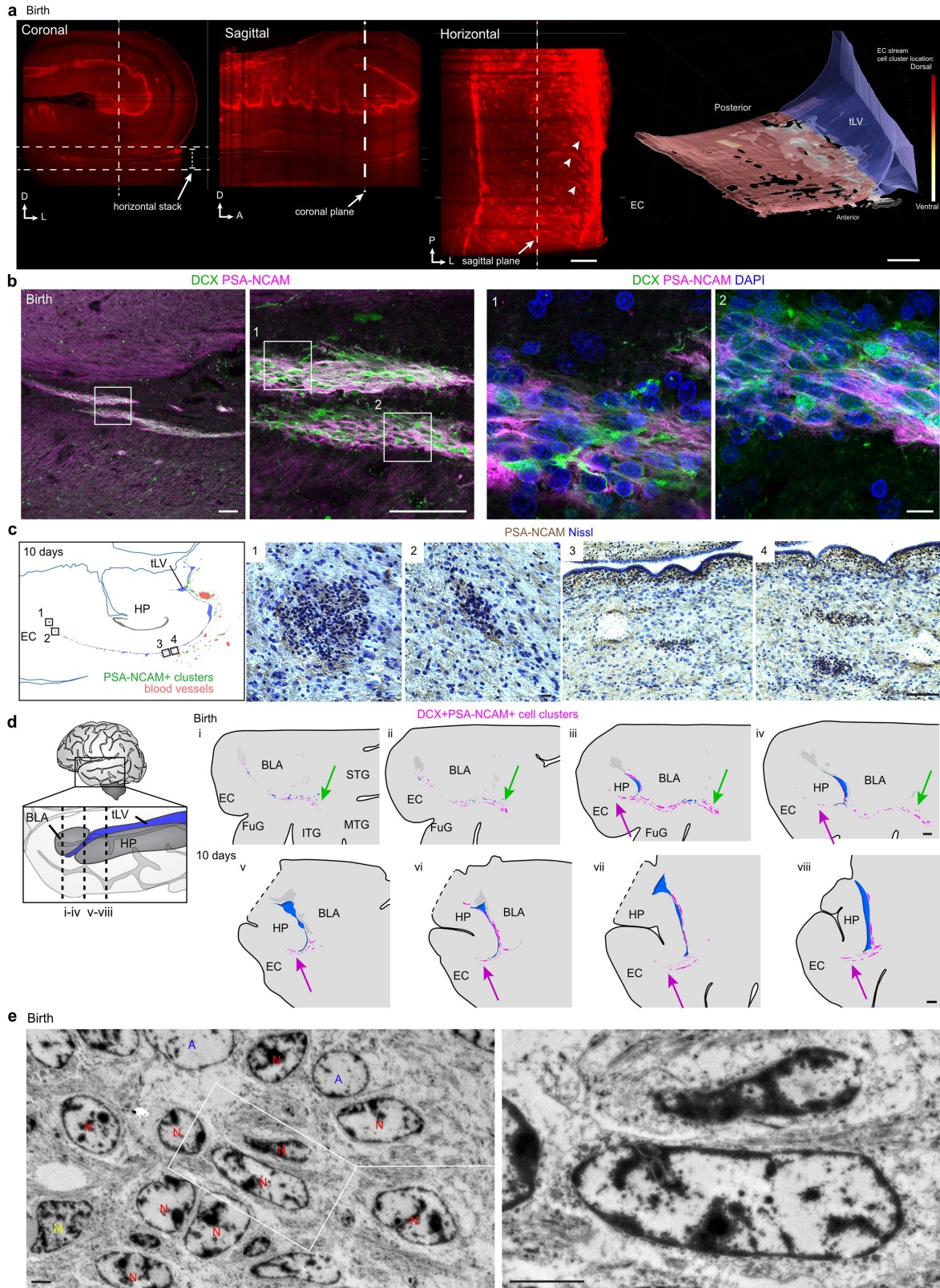

**Extended Data Fig. 1** | See next page for caption.

**Extended Data Fig. 1 | The EC stream of migratory neurons extends medially from the temporal lateral ventricle as a multilayered lamina along the anterior–posterior axis. a**, Three orientations of the cleared tissue block in Fig. 1a stained with TOPRO and imaged using a light-sheet microscope. Dense chains emerging from the ventricle can be seen escaping from the ventricle in horizontal view (arrowheads). Reconstruction of the EC stream is shown from a ventral view facing anterior-medially. **b**, Immunostained sections of the EC stream at birth showing dense clusters of DCX⁺PSA-NCAM⁺ cells. **c**, Map at 10 days postnatal showing the location of insets (1–2) with a cross section of DAB immunostained PSA-NCAM⁺ cell clusters. Boxes 3 and 4 show a location next to the ventricle with DAB immunostained PSA-NCAM⁺ cells in a continuous layer next to the ventricle and in distinct clusters at varying distances from the ventricle. **d**, Map at birth of DCX⁺PSA-NCAM⁺ cell clusters in a coronal series of sections of the medial temporal lobe. Clusters distributed along a medial-lateral orientation are present ventral to the basolateral amygdala between the EC and medial temporal gyrus (i-iv). The EC stream (arrow) is visible from the first section containing the temporal lobe ventricle and at every section examined posteriorly across the hippocampus (iii-viii). Note at levels iii and iv an additional laterally oriented migratory stream (green arrowhead). **e**, TEM of immature neurons in the human EC stream at birth showing densely packed neurons (red, N) with compacted chromatin and fusiform morphology (inset), astrocytes (blue, A) and microglia (yellow, M). Scale bars: 1 mm (**a**, **d**), 100 μm (**b** left panels, **c** 3, 4), 20 μm (**c** 1,2), 10 μm (**b** right 1,2), 2 μm (**e**). Abbreviations: BLA basolateral amygdala; EC: entorhinal cortex; FuG: fusiform gyrus; HP: hippocampus; ITG: inferior temporal gyrus; MTG: medial temporal gyrus, STG: superior temporal gyrus; sub: subiculum; tLV: temporal lobe lateral ventricle.

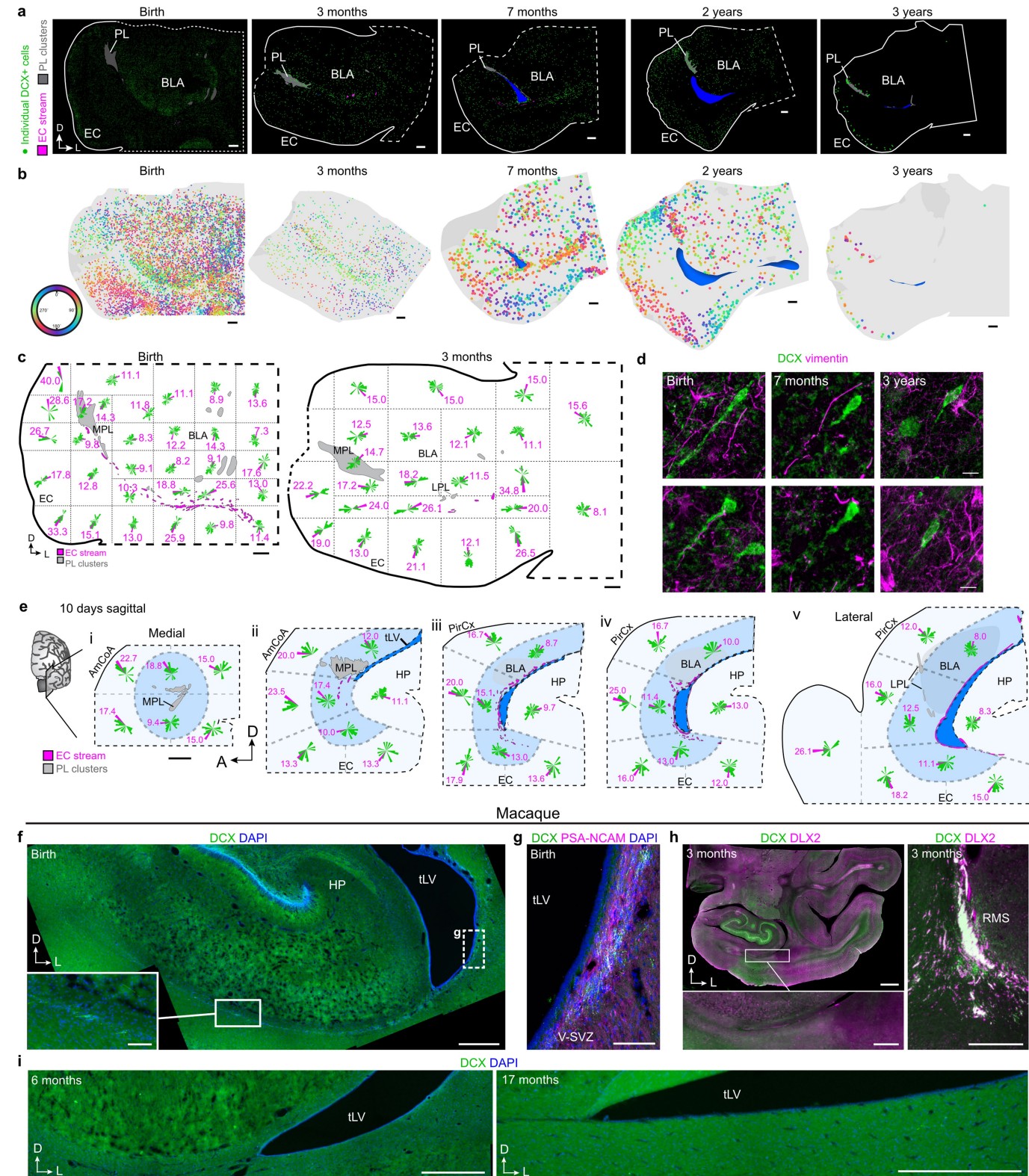

**Extended Data Fig. 2** | See next page for caption.

**Extended Data Fig. 2 | Individual DCX⁺ neurons migrate into the EC from birth to three years of age. a**, Maps of DCX⁺ neurons (green) in coronal sections between birth and 3 years. **b**, DCX⁺ cell orientations between birth and 3 years of age in the same sections shown in Fig. 2b. Here colours depict the direction of the leading process, with colours representing a 180 degree arc. **c**, Coronal maps at birth and 3 months of age segmented into sectors (dotted lines) with rose frequency histograms. **d**, Individual DCX+ neurons at birth, 7 months, and 3 years of age in the EC and their relationship to vimentin+ processes. **e**, Sagittal maps at 10 days postnatal from medial (i) to lateral (v) segmented into sectors (dotted lines). Within each sector, the orientations of the leading process of all individual DCX+ cells are plotted on a 360-degree (rose) frequency histogram. The magenta bar indicates the percentage of cells that fall into the most frequently occupied orientation within that sector. **f**–**i**, Macaque temporal lobe immunostaining between birth and 17 months.

**f**, Tilescan of the macaque medial temporal lobe at birth showing location of **g** in adjacent section and dispersed DCX⁺ cell labelling in the EC stream region (insets). **g**, DCX⁺PSA-NCAM⁺ cell clusters are present in the lateral wall of the macaque temporal lobe, closer to the CGE region at birth. **h**, Macaque EC stream region at 3 months stained for DCX and DLX2, with no evidence of DCX-immunolabelled neurons (inset) in sharp contrast to the robustly DCX⁺DLX2⁺ RMS (right). **i**, DCX immunostaining uncovers no labelled cells in the EC stream region in the macaque at 6 months and 17 months. Scale bars: 2 mm (**c**, **h** top left), 1 mm (**a**, **b**, **d**), 500 µm (**g** overview, **h** bottom and right, **i**), 100 µm (**f**, **g** inset), 10 µm (**e**). Abbreviations: BLA basolateral amygdala; EC: entorhinal cortex; HP: hippocampus; LPL: lateral paralaminar amygdala; MPL medial paralaminar amygdala; PL: paralaminar amygdala; RMS: rostral migratory stream; tLV: temporal lobe lateral ventricle; V-SVZ: ventricular–subventricular zone.

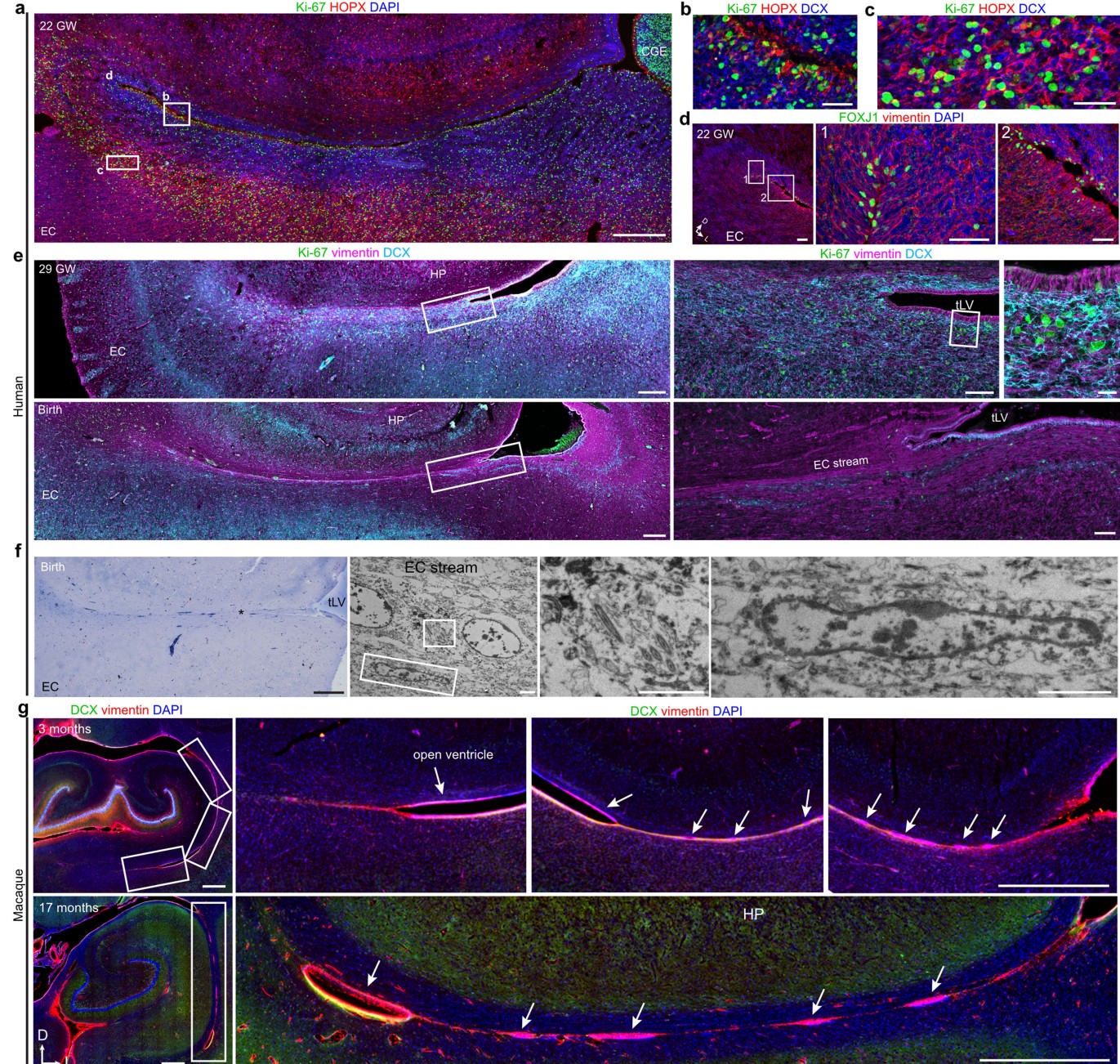

**Extended Data Fig. 3 | The EC stream forms next to a fused ventricle in humans during gestation. a**, EC stream region at 22 GW stained for Ki-67[+] and HOPX[+] cells showing location of insets in **b**–**d**. **b**,**c**, Ki-67[+], HOPX[+], and DCX[+] cells in the VZ (**b**) and outer subventricular zone (**c**) along the closely opposed walls of the ventricle in the EC stream anlage. **d**, Staining of an immediately adjacent section for FOXJ1 and vimentin in the region indicated in **a** shows FOXJ1[+] cells not contacting the ventricle. **e**, At 29 GW (top) and birth (bottom), the ventricle has closed in this region which contains a field of DCX[+] cells at 29 GW and multilayered lamina of coalesced streams at birth between the EC and the V-SVZ. In the wall of the ventricle the ependymal layer contains vimentin[+] cells in the VZ and Ki-67[+] cells in the V-SVZ (insets). **f**, Semithin section stained with toluidine blue showing the EC stream at birth (left), and TEM of cells displaying motile cilia (square inset) and an elongated neuron with ultrastructural features of a migrating cell in the EC stream at birth (right). **g**, Macaque temporal lobe at 3 and 17 months stained for DCX and vimentin showing open ventricle within the same region where a closed ventricle and the EC stream is found in humans (arrows). Scale bars: 1 mm (**g** top row), 500 μm (**a**, **e** left overviews, **f** left, **g** bottom row), 100 μm (**d** left, **e** top middle bottom right), 50 μm (**b**, **c**, **d** middle and right), 20 μm (**e** top right), 2 μm (**f** right panels). Abbreviations: CGE: caudal ganglionic eminence; EC: entorhinal cortex; HP: hippocampus; tLV: temporal lobe lateral ventricle.

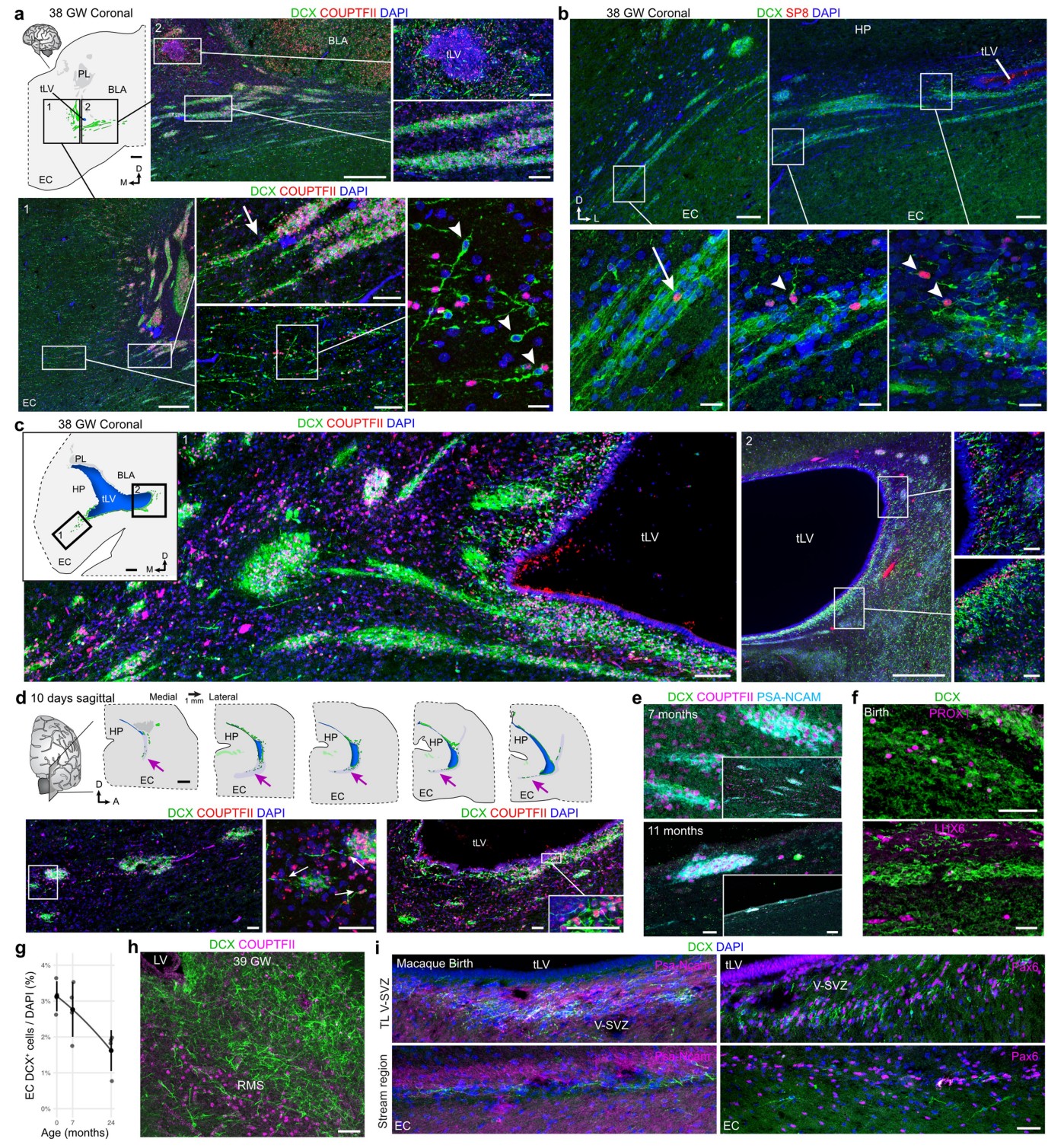

**Extended Data Fig. 4** | See next page for caption.

**Extended Data Fig. 4 | EC stream neurons express COUPTFII between 38 GW and 11 months. a**, Map at 38 GW of DCX$^+$COUPTFII$^+$ cell clusters in a coronal section at the level of the anterior tip of the temporal lobe lateral ventricle (tLV) as shown in Fig. 1d. DCX$^+$COUPTFII$^+$ cells are located within and emanating from dense streams (arrow) at this age alongside individual DCX$^+$ cells (arrowheads). **b**, At 38 GW, DCX$^+$SP8$^+$ cells individually migrating (arrowheads) and within the EC stream (arrow) are infrequent (~2%). **c**, Coronal map at 38 GW of the cell clusters shown in Fig. 1b from a level caudal to the map in (**a**). These DCX$^+$ cells are COUPTFII$^+$ and the stream directed to the EC breaks away from the ventricle at its closest point to the EC. **d**, Sagittal maps at 10 days postnatal spaced by 1 mm showing the whole tissue sections that are magnified in Fig. 1e. Dense DCX$^+$COUPTFII$^+$ cell clusters are present between the ventricle and EC and processes and individual cells are visible extending from the clusters (arrows). **e**, DCX$^+$PSA-NCAM$^+$ COUPTFII$^+$ clusters in the EC stream at 7 months and 11 months and low magnification views of the magnified region shown (inset). **f**, Immunostaining of DCX$^+$ cells in the EC stream at birth shows a small subset co-expressing PROX1 or LHX6. **g**, The percentage of DCX$^+$ nuclei in the human EC between birth and 2 years; n = 3 individuals across 2 independent experiments, data points are individual images, large dots and bars are mean values +/− SD for each individual. **h**, COUPTFII$^-$ DCX$^+$ cells in the human RMS at 39 GW. **i**, At birth in the macaque, immunostaining in the temporal lobe V-SVZ and EC stream region shows dense DCX$^+$PSA-NCAM$^+$ cell clusters that do not co-express PAX6. Scale bars: 2 mm (**d** maps), 1 mm (**a** and **c** maps), 500 μm (**a** insets 1 and 2 left, **c** inset 2 left), 100 μm (**a** inset 1 middle panels inset 2 right panels, **b** top row, **c** inset 1, **e** inset), 50 μm (**c** inset 2 right panels, **d** bottom row, **f**, **h**, **i**), 20 μm (**a** inset 1 right, **b** bottom row, **e** high magnification view). Abbreviations: BLA basolateral amygdala; EC: entorhinal cortex; HP: hippocampus; PL: paralaminar amygdala; RMS: rostral migratory stream; tLV: temporal lobe lateral ventricle; V-SVZ: ventricular–subventricular zone.

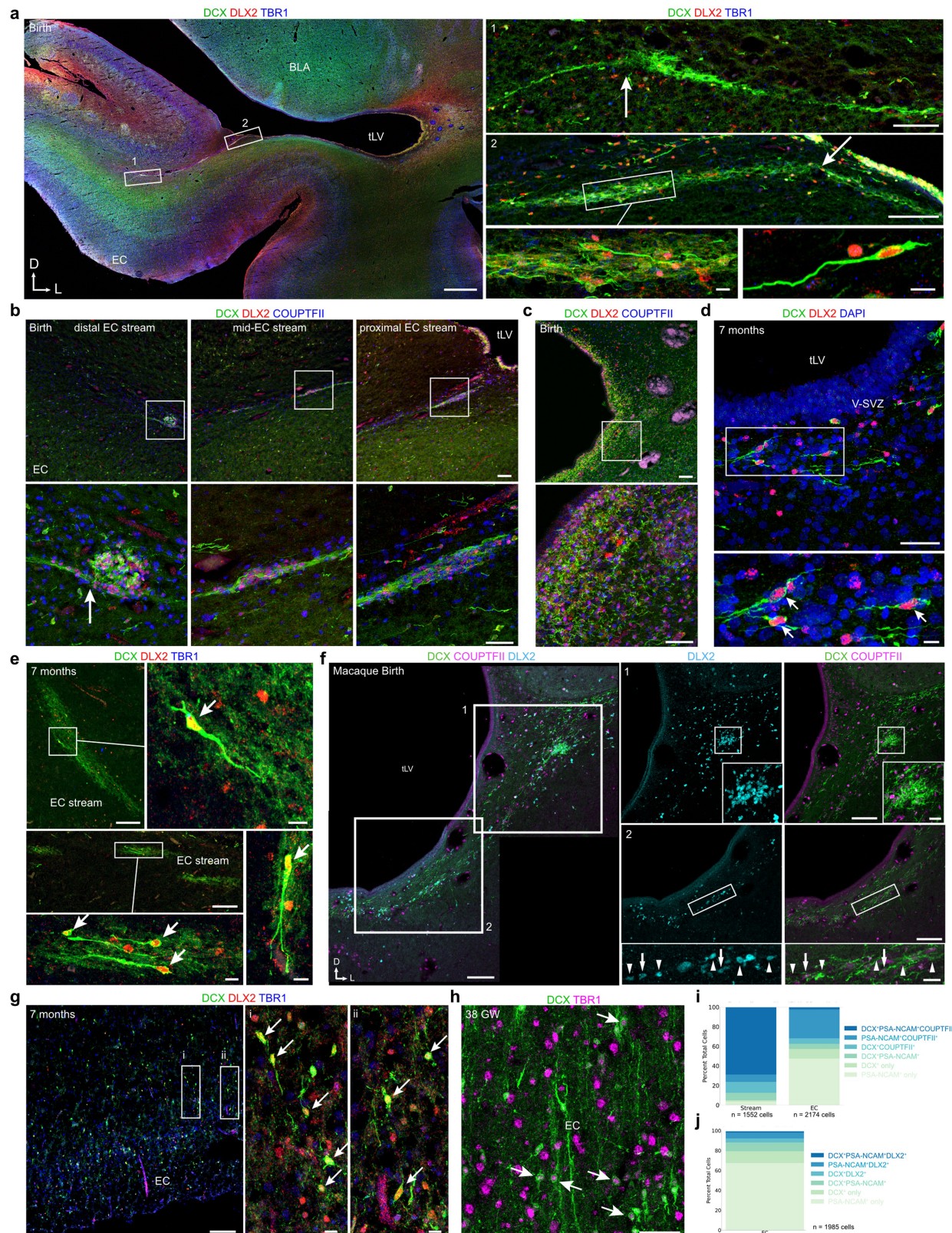

**Extended Data Fig. 5** | See next page for caption.

**Extended Data Fig. 5 | The EC stream supplies interneurons to the temporal lobe. a**, Coronal section at birth showing DCX$^+$DLX2$^+$TBR1$^-$ cells with processes emanating at a branch-point (arrow) from the EC stream (1) and a stream cluster branch-point (arrow) away from the ventricle (2). **b**, DCX$^+$ cells in the EC stream clusters at birth are DLX2$^+$COUPTFII$^+$ at distal, middle, and proximal distances from the tLV. Cells can be observed sending processes out of the dense cell clusters toward the EC (arrow). **c**, In the same section as **b**, the lateral wall of the tLV contains large collections of DCX$^+$DLX2$^+$COUPTFII$^+$ cells. **d**, At 7 months of age, individual DCX$^+$DLX2$^+$ cells are found ventrally to the tLV (arrows). **e**, At 7 months the EC stream contains DCX$^+$DLX2$^+$TBR1$^-$ cells (arrows). **f**, At birth, the macaque temporal lobe V-SVZ contains cell clusters with DCX$^+$DLX2$^+$ cells that are COUPTFII$^-$ (arrowheads), and COUPTFII$^+$ (arrows). **g**, At 7 months the EC contains DCX$^+$DLX2$^+$TBR1$^-$ cells (arrows). **h**, In the EC at 38 GW, DCX$^+$ cells can be observed that co-express the cortical excitatory neuron transcription factor TBR1. These cells typically had a rounded morphology (arrows) and lacked migratory features. **i,j**, Quantifications of marker co-expression by cells in the EC stream and in the EC between birth and 11 months for DCX, PSA-NCAM, and COUPTFII (**i**) or in the EC stream for DCX, PSA-NCAM, and DLX2 (**j**). Scale bars: 1 mm (**a**), 100 µm (**a** insets 1 and 2, **b** top row, **c** top, **e** top and middle left, **f** left overview and inset overviews 1 and 2, **g** left), 50 µm (**b** bottom row, **c** bottom, **d** top, **h**), 20 µm (**f** insets), 10 µm (**a** bottom right insets, **d** bottom, **e** right, bottom, and bottom right insets, **g** insets i and ii). Abbreviations: BLA basolateral amygdala; EC: entorhinal cortex; HP: hippocampus; tLV: temporal lobe lateral ventricle.

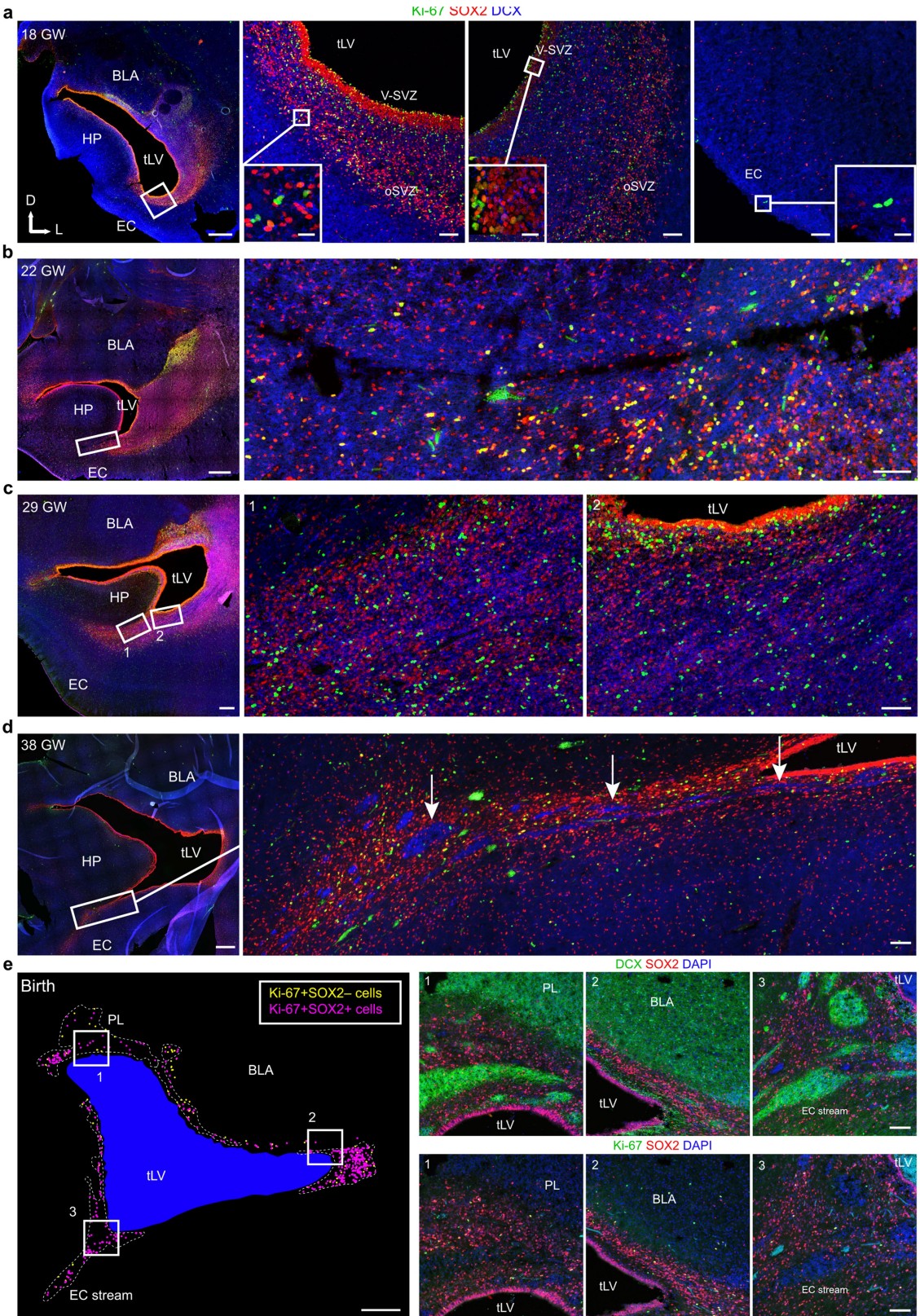

**Extended Data Fig. 6** | See next page for caption.

**Extended Data Fig. 6 | The EC stream forms between 22 and 27 GW.**
**a**–**d**, Anatomically matched coronal sections of the medial temporal lobe at the rostral tip of the uncus of the hippocampus across ages. At 18 GW (**a**), Ki-67⁺SOX2⁺ cells line the walls of the temporal lobe ventricle (same section level shown in (ii) in Extended Data Fig. 8c), and the medial wall facing the EC and hippocampus has fewer of these cells and no anatomical features separating the hippocampus and EC. At this age, there is a slight tissue protrusion of the hippocampus in the lateral direction towards the ventricle, a feature that becomes rapidly more prominent in the next weeks. At 22 GW (**b**), the hippocampus and surrounding tissue have all grown larger and a seam has begun to form between the hippocampus and EC. At this age, there is no heightened accumulation of Ki-67⁺ SOX2⁺ or DCX⁺ cells within this region. At 27 GW (**c**), this region has formed a higher concentration of Ki-67⁺SOX2⁺ cells as well as DCX⁺ cell clusters indicating that the EC stream has formed by 27 GW. At 38 GW (**d**), the density of both the Ki-67⁺ SOX2⁺ cells and the DCX⁺ cell clusters is increased, with many of both populations present within the EC stream extending from the medioventral tLV to the EC. **e**, Map of the location of Ki-67⁺SOX2⁻ (yellow) and Ki-67⁺SOX2⁺ (magenta) cells surrounding the tLV at 38 GW. Most Ki-67⁺ cells at this age are SOX2⁺, and are found in the paralaminar nucleus of the amygdala (PL) (1), the lateral V-SVZ (2), and in the EC stream (3). Scale bars: 1 mm (a–e all left panel overviews), 100 μm (**a**–**d** right panel overviews, **e** 1–3), 20 μm (**a** right insets). Abbreviations: BLA basolateral amygdala; EC: entorhinal cortex; HP: hippocampus; oSVZ: outer subventricular zone; PL: paralaminar nucleus; tLV: temporal lobe lateral ventricle; V-SVZ: ventricular–subventricular zone.

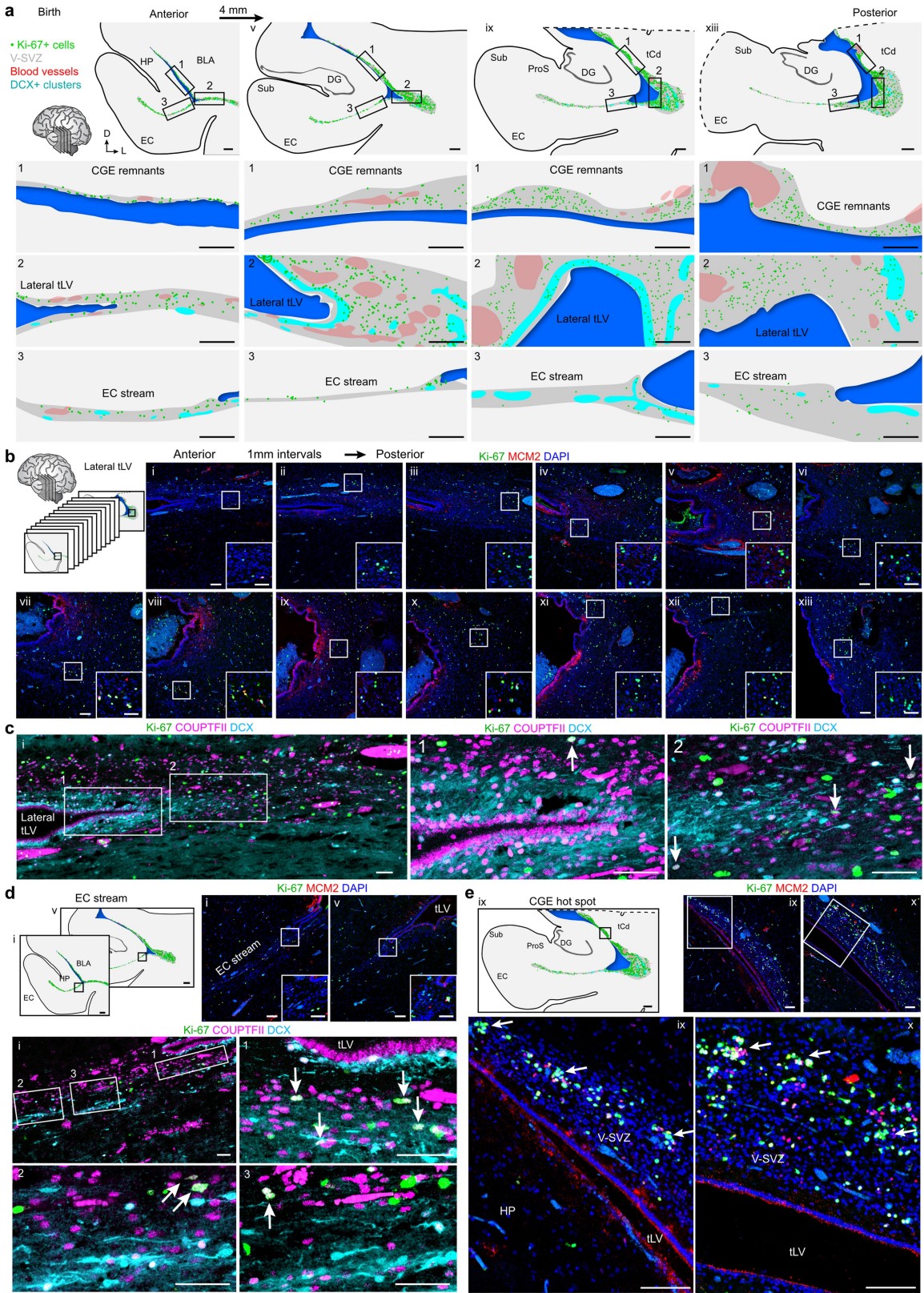

**Extended Data Fig. 7** | See next page for caption.

**Extended Data Fig. 7 | Clusters of dividing progenitors and immature neurons persist along the walls of the temporal lobe at birth. a**, Maps of coronal sections of the temporal lobe lateral ventricle (tLV) indicating Ki-67⁺ cells (green dots), the V-SVZ region (grey), blood vessels (BV, red), and DCX⁺ cell clusters (cyan). Sections are spaced by 4 mm along the rostral-caudal axis beginning at the anterior/ventral uncus of the hippocampus (i), extending across the rostral dentate gyrus (DG) (v), and more caudally across the DG (ix, xiii). The three inset regions are (1) remnants of the CGE along the dorsolateral wall of the tLV, (2), the most lateral extension of the tLV, and (3) the EC stream. **b**, Dividing Ki-67⁺ cells expressing MCM2 are present along the lateral extension of the tLV in the temporal lobe in the V-SVZ. Sections are spaced by 1 mm. **c**, A stream of DCX⁺ neurons wraps around the lateral extension of the tLV (1)

and is present near Ki-67⁺COUPTFII⁺ cells (arrows) (2) which are a subset of the Ki-67⁺ population in the lateral wall at birth. **d**, A subset of Ki-67⁺ cells in the EC stream express COUPTFII (arrows) shown at level (i). **e**, Sections stained for Ki-67 and MCM2 in the CGE-remnant region in sections adjacent to one in main Fig. 5e showing cell clusters within the V-SVZ (arrows). Scale bars: 1 mm (**a** top row, **d** maps, **e** map), 500 μm (**a** insets 1,2 and 3), 100 μm (**b** i–xiii overviews, **d** top right i and v overviews, **e** top right ix and x overviews and bottom insets), 50 μm (**b** i–xiii insets, **c**, **d i**, v top right insets and **i** insets 1–3). Abbreviations: BLA: basolateral amygdala; CGE: caudal ganglionic eminence; DG: dentate gyrus; EC: entorhinal cortex; HP: hippocampus; ProS: prosubiculum; Sub: subiculum; tCd: tail of the caudate nucleus; tLV: temporal lobe lateral ventricle; V-SVZ: ventricular–subventricular zone.

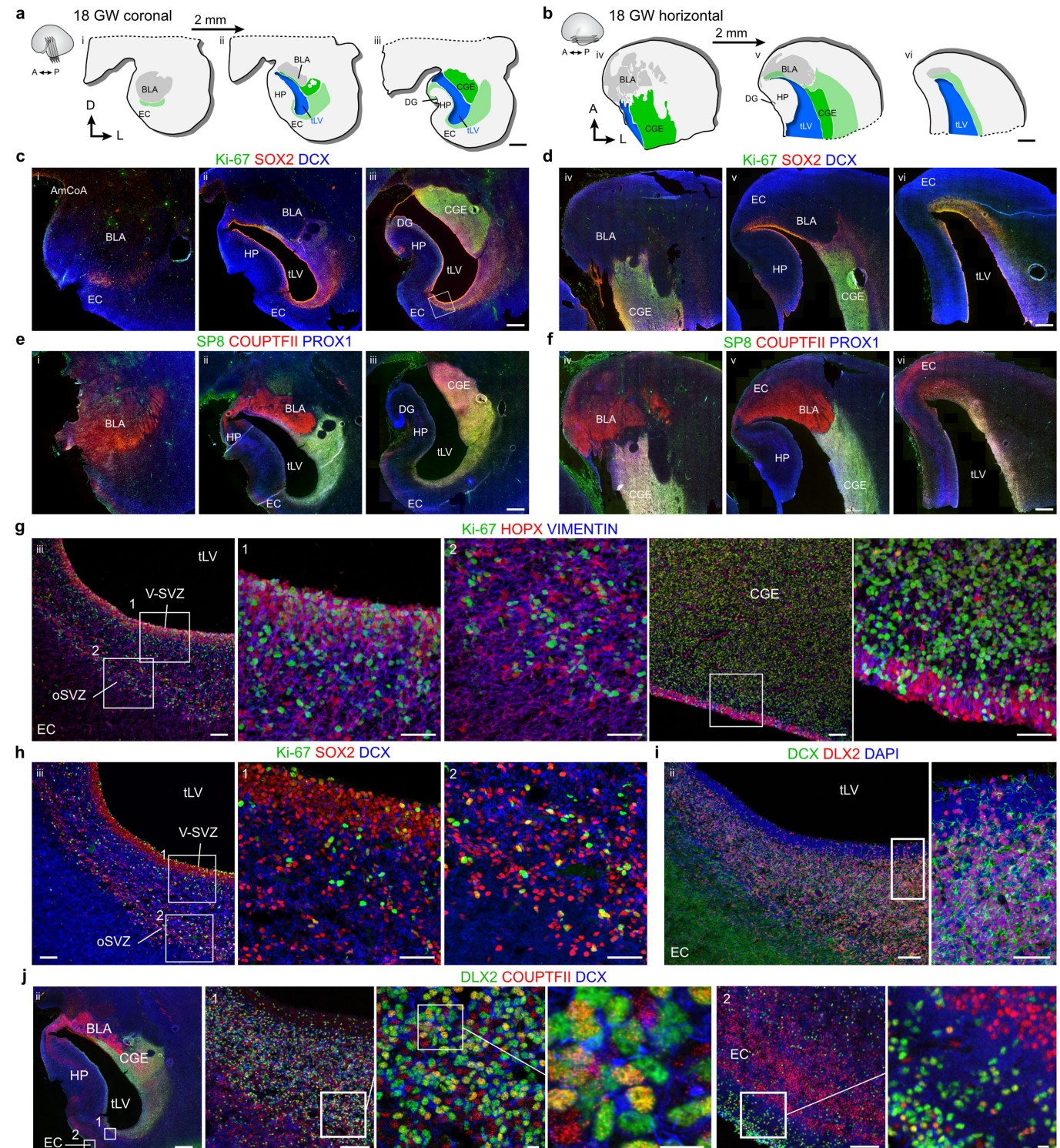

**Extended Data Fig. 8 | At 18 GW, the temporal lobe ventricle is surrounded by dividing progenitors in the CGE and DCX⁺DLX2⁺COUPTFII⁺ neurons extending toward the EC. a,b**, Diagrams of coronal (**a**) and horizontal (**b**) sections of the anterior temporal lobe at 18 GW showing the location of the CGE relative to the ventricle and EC. **c,d**, Immunostaining for Ki-67⁺SOX2⁺ progenitors and DCX⁺ young neurons in sections corresponding to those diagrammed in **a,b**. (Note: level ii corresponds to Extended Data Fig. 6a and level iii corresponds to Fig. 3a.) **e,f**, Immunostaining of sections immediately adjacent to those in **c,d** for SP8, COUPTFII and PROX1. **g**, On the ventral wall of the temporal lobe ventricle, many Ki-67⁺ cells expressing vimentin and HOPX are present in the V-SVZ and oSVZ. Similar cells are observed more dorsally along the lateral wall, in the CGE. **h,i**, The same region in **g** facing the EC contains

Ki-67⁺SOX2⁺ cells mixed with DCX⁺ neurons (**h**), the vast majority of which are DLX2⁺ (**i**). **j**, Immunostaining of a section adjacent to the middle section in the coronal series (**a**) reveals a mixed population of COUPTFII⁺ and COUPTFII⁻ cells expressing DLX2 along the ventricle facing the EC (1) and within the EC (2). Scale bars: 2 mm (**a**, **b**), 1 mm (**c–f**, **j** left panel), 100 µm (**g** left overview, right CGE overview, **h** left panel, **i** left panel, **j** 1 and 2 left panels), 50 µm (**g** 1, 2, CGE right panel, **h** 1, 2, **i** right panel), 10 µm (**j** 1, two right panels, 2 right panel). Abbreviations: BLA: basolateral amygdala; CGE: caudal ganglionic eminence; DG: dentate gyrus; EC: entorhinal cortex; HP: hippocampus; oSVZ: outer subventricular zone; tLV: temporal lobe lateral ventricle; V-SVZ: ventricular–subventricular zone.

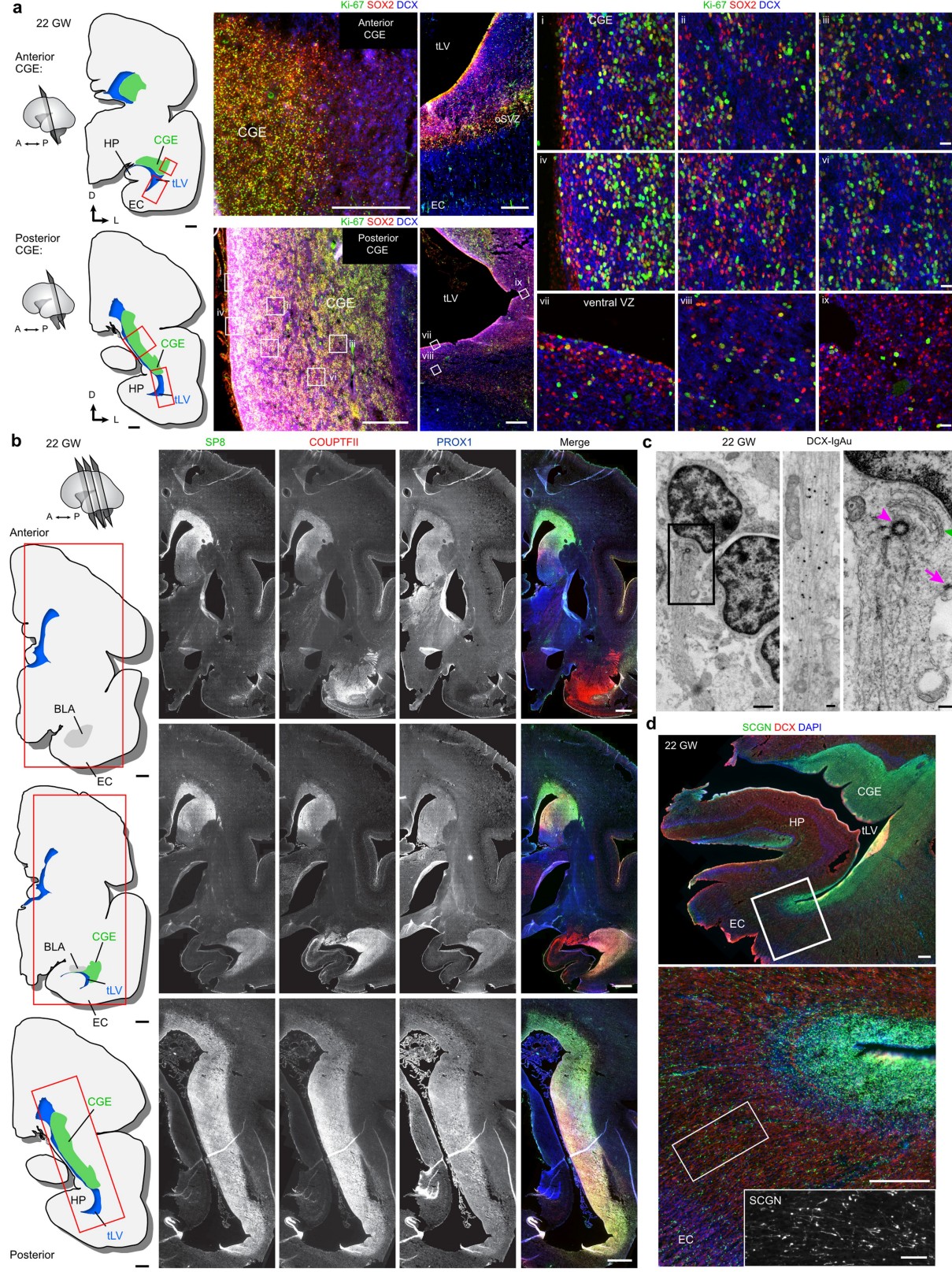

**Extended Data Fig. 9** | See next page for caption.

**Extended Data Fig. 9 | At 22 GW, the CGE contains Ki-67$^+$SOX2$^+$ progenitors and many SP8$^+$, COUPTFII$^+$ and PROX1$^+$ cells. a**, Diagrams of coronal sections of one hemisphere of the human brain at 22 GW at one anterior and one posterior level of the CGE. Red boxes indicate inset locations of immunostaining for Ki-67$^+$SOX2$^+$ cells. Immature DCX$^+$ neurons are present in clusters between the Ki-67$^+$SOX2$^+$ cells (i-vi). In the ventral V-SVZ, fewer Ki-67$^+$SOX2$^+$ cells are visible (vii-ix). **b**, Diagrams of coronal sections at three cross-sections across the temporal lobe and immunostains for SP8, COUPTFII, and PROX1 at each level showing that each marker is highly expressed throughout the CGE. **c**, Ultrastructure of a migratory young neuron at 22 GW in the temporal lobe CGE. This cell has a classical localization of the Golgi (green arrow) and centrosome (magenta arrowhead) in the leading process filled with microtubules and displays an adherens junction (magenta arrow). Immunogold labelling for DCX reveals processes filled with microtubules at this age. **d**, Immunostaining of the 22-GW human temporal lobe at the level of the anterior hippocampus. Insets show higher magnification of SCGN+ cells densely clustered near the ventral extension of the temporal lobe lateral ventricle and extending into the EC. Scale bars: 2 mm (**a** maps, **b** all panels), 500 μm (**a** left immunostaining overviews, **d**), 100 μm (**d** inset), 20 μm (**a** right panels i-ix), 1 μm (**c** top left), 200 nm (**c** top right, bottom panels). Abbreviations: BLA: basolateral amygdala; CGE: caudal ganglionic eminence; EC: entorhinal cortex; HP: hippocampus; oSVZ: outer subventricular zone; tLV temporal lobe lateral ventricle; V-SVZ: ventricular–subventricular zone.

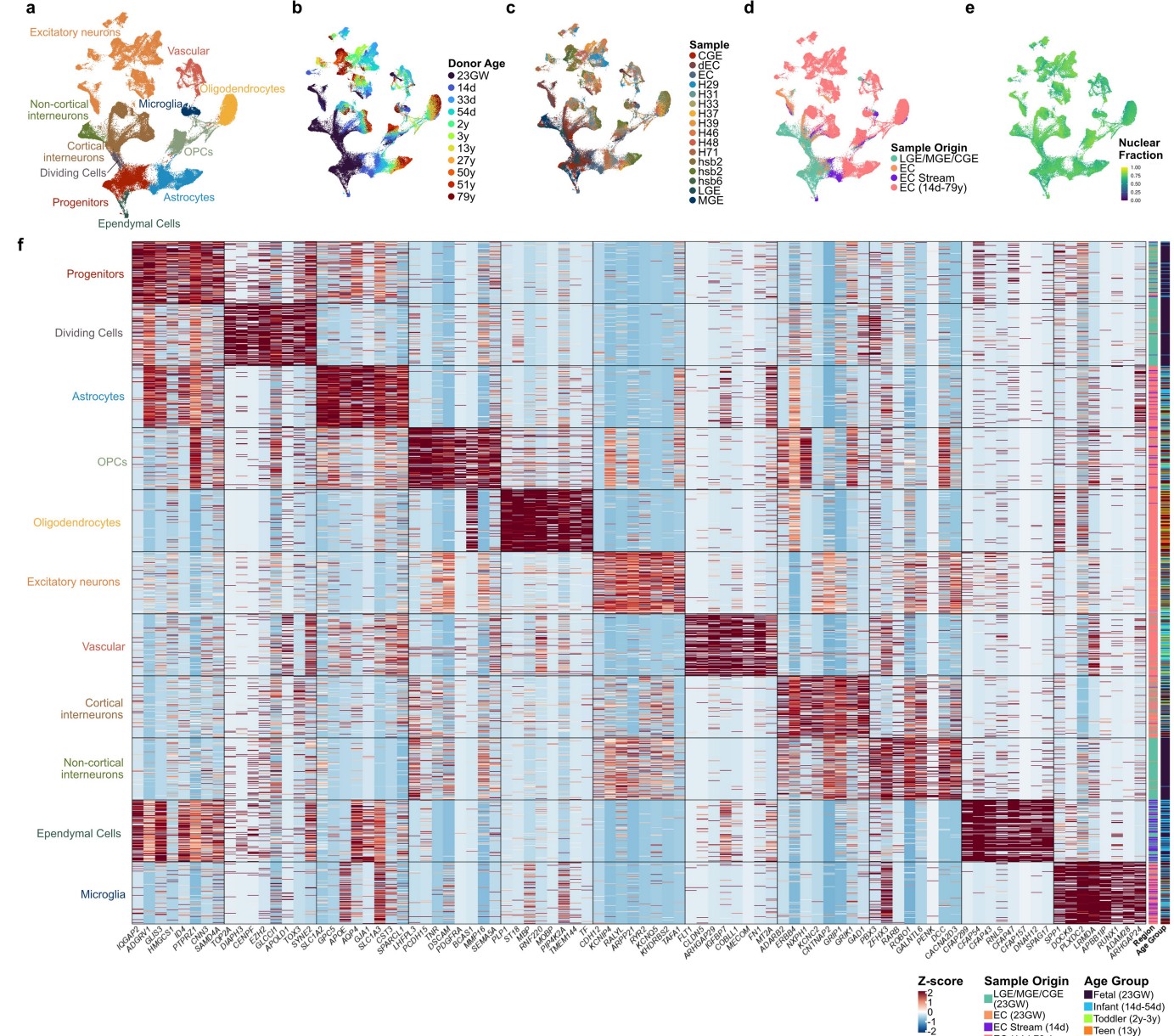

**Extended Data Fig. 10 | Composition of the entire snRNA-seq dataset.**
**a**, All cells from the merged dataset, coloured by the main cell types. **b**–**e**, UMAP plots showing (**b**) donor age, (**c**) individual samples, (**d**) regions of origin, and (**e**) nuclear fraction (fraction of reads containing intronic regions). **f**, Heat map of the top upregulated protein-coding DE genes for each cell type in **b**. Clusters are ordered by the results of hierarchical clustering.

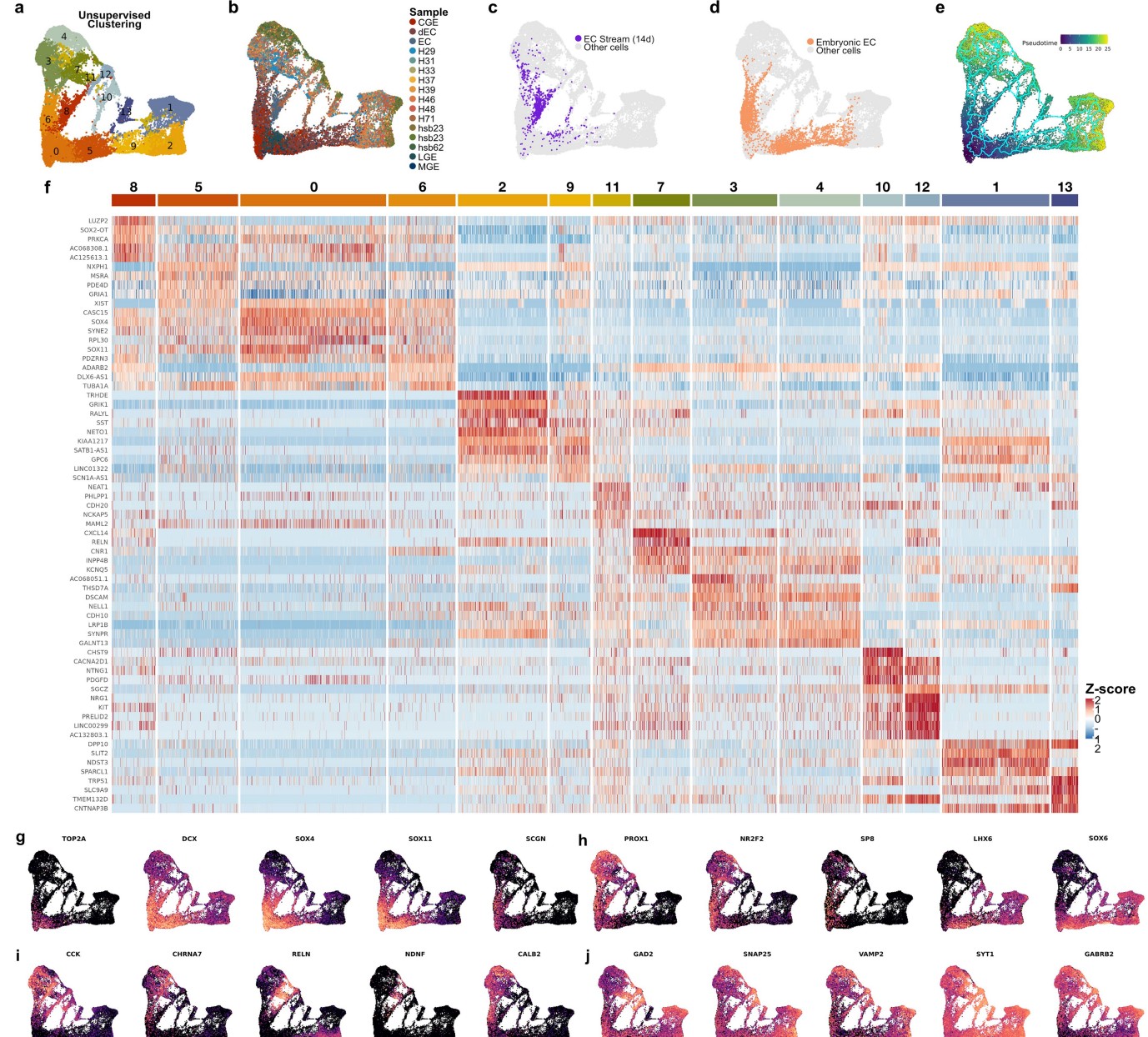

**Extended Data Fig. 11 | Composition of the snRNA-seq interneuron maturation dataset. a–e,** All cells from the interneuron maturation dataset. UMAP plots showing (**a**) results of unsupervised clustering, (**b**) their sample identifiers, (**c**) the distribution of cells from the EC stream, (**d**) distribution of cells from the embryonic EC at 23 GW, and (**e**) the calculated pseudotime. **f,** Heat map of top DE genes that are broadly expressed (>70% of cells) in each of the

clusters identified in (a). **g–j,** Feature plots highlighting expression of key genes. Expression of genes associated with (**g**) dividing cells (*TOP2A*) and immature neurons (*DCX*, *SOX4*, *SOX11*, and *SCGN*); (**h**) interneuron origins (*PROX1* and *NR2F2* highly co-expressed in CGE-derived interneurons, *LHX6* in MGE-derived neurons); (**i**) common CGE-derived interneuron subpopulations; and (**j**) mature neuron synaptic communication.

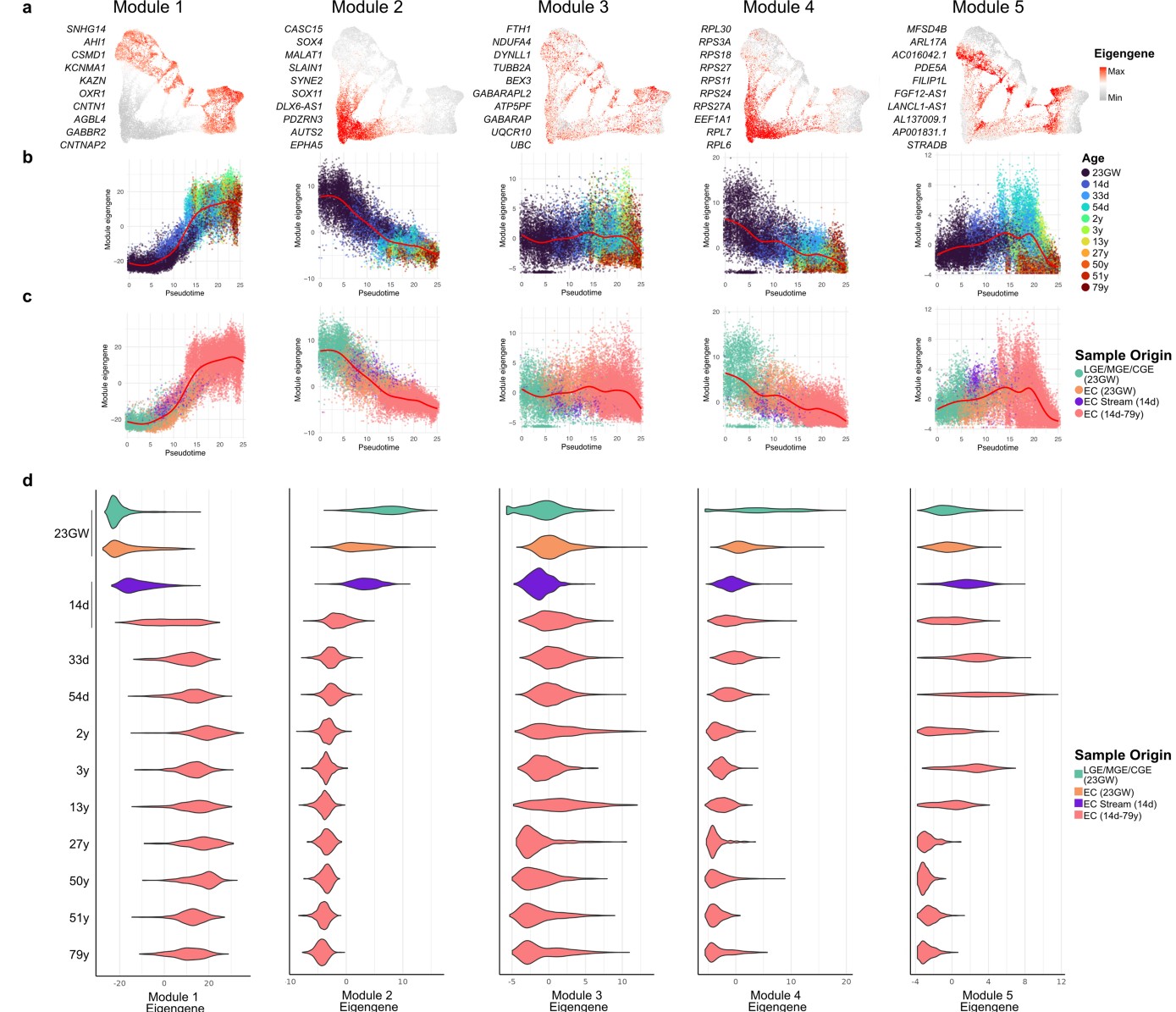

**Extended Data Fig. 12 | WGCNA analysis of inhibitory-neuron maturation.**
**a**, WGCNA revealed five modules of genes that are highly co-expressed. The most connected genes for each module are listed and their eigengene values, a proxy for the combined gene expression within each module is shown in UMAP plots. **b**,**c**, Eigengene for each module plotted against pseudotime, with individual cells coloured by (**b**) donor age and (**c**) region of origin. **d**, Violin plots showing the distribution of eigengene across different regions and ages.

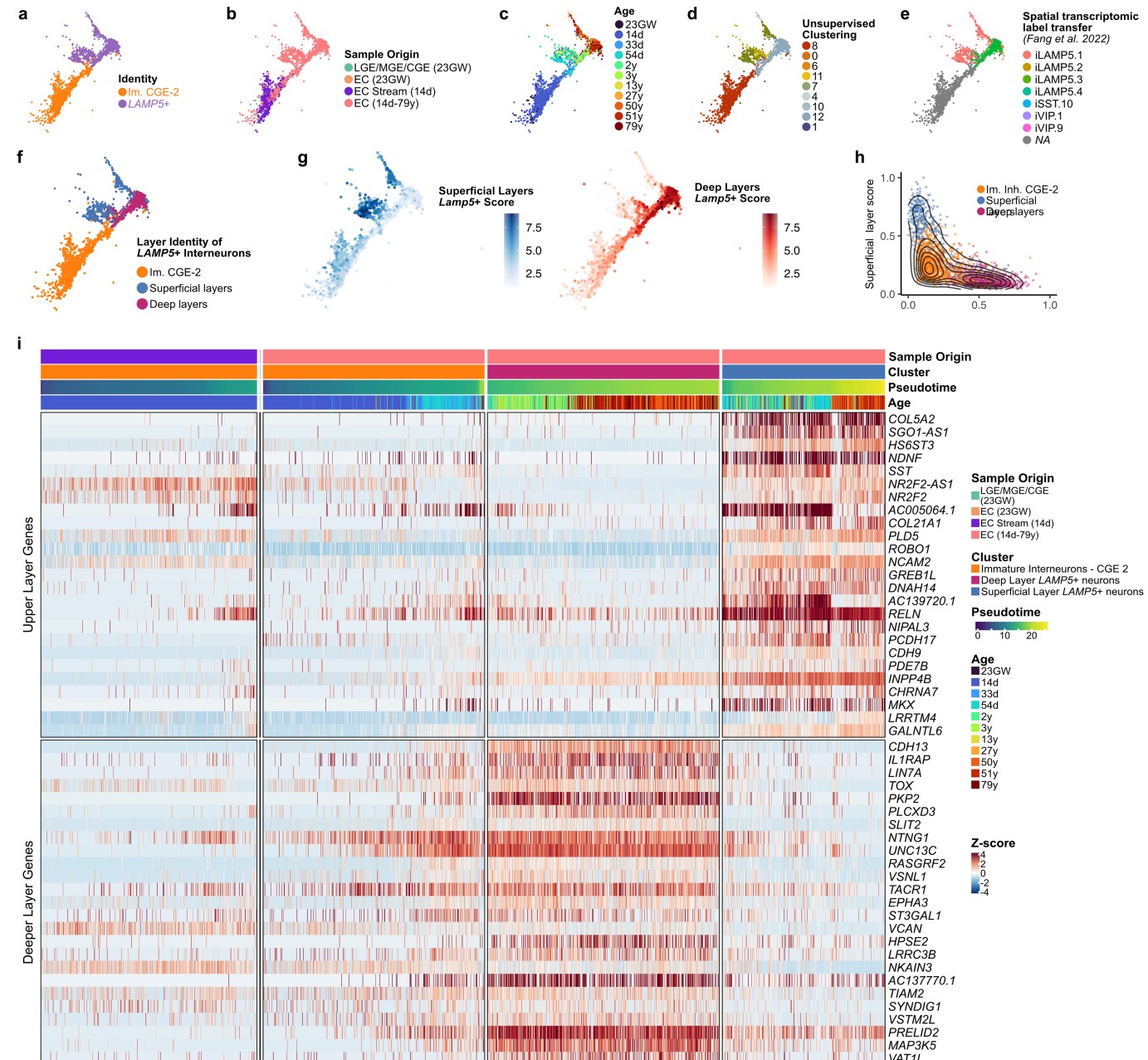

**Extended Data Fig. 13 | Spatial transcriptomic label transfer reveals superficial- and deep-layer *LAMP5*⁺ neurons. a–g,** A subset of the interneuron maturation dataset consisting of (**a**) only maturing CGE-derived LAMP5+ cells (Im. CGE-2 and *LAMP5*⁺). UMAP plots showing the (**b**) region of origin, (**c**) donor age, (**d**) cluster number, (**e**) spatial transcriptomic label transfer result and (**f**) inferred layer identities of *LAMP5*⁺ cells in our dataset. **g**, Module scores for superficial- and deep-layer *LAMP5*⁺ cells based on the top 10 DE genes between the two subpopulations. **h**, Module scores for each of the cells in **a–g**, colored by their inferred layer identities as in **f**. **i**, Heat map showing the top 25 most DE genes between superficial- (top panel) and deep-layer (bottom panel) cells. Cells were split by sample origin (first annotation row on top of heat map) and layer identity (second row). In each column, cells were ordered by pseudotime (third row) and donor age information is also shown (fourth row). Among immature *LAMP5*⁺ neurons derived from the EC stream (leftmost column), cells had a mixed expression of genes associated with superficial and deep layers, with some cells starting to express some markers such as *NCAM2*, *PLD5*, *NR2F2-AS1* and *ROBO2*.

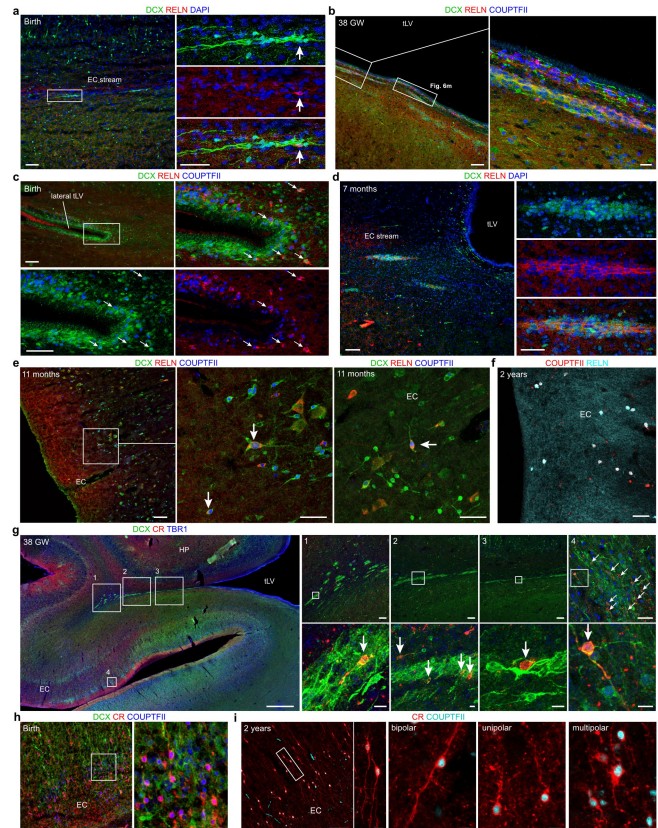

**Extended Data Fig. 14 | The EC stream supplies CR⁺ and RELN⁺ interneurons to the EC. a–c**, At birth, only a few individual DCX⁺RELN⁺ neurons (arrow) are present in the EC stream (**a**), whereas the ventral wall of the tLV contains multiple chains of DCX⁺RELN⁺COUPTFII⁺ neurons (**b**), location of inset shown in Fig. 6m. Some DCX+RELN+COUPTFII+ neurons are observed in the DCX⁺ chains wrapping around the lateral edge of the tLV (**c**). **d**, At 7 months of age, DCX⁺RELN⁺ cells are present in some dense clusters within the EC stream. **e**, Examples of DCX⁺RELN⁺COUPTFII⁺ cells in the EC at 11 months of age (arrows). **f**, Examples of of RELN⁺COUPTFII⁺ cells in the upper layers of the EC at 2 years of age. **g**, 38 GW coronal section with high magnification insets of DCX⁺CR⁺TBR1⁻ cells (arrows) in the EC stream clusters and EC (1–4). These cells are detectable in the EC stream at distal (1), middle (2), and proximal (3) levels relative to the medioventral tLV where the stream breaks away from the ventricle. Many additional DCX⁺CR⁺TBR1⁻ cells (arrows) are present in the EC (4) at birth, at the putative termini of the stream. **h**, At birth the EC contains DCX⁺CR⁺COUPTFII⁺ cells. **i**, At 2 years of age the vast majority of CR⁺ neurons in the EC co-express COUPTFII (inset). These cells display multiple morphologies including unipolar, bipolar, and multipolar. Scale bars: 1 mm (**g** left), 100 μm (**g** 1–3 top, **h** left, **i** left, **a** left, **b** left, **c** top left, **d** left, **e** left), 50 μm (**g** 4 top, **a** right insets, **b** right insets, **c** bottom and right insets, **d** right insets, **e** right, **f**), 10 μm (**g** 1–4 bottom, **h** right, **i** right). Abbreviations: EC: entorhinal cortex; HP: hippocampus; tLV temporal lobe lateral ventricle.

Arturo Alvarez-Buylla
Shawn F. Sorrells

# Reporting Summary

## Statistics

For all statistical analyses, confirm that the following items are present in the figure legend, table legend, main text, or Methods section.

| n/a | Confirmed | |
|---|---|---|
| ☒ | ☐ | The exact sample size (*n*) for each experimental group/condition, given as a discrete number and unit of measurement |
| ☐ | ☒ | A statement on whether measurements were taken from distinct samples or whether the same sample was measured repeatedly |
| ☐ | ☒ | The statistical test(s) used AND whether they are one- or two-sided<br>*Only common tests should be described solely by name; describe more complex techniques in the Methods section.* |
| ☒ | ☐ | A description of all covariates tested |
| ☒ | ☐ | A description of any assumptions or corrections, such as tests of normality and adjustment for multiple comparisons |
| ☐ | ☒ | A full description of the statistical parameters including central tendency (e.g. means) or other basic estimates (e.g. regression coefficient) AND variation (e.g. standard deviation) or associated estimates of uncertainty (e.g. confidence intervals) |
| ☐ | ☒ | For null hypothesis testing, the test statistic (e.g. *F*, *t*, *r*) with confidence intervals, effect sizes, degrees of freedom and *P* value noted<br>*Give P values as exact values whenever suitable.* |
| ☒ | ☐ | For Bayesian analysis, information on the choice of priors and Markov chain Monte Carlo settings |
| ☒ | ☐ | For hierarchical and complex designs, identification of the appropriate level for tests and full reporting of outcomes |
| ☒ | ☐ | Estimates of effect sizes (e.g. Cohen's *d*, Pearson's *r*), indicating how they were calculated |

*Our web collection on statistics for biologists contains articles on many of the points above.*

## Software and code

Policy information about availability of computer code

| Data collection | Leica LAS X and Neurolucida v2020.2.4 |
|---|---|
| Data analysis | CellRanger v7, Popscle-freemuxlet (v0.1), Seurat (v4.2), Monocle (v3), DoubletFinder (v2.0.3), DropletUtils (v1.18.1), Neurolucida (v2020.2.4), Imaris (v9.7.1), Graphpad Prism (v9), R (v4.1), ggplot2 (v3.3.5), sf package (v1.0-6), ImageJ (v1.54c), Adobe Photoshop (2023), edgeR(v.3.42) |

For manuscripts utilizing custom algorithms or software that are central to the research but not yet described in published literature, software must be made available to editors and reviewers. We strongly encourage code deposition in a community repository (e.g. GitHub). See the Nature Portfolio guidelines for submitting code & software for further information.

## Data

Policy information about availability of data

All manuscripts must include a data availability statement. This statement should provide the following information, where applicable:
- Accession codes, unique identifiers, or web links for publicly available datasets
- A description of any restrictions on data availability
- For clinical datasets or third party data, please ensure that the statement adheres to our policy

The raw data for the newly generated sn-RNAseq data, the aligned counts and metadata for each nuclei in the integrated dataset have been deposited in NCBI's Gene Expression Omnibus and are accessible through GEO Series accession number GSE199762. Raw data for additional adult human entorhinal cortex snRNA-seq can be found through GEO Series accession number GSE186538.

# Research involving human participants, their data, or biological material

Policy information about studies with human participants or human data. See also policy information about sex, gender (identity/presentation), and sexual orientation and race, ethnicity and racism.

| | |
|---|---|
| Reporting on sex and gender | The results in this study were based on tissue from donors of both sexes. Detailed information about the sex of individual tissue donors can be found in Supplementary Table 1. 65% (36/55) of donors were male and 35% (19/55) were female. No obvious sex-associated differences were found in our study. A more detailed comparison between sexes is particularly challenging as a much larger number of exact-age-matching donors of both sexes would be needed at multiple ages. |
| Reporting on race, ethnicity, or other socially relevant groupings | Our study does not group our findings based on race, ethnicity or any other socially relevant categorization. |
| Population characteristics | Detailed information on sex, age, and clinical history of all samples analyzed can be found in Supplementary Table 1. In short, 27% (15/55) of all samples come from gestational ages, 22% (12/55) from 38 wpc to 42 wpc, 24% (13/55) from 42 wpc to 1 year of age and 27% of samples (15/55) from donors older than 1 year of age. |
| Recruitment | Participants in this study were research consented postmortem autopsy or epilepsy patients.  No active recruitment was implemented.  Postmortem human brain samples with no known neuropathological condition were included and were collected from UCSF, UPMC, and the NIH Neurobiobank, all banks in the United States, and the University of Valencia in Spain (see Supplementary Table 1 for details). For epilepsy cases, all tissue was from intractable seizure cases. This is described in the Materials and Methods section, under "Human Tissue Collection" as well as Supplementary Table 1.  There is no self-selection bias in these inclusion criteria, however we cannot rule out possible bias from the patient populations and countries sampled or unidentified disease conditions. |
| Ethics oversight | Human Gamete, Embryo and Stem Cell Research Committee (Institutional Review Board) at UCSF, The Ethical Committee for Biomedical Investigation, Hospital la Fe (2015/0447) and the University of Valencia Ethical Commission for Human Investigation, The Committee for Research on Decedents (CORID) at the University of Pittsburgh. Specimens collected at UPMC had IRB approved research informed consents along with HIPAA authorizations signed by parents or responsible guardians |

Note that full information on the approval of the study protocol must also be provided in the manuscript.

# Field-specific reporting

Please select the one below that is the best fit for your research. If you are not sure, read the appropriate sections before making your selection.

☒ Life sciences          ☐ Behavioural & social sciences          ☐ Ecological, evolutionary & environmental sciences

For a reference copy of the document with all sections, see nature.com/documents/nr-reporting-summary-flat.pdf

# Life sciences study design

All studies must disclose on these points even when the disclosure is negative.

| | |
|---|---|
| Sample size | We did not use statistical methods to determine sample size a priori. Our work describes a migration process that was consistently observed at young ages and not in adults across 55 total samples.  For the ages and species evaluated this sample size is equivalent to or exceeds sample sizes used in comparable investigations. |
| Data exclusions | In our single-nuclei RNAseq analyses, barcodes that were assigned as doublets by Freemuxlet our DoubletFinder or nuclei that did not meet our QC parameters (min. UMI count, min. gene count, max. percentage of reads belonging to mitchondrial genes, min. nuclear fraction or nuclei clustered with them) |
| Replication | Each individual antibody staining was replicated at least 3 times in different experiments across a minimum of 3 sections in each sample. Single-nuclei RNAseq data consisted of two runs: 1) a multiplexed snRNA-seq run where samples of the same cortical region (EC) from 5 different donors (ages ranging 33d-2y) were pooled together and ran in duplicate (in two 2 different wells of a 10x chip), allowing us to confirm that batch effects between wells were minimal. 2) A non-multiplexed snRNA-seq run that did not contain technical replicates for each biological sample, but consisted of additional samples of EC comparable to the ones in the first run and additional samples from LGE, MGE, CGE, and EC stream. These additional samples did not have replicates due the limited availability of samples containing the microdissected regions. |
| Randomization | Randomization is not relevant since there are no experimental and control groups in this study and all samples were categorized according to their biological age. |
| Blinding | Samples were not allocated to different groups, so investigators did not have to be blind to group allocations. It is not possible to completely blind investigators to age or region due to obvious cellular differences between regions sampled, the decrease in cell density and large increases in brain size during development. Samples were not quantified in order of age, but were evaluated in a random order. |

# Reporting for specific materials, systems and methods

We require information from authors about some types of materials, experimental systems and methods used in many studies. Here, indicate whether each material, system or method listed is relevant to your study. If you are not sure if a list item applies to your research, read the appropriate section before selecting a response.

## Materials & experimental systems

| n/a | Involved in the study |
|---|---|
| ☐ | ☒ Antibodies |
| ☒ | ☐ Eukaryotic cell lines |
| ☒ | ☐ Palaeontology and archaeology |
| ☐ | ☒ Animals and other organisms |
| ☒ | ☐ Clinical data |
| ☒ | ☐ Dual use research of concern |
| ☒ | ☐ Plants |

## Methods

| n/a | Involved in the study |
|---|---|
| ☒ | ☐ ChIP-seq |
| ☒ | ☐ Flow cytometry |
| ☒ | ☐ MRI-based neuroimaging |

## Antibodies

| | |
|---|---|
| Antibodies used | Detailed information on antibodies used, their dilutions, cat. and lot numbers can be found in Supplementary Table 3 |
| Validation | Primary antibodies were selected based on prior validation in human tissue and vendor reported information about antigen target specificity and reactivity in human tissue (See Supplementary Table 3 for manufacturer descriptions about specificity for each antibody). For all antibodies we performed titration experiments (including controls with secondary antibody only) and compared staining patterns to previously published information about developmental or regional expression patterns and intracellular localization of targets. |

## Animals and other research organisms

Policy information about studies involving animals; ARRIVE guidelines recommended for reporting animal research, and Sex and Gender in Research

| | |
|---|---|
| Laboratory animals | Rhesus macaque (Macaca mulatta), ages ranging from newborn to 1.5 years of age (Supplementary Table 2) |
| Wild animals | This study did not involve wild animals |
| Reporting on sex | Information about sex of animals in this study can be found in Supplementary Table 2. Sex-based analyses were not performed as the main variable in the study was age/developmental stage. |
| Field-collected samples | This study did not involve samples collected from the field. |
| Ethics oversight | All experiments were conducted in accordance with Fudan University Shanghai Medical College and University of Pittsburgh guidelines |

Note that full information on the approval of the study protocol must also be provided in the manuscript.

