## [Peer Review File · Nature]

Manuscript Title: Protracted Neuronal Recruitment in the Temporal Lobe of Young Children

Reviewer Comments & Author Rebuttals

Reviewer Reports on the Initial Version:

Referees' comments:

Referee #1 (Remarks to the Author):

The authors provide compelling data using immunofluorescence and snRNA-seq in a relatively broad set of 40 postnatal human samples to characterize a novel population of immature cells in the entorhinal cortex. These cells comprise a migratory stream of young interneurons or subtype-restricted neuroblasts that emerges during prenatal neurodevelopment and persists robustly through 2-3 years of life. The authors also find that during infancy, the lineage of the entorhinal cortex migratory stream cells shifts from a mixture of MGE- & CGE-derived interneurons to near-exclusively CGE-derived interneurons of diverse subtypes marked by CR, RELN and VIP. This process is potentially relevant for cellular and functional plasticity of the entorhinal cortex as it takes on numerous complex cognitive and behavioral tasks during life. Overall, this paper presents strong evidence, appears conceptually and methodologically sound, and the manuscript text is fluid and well-organized.

Overall, the authors are especially commended for the extensive, high-quality staining of human tissue, as this includes many technical challenges beyond that for other experimental systems. Especially within the supplemental figures, the level of detail that can be surmised from the experiments in this study could be a great addition to the field. As noted throughout the comments below, I hope that certain additions, re-combinations and deeper analysis of this data could bring this project to its full potential.

The authors are commended for taking up this interesting and important question, for constructing a logical yet innovative experimental paradigm using a notably substantial set of samples, and for the technical expertise demonstrated in their immunofluorescence studies. However, there are also a few questions that arise, especially with respect to the depth of cell lineage analysis and the organization of the text, main figures, and other findings relegated to the expanded data images, and which the authors will need to address. My major and minor concerns are listed sequentially below.

First, in very broad terms, and before addressing specific points below, I might prompt the authors to consider the overarching layout of this study, with reference to Figure 1 and all subsequent sections. To optimize the linear progression of experiments and experimental rationale, the authors may consider the features of Figure 1 that are discrete, unique, and a crucial introduction to the project, versus those that convey a gestalt for the overall project and that are subsequently retold in better detail by Figures 2-4.

Major concerns:

1. It would be useful for the authors to provide more context on the DCX and PSA-NCAM labels that were used, especially since cell populations expressing these proteins often are inferred to have important and unique properties beyond being merely "migratory neurons." Here, and throughout much of the manuscript, this may be part of a strategy to avoid courting controversy that too often overshadows this field. However, if the authors are setting out to propose that this population of postnatal migratory cells that originates from a major neurogenerative zone is truly

interesting and potentially impactful, then they will have to “address the elephant in the room” ...since without that elephant, the room is much less exciting. At minimum, the authors should spell out the acronyms, state what they commonly label, and explain what typically is signified by double-positive cells.

2. Following from the point above, in addition to the doubly labeled cell populations, it would be interesting for the authors to mention the numbers/proportions of cells labeled DCX+ or PSA-NCAM+ individually. Furthermore, although this is referenced later in this section and in Suppl figs 2, null4, the initial characterization of this migratory stream should include some additional characterization of the “age” or “birthdating” of cells in this stream. Practically, since Figure 1 already contains a large number of panels, this could be done as supplemental figure. Thematically, this is important also related to my comments immediately above and below.

3. In the description of the EC postnatal migratory stream, I feel that the authors shouldn't avoid adding at least a brief reference to, and comparison with, other developmental and postnatal ... the other major proliferative and migratory zones that have been a crucial part of the literature leading to this study. My strong feeling is that by not mentioning the SVZ-RMS and SGZ-GCL makes the implicit need for comparison more conspicuous, and that this is a major missed opportunity to begin to address head-on the most interesting features of this EC migratory stream. Specifically, by not even providing some reference to known features of RMS neuroblasts or transit amplifying progenitor cells—that are known not to be abundant in humans—the characterization of the EC migratory stream in this paper lacks context and appears more obscure.

4. Within the last portion of the description of findings for Fig 1, from lines 86-102, more attention is paid to the subclassification of cell populations in the region of the EC migratory stream and known features of the caudal ganglionic eminence lineage. However, a more precise and comprehensive characterization still could be applied using a number of potential techniques, and the potential relevance of these cells could be foreshadowed. This could involve more extensive longitudinal characterization of cell fate commitment and cell subtype identity markers within the EC interneuron lineage (possibly compared with that known of other adjacent niches and lineages). The authors already have performed an extensive screen of cells subtypes using Ki67, SOX2, Vimentin, MCM2, SP8, DLX2, DCX, PSA-NCAM, COUPTFII and PROX1 markers in various combinations in Fig 1 and extended Figs 1-4. However, the quantification, analysis and meaning of this does not come through to its full potential in the text or the main figures (e.g., lines 94-100 and elsewhere). This issue also leads into comments noted below concerning evidence that mature EC interneurons actually include cells derived from the postnatal/juvenile EC migratory stream...or that immature postnatal EC stream interneurons attain functional importance. Indeed, more information or analysis that could support the functional relevance of such cells could strengthen this manuscript substantially.

5. The description of the establishment of the anlage that will become the EC migratory stream in lines 114-127 and 142-147 is particularly fascinating. This process sounds similar to the temporally and anatomically adjacent establishment of the hippocampal DG/SGZ anlage as well. It would be especially interesting if the authors were able to mention this similar phenomenon and then compare/contrast this new neurodevelopmental process. Otherwise, some readers may wonder if this EC stream is, indeed, distinct from dentate gyrus/hippocampal development occurring nearby through relatively similar processes. Additionally, could some of the basic features also be described with a corroborating histological technique? Immunohistochemistry vs immunofluorescence, or even in situ hybridization?

6. Following the first major point of this study, that the EC (immature neuron) migratory stream exists, the second major point of this study is that many of its components are depleted over time. However, when the latter findings are based upon human tissue, it is important to exclude disease-related processes potentially related to the early demise of young donors that also could have a major confounding influence on such cell populations. Indeed, cancer, seizures, chronic

inflammatory state, neurodevelopmental disorders are all profoundly overrepresented in patients ultimately contributing to brain banks, and these pathological states have a major impact on the health and persistence of generative regions and processes in the brain. At minimum this should be addressed in the discussion, and ideally could be conveyed in parallel with the results to provide clarity to the reader and solidity to the findings at the time. If such assurances cannot reasonably be made based upon known features of donors—must also be clarified and expanded within the Methods section—then the authors should strongly consider performing a parallel experiment from primate tissue for confirmation. It stands to reason that, if these findings are real and important, they will be reproduced in closely related species and will be profoundly fortified by such corroboration.

7. Following from earlier comments concerning more detailed fate mapping and characterization of EC migratory stream DCX+PSA-NCAM+ double positive cells, I am curious if more detailed subclassification can be done at later ages and for the differentially migrating radial vs tangential patterns described in lines 160-175. Although aspects of this question are addressed in the following two sections, I think it would be interesting to note here whether radial vs tangential orientations had different patterns of DCX+ or PSA-NCAM+ single vs double labeling, and whether co-expression of Ki67 or PROX1, etc. also could be assessed. Establishing age-related migration patterns of more/less mature cells within and from the EC migratory stream could be particularly interesting...especially as this might relate to their functional or therapeutic potential. Additionally, within this section of the following section detailing the late-migrating interneurons, could high-resolution microscopy be included to define some structural-functional features of these cells? Does one group or the other have different patterns of +/- en passant synapse formation or other functional interaction with neighboring cells? Is there evidence of neurotransmitter synthesis yet? To elaborate upon immunostaining, could the authors apply snRNA-seq to profile the components of the EC stream in greater detail? And could this lead to a more expansive characterization of the EC migratory stream interneuron lineage relative to age and cell characteristics as also portrayed in this section?

8. At risk of asking the authors to perform major additional histological analyses, I wonder if the description of EC migratory stream neuroblast fate commitment toward GABAergic interneuron identity could be expanded to provide finer analysis and deeper context using staining that already might have been performed. Although DCX+DLX2+ cells were noted, could PSA-NCAM+DLX+ double positive cells also be assessed? Would this help draw parallels or distinctions with transit amplifying cells and interneuron progeny in the RMS? When comparing the alternate fate commitment route suggested by DCX+TBR1+ double positive cells, was staining with PROX1 (already reported in other contexts) also considered? For instance, would DCX+PROX1+ double positive cells indicate a different subset from the DCX+DLX2+ (or PSA-NCAM+DLX2+) double positive cells that are the population at the center of this finding? If this broader and more detailed context can be supplied from the analysis of staining that might already have been performed, I think this could allude more clearly to a biological meaning of this proposed cellular/developmental process.

9. Findings concerning EC migratory stream DCX+ cell features with respect to MGE vs CGE lineages are potentially quite compelling. However, further validation, contextualization and analysis could be helpful. Firstly, in this regard, I wonder if much of the depth of analysis that could be gleaned from experiments portrayed in Suppl Fig 7 are sufficiently explained in the text or are conveyed via the related panels of primary Fig 3. In this respect, this topic could be very nicely corroborated and expanded using snRNA-seq methods as was reported in the following Results section (outlined in Fig 4). However, it is not clear that this also was intended to lend validation to the breadth of immunostaining findings provided in Suppl Fig 7, as this still would be helpful.

10. As noted in the comment above, addition of snRNA-seq experiments can provide great benefit to this study overall. Experimental samples in this section appear to match the postnatal age

ranges covered in experiments above, aside from the brief reference to the population of DCX+TBR1+ cells that could be measured at age 6 and 13 years. Later in this section, and with regard to more diverse lineages of interneuron subtypes, it's interesting to see the increased expression of multiple ATP-synthesis genes. However, since some of these genes were encoded by mitochondria DNA, is it possible that the differences are caused by quality differences between the two datasets, i.e., one dataset has higher mitochondria reads? The authors also might check other mitochondria genes and evaluate whether this increase is specific to ATP-synthesis genes alone, or broadly for all mitochondria genes. Similarly, the authors also could check if all vs a subset of ribosomal genes are differentially expressed.

Finally, if possible, the authors should include an additional/alternative analysis (e.g., RNA velocity) to confirm the lineage progression in Figure 4. Currently it seems that the real ages have a lower correlation with pseudotime in certain lineages (e.g., lineages 7,8 and 9).

11. The authors should revise the manuscript's title, since "persistent postnatal" may imply a fully realized, lifelong process; yet experiments in this paper span the young juvenile period that may be thematically more "developmental" than "adult." Likewise, the authors may consider providing a more nuanced moniker for the cell populations described in this study, since the word "interneurons" alone does not necessarily convey their status as putative immature, recently born, pre-interneurons or even (subtype-restricted?) neuroblasts.

Minor concerns:

The manuscript file provided for review was not formatted with an abstract or with the designation of a Background or Introduction section before beginning with the first Results section of "Extensive migration in..."

In the introduction or within the first results sections, the authors should consider including a brief reference to the source of the tissue analyzed and characteristics of donors. Even something as simple as "we stained human temporal lobe sections from non-diseased brain bank donors at birth for doublecortin (DCX) and..." in lines 56-57 could provide better context concerning the important features of this study.

In line 192, the authors introduce a new transcription factor by first spelling out "T brain 1" followed by the parenthetical designation of its abbreviation "(TBR1)". I would encourage the authors to use the same format for all other labels or markers reported throughout the manuscript. Common usage may also advise including a brief definition or explanation of function, as is also included here, with "the cortical excitatory neuron transcription factor." Please note the contrast with DCX and PSA-NCAM in line 57, and other examples throughout the manuscript.

Line 231-232: The authors said these cells consist of a mixture of MGE and CGE-derived interneurons, yet these cells were labeled as MGE-derived in Fig. 4d. This seems inconsistent or confusing. These cells are the LAMP5+/LHX6+/NKX2-1+ cells and previous studies suggested MGE progenitors can give rise to these cells (Tasic et al., 2018). This sentence needs to be updated. Finally, clusters 3,4,5,21 also are positive for LHX6 and NR2F2, yet these cells appear to be ignored.

Line 263-265: It's challenging for the reader to get this claim from Fig. 4h. For clarity, the authors should better explain the plot or provide additional analysis.

Line 336-338: comments about RELN+ cells and Alzheimer Disease pathology in this context seems like a forced association.

Line 531-532: should be "UMI counts" rather than "reads".

Referee #2 (Remarks to the Author):

The manuscript by Nascimento and authors continues the remarkable expansion of inhibitory cell types that are made postnatally in humans. Here by exploring the entorhinal cortex the authors have discovered a late maturing CGE-derived population that appears to produce a range of CGE derived cells that may give rise to a remarkable diversity of CGE interneurons in the entorhinal cortex. Like many of the studies that preceded this, it uses a combination of RNA and antibody stains to longitudinally characterize and detail the types and ages that these populations become integrated, in this case in the entorhinal cortex. Fascinating as this detailing of cell types is, the analysis seems rather limited both in their analysis of the origins of these cells and the types they give rise to compared to other from the Alvarez-Buylla group, in particular the recent publication from Pollen et al. The methods used are careful and well illustrated but descriptive and given no direct inkling of the function of these populations beyond describing their existence. In sum, while a careful and interesting addition to the existing literature, it does not reach a level that would make it appropriate for publication in Nature. Among the things that are too superficial to argue that this a sufficient advance are the lack of insight this work provides with regard to the origin, diversity or potential significance of these observations beyond what has been shown extensively by both the Kriegstein and Alvarez-Buylla laboratories. What is described as lineages amount to gene expression trajectories which neither identify that they arise from a distinct progenitor pool nor delineate how the 9 lineage type relate to functional classes. Given the reliance on markers but lack of functional analysis the best the authors can conclude is that further neurogenesis akin to that described by both Alvarez-Buylla, Nowakowski, Kriegstein and Pollen can now be extended to the entorhinal cortex. While this is a solid descriptive study it fall far short of an advance that would be suitable for publication in Nature.

Author Rebuttals to Initial Comments:

Referee #1 (Remarks to the Author):

The authors provide compelling data using immunofluorescence and snRNA-seq in a relatively broad set of 40 postnatal human samples to characterize a novel population of immature cells in the entorhinal cortex. These cells comprise a migratory stream of young interneurons or subtype-restricted neuroblasts that emerges during prenatal neurodevelopment and persists robustly through 2-3 years of life. The authors also find that during infancy, the lineage of the entorhinal cortex migratory stream cells shifts from a mixture of MGE- & CGE-derived interneurons to near-exclusively CGE-derived interneurons of diverse subtypes marked by CR, RELN and VIP. This process is potentially relevant for cellular and functional plasticity of the entorhinal cortex as it takes on numerous complex cognitive and behavioral tasks during life. Overall, this paper presents strong evidence, appears conceptually and methodologically sound, and the manuscript text is fluid and well-organized.

Overall, the authors are especially commended for the extensive, high-quality staining of human tissue, as this includes many technical challenges beyond that for other experimental systems. Especially within the supplemental figures, the level of detail that can be surmised from the experiments in this study could be a great addition to the field. As noted throughout the comments below, I hope that certain additions, re-combinations and deeper analysis of this data could bring this project to its full potential.

The authors are commended for taking up this interesting and important question, for constructing a logical yet innovative experimental paradigm using a notably substantial set of samples, and for the technical expertise demonstrated in their immunofluorescence studies. However, there are also a few questions that arise, especially with respect to the depth of cell lineage analysis and the organization of the text, main figures, and other findings relegated to the expanded data images, and which the authors will need to address. My major and minor concerns are listed sequentially below.

First, in very broad terms, and before addressing specific points below, I might prompt the authors to consider the overarching layout of this study, with reference to Figure 1 and all subsequent sections. To optimize the linear progression of experiments and experimental rationale, the authors may consider the features of Figure 1 that are discrete, unique, and a crucial introduction to the project, versus those that convey a gestalt for the overall project and that are subsequently retold in better detail by Figures 2-4.

We have reorganized the manuscript, taking into consideration this suggestion: We have combined the previous data in Fig. 1 which is crucial for the introduction to the project and moved other data to support later sections. As a result, each figure is smaller and focuses on one or two themes about each aspect of the investigation, as summarized below:

Fig 1- EC-stream location at birth.

Fig 2- Age series and duration of the EC stream in humans and new data from the macaque.

Fig 3- Embryonic ventricular collapse forming the scaffold for migration.

Fig 4- The EC stream supplies migratory interneurons to the EC including new snRNA-seq of the EC stream.

Fig 5 - Migratory neurons in the EC stream are generated in the CGE with proliferation quantifications.

Fig 6 - Identity of EC stream migratory interneurons and their final position in the EC with a large new sn-RNA seq dataset and analysis of microdissected samples from mid-gestation, infancy, and adult.

Major concerns:

1. It would be useful for the authors to provide more context on the DCX and PSA-NCAM labels that were used, especially since cell populations expressing these proteins often are inferred to have important and unique properties beyond being merely “migratory neurons.” Here, and throughout much of the manuscript, this may be part of a strategy to avoid courting controversy that too often overshadows this field. However, if the authors are setting out to propose that this population of postnatal migratory cells that originates from a major neurogenerative zone is truly interesting and potentially impactful, then they will have to “address the elephant in the room” ...since without that elephant, the room is much less exciting. At minimum, the authors should spell out the acronyms, state what they commonly label, and explain what typically is signified by double-positive cells.

Response:

1. We agree with the reviewer that marker expression alone can be controversial. In our revised manuscript we now define acronyms and clarify our interpretations of marker expression. This is an important point raised by the reviewer because DCX can label non-neuronal cells and non-immature neurons and PSA-NCAM can label non-immature neurons. Together, DCX & PSA-NCAM restrict labeled cells to young neurons much more extensively than either alone (see also new **Extended Data Fig. 5i,j** and response to points 2,7, and 8). We also provide new histological and snRNA-seq data that provide additional context for these labels and allow us to not rely on only a few specific markers. For our interpretation of the migratory properties of these neurons, we provide cellular morphology, ultrastructure, and anatomical orientations of individual neurons. These criteria together with our new transcriptomic analysis of the microdissected stream, together with their transitory nature, support that these are migratory young interneurons. The main observation of our study is the continuing incorporation of such a large population of young migratory neurons into entorhinal cortex circuits up to at least two years of age and could extend plasticity.

This comment from the reviewer may also refer to our observation of some dividing RG located within the stream itself. Our new sn-RNAseq data from the EC stream (see new **Fig. 5d**) identify the cells produced by these RG (now **Fig. 5b,c, Extended Data Fig. 6,7**). Interestingly, our data confirm that these progenitor cells are still active postnatally, and indicate that they are mainly generating cells committed to an oligodendrocytic fate, with a smaller fraction of excitatory neurons (**Fig. 5d**). There was no evidence that these late progenitor cells in the EC stream generate the large population of migratory interneurons (**Figs. 4, 5**). Based on the transcriptional profile of the migratory EC stream neurons which indicates a CGE origin, we investigated the proliferation in the CGE from mid-gestation to birth. We found high levels of proliferation in the CGE into the third trimester, and still some dividing progenitors at birth (**Fig. 5e-h**). Taken together our data indicate that these migratory neurons are born in the CGE, and not from the RG we observed in the stream.

2. Following from the point above, in addition to the doubly labeled cell populations, it would be interesting for the authors to mention the numbers/proportions of cells labeled DCX+ or PSA-NCAM+ individually. Furthermore, although this is referenced later in this section and in Suppl figs 2,3,4, the initial characterization of this migratory stream should include some additional characterization of the “age” or “birthdating” of cells in this stream. Practically, since Figure 1 already contains a large number of panels, this could be done as supplemental figure. Thematically, this is important also related to my comments immediately above and below.

2. We include new quantifications of the number of double-labeled DCX+PSA-NCAM+ cells, and cells only labeled by one of these markers in the EC stream and EC from birth to 11 months (See new **Extended Data Fig. 5i,j** and also the response to points 1, 7, and 8). The vast majority (76.4%) of cells expressing either DCX or PSA-NCAM within the EC stream were double-labeled. We believe this, together with the frequent presence of a leading process, strongly supports the conclusion that this is a population of young neurons in the infant human brain.

Experimental birth dating of individual cells is not possible at these ages in humans, however, in response to this question, we performed several new analyses. We analyzed 6 new gestational ages including 23 GW, 28 GW, 29 GW, and 34 GW to perform additional (1) snRNA-seq and (2) histological quantifications of dividing progenitors. Our new snRNA-seq analysis allowed us to identify interneuron maturation trajectories and calculate a maturation-associated pseudotime across all samples and ages (**Fig. 6g**). Gene co-expression analysis revealed modules of genes that had their expression associated with pseudotime as well as known sample age, identifying groups of genes that have their expression turned on or turned off as interneurons mature. This transcriptomic profile allowed us to infer the maturation state of different cell populations in our analysis instead of relying on individual markers such as DCX and PSA-NCAM (**Fig. 6j, Extended Data Fig. 12**). The postnatal interneurons in the EC stream at 14 days were at a maturation state very close to the interneurons in the 23GW EC or 23GW germinal zone (**Fig. 6k**). In contrast, the interneurons in the EC at 14 days exhibited a more mature transcriptomic profile. This further demonstrates that the neurons found in the postnatal EC stream are young.

Next, to identify the stages in gestation when these young EC stream neurons could possibly be born, we performed immunostaining for dividing cells and neural progenitors in the EC stream region and the nearby CGE from 18 GW, 22 GW, 28 GW, 29 GW, 34 GW, and 38 GW. This investigation shows how highly proliferative the CGE remains into late gestation and early postnatal stages (**Fig. 5g,h**) and further demonstrates that dividing CGE progenitors are closely associated with the origin of the EC stream at each age examined (**Extended Data Figs. 6,7**). Together our analysis finds that the EC stream forms as a migratory route between 23 GW and 27 GW as the ventricle collapses (For examples of pre-closure at 22 GW see **Fig. 3, Extended Data Fig. 3, 6, 9** and for examples of post-closure at 29 GW, see **Fig. 3, Extended Data Fig. 3, 6**). Our new quantitative analysis of dividing cells in the stream and CGE shows that the CGE retains a small population of proliferative cell clusters and migratory DCX+ neurons at birth, which we also corroborated with new ultrastructural images (**Fig. 5f**). Together with the sn-RNA seq data of the EC stream which found that these cells are not produced in the stream, our data support a mid- to late-gestational birthdate within the CGE for the young neurons in the stream. We find that the CGE progenitors continue to divide after birth, possibly generating a small number of EC stream neurons.

3. In the description of the EC postnatal migratory stream, I feel that the authors shouldn't avoid adding at least a brief reference to, and comparison with, other developmental and postnatal ... the other major proliferative and migratory zones that have been a crucial part of the literature leading to this study. My strong feeling is that by not mentioning the SVZ-RMS and SGZ-GCL makes the implicit need for comparison more conspicuous, and that this is a major missed opportunity to begin to address head-on the most interesting features of this EC migratory stream. Specifically, by not even providing some reference to known features of RMS neuroblasts or transit amplifying progenitor cells—that are known not to be abundant in humans—the characterization of the EC migratory stream in this paper lacks context and appears more obscure.

Thank you for this important suggestion. The EC stream shares some but not all of the features observed in other regions containing neurogenesis and migration: It is a unique mix of large chains of migratory interneurons like the RMS (**Figs. 1, 4**) but unlike the RMS it also contains pallial radial glial cells in the region of a collapsed ventricle (**Figs. 3–5**). In our revised manuscript, we now include (1) new images of the human RMS at birth for molecular comparison, (2) new snRNA-seq data of the stream characterizing the types of cells present, and (3) changes in the discussion to include additional comparisons of the EC stream to other major proliferative and migratory zones.

Results show that, unlike the cylindrical nature of the RMS, the EC stream is an extensive lamina filled with migratory chains extending across the medial temporal lobe (**Fig. 1**). We also show that similar to the RMS, the EC stream has glial processes ensheathing its migratory cells (**Fig. 3d,f**). Molecularly, the RMS is enriched in SP8+ neurons, but we find very few (<2%) of EC stream neurons are SP8+ and the vast majority are COUPTFII+ (**Extended Data Fig. 4a–e**). In our revised manuscript we investigated COUPTFII expression in the human RMS at birth, and found very low co-expression with DCX (**Extended Data Fig. 4g**). Together these results demonstrate that the interneurons in the RMS and EC stream express different transcription factor markers, but share some glial structural architectural characteristics. Our revised results also contain further cell-type analyses from snRNA-seq of the EC stream, showing that the neurogenic aspect of the EC stream is a remnant of the excitatory neurogenesis of the pallium. In addition to pallial-like radial glial cells, we captured *EOMES*+ intermediate progenitors in the EC stream (**Figs. 4b, 5d**) confirming that the neurogenic process found in the EC stream reflects what is found in cortical development, and does not appear to be generating the EC stream interneurons.

In our revised discussion we make a more explicit comparison to other regions including the SVZ-RMS and SGZ-GCL. We describe that, like other streams that have been described before, the RMS and Arc, and MMS, all seem to be decreasing with postnatal age. In contrast, the EC stream cells are still being recruited into the EC throughout infancy and in the brains of young children as old as 2-3 years. In the RMS, it is believed that the stream forms at the site of a formerly open extension of the lateral ventricle. Here we illustrate for the EC stream the ventricular collapse that occurs in the temporal lobe and describe the perior when this happens. We show how the V-SVZ cell populations persist until postnatal ages in the region of the collapsed ventricle.

4. Within the last portion of the description of findings for Fig 1, from lines 86-102, more attention is paid to the subclassification of cell populations in the region of the EC migratory stream and known features of the caudal ganglionic eminence lineage. However, a more precise and comprehensive characterization still could be applied using a number of potential techniques, and the potential relevance of these cells could be foreshadowed. This could involve more extensive longitudinal characterization of cell fate commitment and cell subtype identity markers within the EC interneuron lineage (possibly compared with that known of other adjacent niches and lineages). The authors already have performed an extensive screen of cells subtypes using Ki67, SOX2, Vimentin, MCM2, SP8, DLX2, DCX, PSA-NCAM, COUPTFII and PROX1 markers in various combinations in Fig 1 and extended Figs 1-4. However, the quantification, analysis and meaning of this does not come through to its full potential in the text or the main figures (e.g., lines 94-100 and elsewhere). This issue also leads into comments noted below concerning evidence that mature EC interneurons actually include cells derived from the postnatal/juvenile EC migratory stream...or that immature postnatal EC stream interneurons attain functional importance. Indeed, more information or analysis that could support the functional relevance of such cells could strengthen this manuscript substantially.

4. As indicated above, we completely adopted this recommendation. Our revised manuscript now includes new snRNA-seq datasets from (1) microdissected progenitor regions in mid-gestation (2) microdissected cells from the EC stream at 14 days of age and (3) microdissected EC from additional postnatal ages. As the reviewer suggests, this has permitted a more extensive longitudinal characterization of the cellular fate commitment stage of the neurons in the EC stream itself as well as interneuron subtype identity markers within the lineage. Specifically, the new data fully support the CGE origin of the majority of neurons within the postnatal EC stream and reveals a very interesting interneuron subtype that is enriched within the chains of migrating neurons (LAMP5+ cells). We apologize for the confusion with the immunocytochemical description; the majority of EC stream cells express DLX2 and COUPTFII, consistent with the new transcriptomic data. We have reorganized the presentation of these data and moved from supplementary figures the staining for Ki-67, DLX2, and COUPTFII co-stains into the main figures (**Figs. 4e,f,h,i, 5a,e**). By “mature EC interneurons” we assume that the reviewer is asking about cells that originate in the EC stream. We find no evidence in the new analysis of mature EC interneurons derived from primary progenitors in the EC stream. The majority appear to be derived from the CGE. If the reviewer is referring to interneurons that mature in the EC stream, we find no evidence of neurons that settle in this region; more evidence in support that this is a transient structure containing migrating young neurons on their way to other locations.

5. The description of the establishment of the anlage that will become the EC migratory stream in lines 114-127 and 142-147 is particularly fascinating. This process sounds similar to the temporally and anatomically adjacent establishment of the hippocampal DG/SGZ anlage as well. It would be especially interesting if the authors were able to mention this similar phenomenon and then compare/contrast this new neurodevelopmental process. Otherwise, some readers may wonder if this EC stream is, indeed, distinct from dentate gyrus/hippocampal development occurring nearby through relatively similar processes. Additionally, could some of the basic features also be described with a corroborating histological technique? Immunohistochemistry vs immunofluorescence, or even in situ hybridization?

5. We agree that the comparison with the DG/SGZ migratory stream is interesting as in both regions there appears to be an entrapment of RG away from the ventricle within the stream (see also response 3). In the DG, however, this migratory stream does not form in the location of a collapsed ventricle. We added a short comment to the discussion to clarify that despite the proximity of the EC stream to the hippocampus, the migration from the dentate neuroepithelium to SGZ is quite different as it is the primary progenitors that migrate. Also unlike the SGZ, which is neurogenic later into childhood, we see that migration in the EC stream and adjoining EC subsides by 2 years of age. We now include DAB immunohistochemistry for PSA-NCAM as a corroborating histological approach which reveals the clusters of neurons migrating in the EC stream (**Extended Data Fig. 1c**).

6. Following the first major point of this study, that the EC (immature neuron) migratory stream exists, the second major point of this study is that many of its components are depleted over time. However, when the latter findings are based upon human tissue, it is important to exclude disease-related processes potentially related to the early demise of young donors that also could have a major confounding influence on such cell populations. Indeed, cancer, seizures, chronic inflammatory state, neurodevelopmental disorders are all profoundly overrepresented in patients ultimately contributing to brain banks, and these pathological states have a major impact on the health and persistence of generative regions and processes in the brain. At minimum this should be

addressed in the discussion, and ideally could be conveyed in parallel with the results to provide clarity to the reader and solidity to the findings at the time. If such assurances cannot reasonably be made based upon known features of donors—must also be clarified and expanded within the Methods section—then the authors should strongly consider performing a parallel experiment from primate tissue for confirmation. It stands to reason that, if these findings are real and important, they will be reproduced in closely related species and will be profoundly fortified by such corroboration.

6. To address this important point, we have, (1) added new samples across critical ages, (2) reported post-mortem intervals and clinical histories (when known) for all samples when available, (3) added epilepsy surgical resections, (4) included a mention of this caveat in the discussion, and (5) analyzed 7 new Rhesus macaque brain samples from birth to 17 months.

To diversify the types of samples analyzed for this study as well as the possible confounds and comorbidities, we have added a total of 15 additional samples (see summary table below) in our revised manuscript including new gestational, infantile, and adult ages. Among the now 55 total cases, we include 2 pediatric samples from anterior temporal lobe epilepsy resections, and in our case table (**Supplementary Table 1**) we report postmortem interval and available clinical histories. Across all of these samples we consistently observed the EC stream between birth and 2 years of age (total = 29 cases between birth and 2 years). We also consistently observed a progressive decline in the number of cells within the EC stream with postnatal age together with their redistribution to the EC along a similar timeline (**Fig. 2a–c, Extended Data Fig. 2a–e**). The redistribution of the cells followed the pattern predicted by the orientation of their leading processes (**Fig. 2b, Extended Data Fig. 2b–d**). We also could not detect this stream, or even any cell clusters similar to what was observed in samples <11 months of age in the older adolescent and adult samples (n = 10 cases between 3 and 77 years). Together we believe that these data strongly indicate that the presence and decline of this migratory stream is not artefactual.

We have also followed this reviewer's suggestion and investigated whether postnatal macaque temporal lobe samples had an EC stream, and this suggestion has resulted in one of the most interesting observations in the revised manuscript. Surprisingly, in the newborn macaque temporal lobe, we did not find the dense chains of migrating neurons that are characteristic of the newborn EC stream in humans. Instead, we only found a few individually migrating DCX+ cells in the region where the EC stream is found in humans (**Fig. 2d,e, Extended Data Fig. 2f,g**). We also did not observe individually migrating DCX+ cells in the cortical plate or anywhere between the EC stream region and the EC. We investigated this further by co-immunostaining the DCX+ neurons in the temporal lobe V-SVZ for DLX2 and found that the majority were interneurons (**Fig. 4g, Extended Data Fig. 5f**). However, we also found that the majority also expressed SP8 instead of COUPTFII, the latter of which co-expressed by the vast majority of human EC stream neurons (**Extended Data Fig. 4h**). We next examined the brains of 4 macaques collected at 3 months of age to search for any evidence of this postnatal migratory route (**Fig. 2d,e, Extended Data Fig. 2h**). Using the RMS as a comparison where we expected to find DCX+ cells at 3 months, we found dense collections of DCX and DLX2 co-expressing neurons in the RMS, but the EC stream region had no DCX+ cells. The same absence of DCX+ cells was further confirmed at 6 months and 17 months (**Extended Data Fig. 2i**). Taken together, these observations suggest that this process is dramatically extended into human early childhood, compared to the macaque.

New cases added during revision:

Case no.	Age	Gender	PMI	Experimenta I Use	Neuropathology	Clinical history
41	23 GW	M	18h	IHC	control	Amnion infection syndrome
42	23 GW	F	<1h	snRNA-seq	control	spontaneous abortion
43	28 GW	M	24h	IHC	control	prematurity
44	29 GW	F	24h	IHC	control	Necrotizing enterocolitis
45	34 GW	M	22h	IHC	HIE	subarachnoid hemorrhage
46	39 GW	M	60h	IHC- RMS	control	Multifocal hemorrhage
47	14 days	M	27h	snRNA-seq	control	Congenital diaphragmatic hernia (CDH)
48	3 months	M	15h	IHC	control	Congenital heart disease with heart failure
49	2 years	F	<30	IHC	ATL	cortical displasia
50	13	F	15h	snRNA-seq	control	not available
51	27	M	13h	snRNA-seq	control	not available
52	55	M	<48h	IHC	control	heart failure
53	55	M	20h	IHC	control	Constrictive pericarditis,
54	74 years	F	<4hr	IHC	control	lung mass and seizures in mid frontal cortex
55	77	M	50h	IHC	control	VHL with hemangiomas

7. Following from earlier comments concerning more detailed fate mapping and characterization of EC migratory stream DCX+PSA-NCAM+ double positive cells, I am curious if more detailed subclassification can be done at later ages and for the differentially migrating radial vs tangential patterns described in lines 160-175. Although aspects of this question are addressed in the following two sections, I think it would be interesting to note here whether radial vs tangential orientations had different patterns of DCX+ or PSA-NCAM+ single vs double labeling, and whether co-expression of Ki67 or PROX1, etc. also could be assessed. Establishing age-related migration patterns of more/less mature cells within and from the EC migratory stream could be particularly interesting... especially as this might relate to their functional or therapeutic potential.

Additionally, within this section of the following section detailing the late-migrating interneurons, could high-resolution microscopy be included to define some structural-functional features of these cells? Does one group or the other have different patterns of +/- en passant synapse formation or other functional interaction with neighboring cells? Is there evidence of neurotransmitter synthesis yet? To elaborate upon immunostaining, could the authors apply snRNA-seq to profile the components of the EC stream in greater detail? And could this lead to a more expansive characterization of the EC migratory stream interneuron lineage relative to age and cell characteristics as also portrayed in this section?

7. In response to this suggestion the revised manuscript now includes (1) quantifications of DCX+ and PSA-NCAM+ single vs. double labeling. (2) Quantifications of Ki-67+ cell densities and histology of Ki-67+ cells and co-stains for PROX1 and LHX6, (3) Deconvolution imaging of the stream achieving higher resolution limits 120nm X/Y, 300nm Z, and additional EM images with ultrastructural details (see new **Figs. 1c, 5f, Extended Data Fig. 1b**) and additional searching for the presence of en passant synapses using these high-resolution approaches, (4) the aforementioned snRNA-seq of the microdissected EC stream which has revealed the stages of cells with neurotransmitter synthesis gene expression. We describe each of these additions in more detail below:

(1) We assessed DCX+ and PSA-NCAM+ single vs. double-labeling between birth and 11 months of age. We compared staining in the stream where the vast majority of cells are tangentially oriented to the EC where the vast majority of cells are radially oriented (**Fig. 2b,c, Extended Data Fig. 2b-d**). This comparison showed that the frequency of double-positive cells was much greater in the EC stream than within the EC, possibly reflecting a maturation progression as the reviewer suggests (see new **Extended Data Fig. 5i,j** and also response to points 1, 2, and 8). Evidence from the EC stream microdissection shows that interneurons isolated from the EC stream were not heterogeneous in their maturation state. These interneurons in the EC stream at 14 days of age had a gene expression profile more similar to immature interneurons found in the germinal zones and the EC at 23GW than to its adjacent EC at 14 days of age, supporting the conclusion that they are immature within the stream and begin to mature upon entering the EC (see also response to point 2).

(2) We present new and reorganized Ki-67 stains and quantifications within the EC stream and CGE from mid-gestation to birth (**Fig 5. a-c, e, g, h, Extended Data Figs. 3a-c, e, 6-9**). DCX+PSA-NCAM+ cells (in chains or individually) were Ki-67 negative. Together these new quantifications and the new snRNA-seq data provide a greater context to the age and maturation state of the stream (see also response to point 2). We also now include transcriptomic analysis and immunostainings for PROX1, which we found to be present in a subpopulation (16.0% at birth) of the migrating cells in the EC stream (**Extended Data Fig. 4f**), consistent with our observation of the CGE origin of EC stream cells. For more explanation of our PROX1 findings, please see response to point 8, below.

(3,4). Our new transcriptomic analysis indicates that migrating cells in the immature neurons found in the EC stream, 23GW EC and germinal zones do express several genes associated with synaptic transmission, (e.g. *GAD1*, *GAD2*, *SNAP25*, *VAMP2*, *SYT1*, Extended Data Fig. 12). GABAergic receptor genes such as *GABRB2* and were highly expressed by more mature interneurons in the cortex of donors older than 2 years and also by some of the migrating neurons in the EC stream (**Fig. 6d**). We do not know if the presence of transcripts of these genes in these cells lead to protein expression in these young cells. Studies indicate that immature migrating interneurons can secrete (non-synaptic) GABA and are responsive to this neurotransmitter (Luhmann et al. 2015. *Frontiers in Cellular Neuroscience*). We did not observe synapses between migrating neurons in the EC stream or in individually migrating cells using high-resolution deconvoluted confocal or electron microscopy (**Fig. 1c,f, Extended Data Fig. 1b,e**), but we were able to observe ultrastructural features associated with a migratory immature neuron (adherens junctions, leading centrosome, few organelles, compacted chromatin) (**Figs. 1f, 3f**). Our maturation trajectory analysis revealed that the expression of many of these synapse-related genes not only correspond to a discrete gene module in our gene co-expression analysis (Module 1 in **Fig. 6j, Extended Fig. 13**), but that also shows that the combined expression of this module is strongly associated with neuronal maturation, i.e. despite

immature migratory neurons expressing these genes, they do at much lower levels than interneurons in the adult EC. We now include a detailed description of synapse-related genes in **Extended Fig. 13** for developing interneurons to show their relative abundance (and upregulation) within these cells compared to the EC at progressive infant and toddler ages.

8. At risk of asking the authors to perform major additional histological analyses, I wonder if the description of EC migratory stream neuroblast fate commitment toward GABAergic interneuron identity could be expanded to provide finer analysis and deeper context using staining that already might have been performed. Although DCX+DLX2+ cells were noted, could PSA-NCAM+DLX+ double positive cells also be assessed? Would this help draw parallels or distinctions with transit amplifying cells and interneuron progeny in the RMS? When comparing the alternate fate commitment route suggested by DCX+TBR1+ double positive cells, was staining with PROX1 (already reported in other contexts) also considered? For instance, would DCX+PROX1+ double positive cells indicate a different subset from the DCX+DLX2+ (or PSA-NCAM+DLX2+) double positive cells that are the population at the center of this finding? If this broader and more detailed context can be supplied from the analysis of staining that might already have been performed, I think this could allude more clearly to a biological meaning of this proposed cellular/developmental process.

8. We prepared new DCX/PSA-NCAM/DLX2 co-stains of the stream at birth, 3 months, 7 months, and 11 months. At birth we found that the majority (82.4%) of the cells in the stream are DCX/PSA-NCAM/DLX2 positive. We assessed DCX and PSA-NCAM single or double-labeling together with either COUPTFII or DLX2 (**Extended Data Fig. 5i,j**) across the same age range from birth to 11 months. This analysis revealed a large fraction of single DCX+ or PSA-NCAM+ cells in the EC also co-expressing either COUPTFII or DLX2, and far fewer of these single-positive cells within the EC stream. We also prepared DCX/PSA-NCAM/PROX1 co-stains at birth, 3 months, 7 months, and 11 months. Across these ages we found a small proportion of DCX+PSA-NCAM+ cells in the stream that were PROX1+, which we quantified at birth and found corresponded to 16.0% of the cells.

Our new transcriptomic analysis is consistent with these observations, showing that the EC stream has a large proportion of *DLX2*-expressing cells and *NR2F2* (*COUPTFII*)-expressing cells (**Fig. 4b-d**). Our RNAseq data indicate that *PROX1* expression is mainly confined to CGE-derived cells in our interneuron maturation dataset, and shows expression within the EC stream interneurons (**Extended Data Fig. 12**). We observe *PROX1* expression in both immature and mature CGE-derived cells and it is evidently not restricted to a particular CGE-derived neuronal class (*VIP+*, *RELN+*, *LAMP5+*) (**Extended Data Fig. 12**). At the protein level, as mentioned above, we only observed 16% co-labeling of EC stream interneurons with PROX1 in neonates. We also observed many brightly-labeled PROX1+ cells in the EC stream region that did not co-stain with DCX. Within the EC stream microdissection, *PROX1* was strongly expressed in the oligodendroglial lineage (consistent with data reported in Bunk et al., *Stem Cells* 2016). Together our data suggest a limited correlation between the abundance of *PROX1* transcripts and PROX1 protein expression within the EC stream and further support the DLX2+COUPTFII+ identity of these neurons.

9. Findings concerning EC migratory stream DCX+ cell features with respect to MGE vs CGE lineages are potentially quite compelling. However, further validation, contextualization and analysis could be helpful. Firstly, in this regard, I wonder if much of the depth of analysis that could be gleaned from experiments portrayed in Suppl Fig 7 are sufficiently explained in the text or are conveyed via the related panels of primary Fig 3. In this respect, this topic could be very nicely

corroborated and expanded using snRNA-seq methods as was reported in the following Results section (outlined in Fig 4). However, it is not clear that this also was intended to lend validation to the breadth of immunostaining findings provided in Suppl Fig 7, as this still would be helpful.

9. We agree, and as mentioned above, we now provide snRNA-seq for the EC stream and present this data in new **Figs. 4 and 5**. In the new **Fig. 6** we analyze them together with cells from mid-gestation germinal zones and mid-gestation, infant, childhood, and adult EC. Together this analysis provides new details of EC interneuron identity and maturation state. Consistent with the histological data, at birth, we find that the vast majority of interneurons in the EC stream are CGE-derived, with far fewer MGE-derived cells present (**Fig. 4b-d**). The majority of these CGE-derived immature interneurons in the EC stream correspond to immature neurons of the *LAMP5+* subtype, a neuronal class that is absent in the EC at 23 GW. The stream also supplies the EC with smaller numbers of young *VIP+* and *RELN+* neurons. We corroborate these findings with stainings for COUPTFII, Lamp5 (*LAMP5+*), Reelin (*RELN+*) and Calretinin (*VIP+*), detecting these double-labeled cells in the stream and in the EC (**Fig. 6l-n**). We were also able to determine that these neuronal types supplied by the EC stream predominantly settle in the upper layers of the EC (**Fig. 6o**).

10. As noted in the comment above, addition of snRNA-seq experiments can provide great benefit to this study overall. Experimental samples in this section appear to match the postnatal age ranges covered in experiments above, aside from the brief reference to the population of DCX+TBR1+ cells that could be measured at age 6 and 13 years. Later in this section, and with regard to more diverse lineages of interneuron subtypes, it's interesting to see the increased expression of multiple ATP-synthesis genes. However, since some of these genes were encoded by mitochondria DNA, is it possible that the differences are caused by quality differences between the two datasets, i.e., one dataset has higher mitochondria reads? The authors also might check other mitochondria genes and evaluate whether this increase is specific to ATP-synthesis genes alone, or broadly for all mitochondria genes. Similarly, the authors also could check if all vs a subset of ribosomal genes are differentially expressed.

Finally, if possible, the authors should include an additional/alternative analysis (e.g., RNA velocity) to confirm the lineage progression in Figure 4. Currently it seems that the real ages have a lower correlation with pseudotime in certain lineages (e.g., lineages 7,8 and 9).

10. We really appreciate this observation and would like to thank the reviewer for pointing this out. Based on this comment, we realized that transcripts of mitochondrial genes should correspond to extra-nuclear transcripts and derive from ambient RNA. In order to address this issue, we revised our QC and pre-processing steps in our sn-RNAseq workflow. We incorporated two steps to minimize the contamination of ambient RNA in our samples: 1) we substantially reduced the abundance of ambient RNA transcripts using SoupX (Young et al. 2020. *GigaScience*) and 2) we eliminated any remaining cells with an abnormally low nuclear fraction (ratio of unprocessed/processed RNA, determined using DropletQC, (Muskovic, et al., 2021. *Genome Biology*) (**Extended Data Fig. 11**). Low nuclear fraction is an indicative of contamination with extra-nuclear transcripts, a common issue that may lead to misinterpretations in single-nuclei RNAseq data from frozen human brain samples (Caglayan, et al., 2022, *Neuron*). We have corrected the MS to remove the inference that mitochondrial genes are enriched in maturing EC interneurons.

One limitation of working with isolated nuclei is the inability to analyze our data with RNAvelocity, as it is based in the amount of spliced and unspliced transcripts to infer trajectories. Nuclei are generally enriched in unspliced transcripts and it is hard to determine if spliced transcripts have a

nuclear origin or are derived from the ambient RNA. Our new microdissections of the germinal zones (CGE, MGE, and LGE) and microdissection of the EC stream, combined with the addition of more ages, supports our pseudotime analysis using Monocle. This analysis shows that the progression of the lineages inferred by Monocle is consistent with the regions and ages of the samples; e.g. lineage trajectories start at dividing cells in the germinal zones at 23GW and end at the postnatal EC at ages older than 2y (**Fig. 6g**). Moreover, the pseudotime and gene modules associated with maturation showed a strong correlation with sample ages (Fig. 6j, Extended Data Fig. 12).

Our gene co-expression analysis revealed a module of genes (Module 4) that also had their combined expression associated with neuronal maturation (Extended Data Fig. 12). This module consists of several ribosomal genes and other non-ribosomal genes. Gene ontology analysis indicate that genes associated with terms such as *cytoplasmic translation*, *ribosome assembly*, and *regulation of ubiquitin protein ligase activity* were significantly overrepresented in this module, confirming our initial findings that a higher abundance of transcripts associated with protein translation is likely associated with an immature neuronal state.

11. The authors should revise the manuscript's title, since "persistent postnatal" may imply a fully realized, lifelong process; yet experiments in this paper span the young juvenile period that may be thematically more "developmental" than "adult." Likewise, the authors may consider providing a more nuanced moniker for the cell populations described in this study, since the word "interneurons" alone does not necessarily convey their status as putative immature, recently born, pre-interneurons or even (subtype-restricted?) neuroblasts.

11. This is a good point, we have modified the title to avoid suggesting that the EC stream we here describe is present for life since this is not the case. The new title is: **Protracted Migration of Immature Neurons in the Temporal Lobe of Young Children.**

Minor concerns

- The manuscript file provided for review was not formatted with an abstract or with the designation of a Background or Introduction section before beginning with the first Results section of "Extensive migration in..."

In our revised manuscript we include a more extensive introduction at the beginning of the results.

- In the introduction or within the first results sections, the authors should consider including a brief reference to the source of the tissue analyzed and characteristics of donors. Even something as simple as "we stained human temporal lobe sections from non-diseased brain bank donors at birth for doublecortin (DCX) and..." in lines 56-57 could provide better context concerning the important features of this study.

We now included a statement about the characteristics of the tissue analyzed at the beginning of the results section.

- In line 192, the authors introduce a new transcription factor by first spelling out "T brain 1" followed by the parenthetical designation of its abbreviation "(TBR1)". I would encourage the authors to use the same format for all other labels or markers reported throughout the

manuscript. Common usage may also advise including a brief definition or explanation of function, as is also included here, with “the cortical excitatory neuron transcription factor.” Please note the contrast with DCX and PSA-NCAM in line 57, and other examples throughout the manuscript.

For all genes that are introduced we now spell out their abbreviations and provide a brief explanation of their function.

- Line 231-232: The authors said these cells consist of a mixture of MGE and CGE-derived interneurons, yet these cells were labeled as MGE-derived in Fig. 4d. This seems inconsistent or confusing. These cells are the LAMP5+/LHX6+/NKX2-1+ cells and previous studies suggested MGE progenitors can give rise to these cells (Tasic et al., 2018). This sentence needs to be updated. Finally, clusters 3,4,5,21 also are positive for LHX6 and NR2F2, yet these cells appear to be ignored.

These cells in the central cluster in our previous dataset were referred to as "MGE-CGE mix" in Fig. 4d. In our current dataset, this CGE-MGE mix of postnatal neurons is no longer present. It is possible that having more immature neurons from 23GW samples in our current dataset resulted in better clustering of these immature neurons, as some of the postnatal EC neurons can be found at slightly earlier points in pseudotime (**Fig. 6f, i**). A fraction of cells in clusters 3,4,5,21 expressed low levels of *NR2F2* and had much higher expression of *LHX6* and *SST*, consistent with an MGE-derived SST+ identity.

- Line 263-265: It’s challenging for the reader to get this claim from Fig. 4h. For clarity, the authors should better explain the plot or provide additional analysis.

We have improved the analysis on interneuron maturation and now provide a much clearer identification of the identity of migratory neurons in the EC stream.

- Line 336-338: comments about RELN+ cells and Alzheimer Disease pathology in this context seems like a forced association.

These comments have been removed.

- Line 531-532: should be “UMI counts” rather than “reads”.

This has been corrected in our revised manuscript.

Referee #2 (Remarks to the Author):

The manuscript by Nascimento and authors continues the remarkable expansion of inhibitory cell types that are made postnatally in humans. Here by exploring the entorhinal cortex the authors have discovered a late maturing CGE-derived population that appears to produce a range of CGE derived cells that may give rise to a remarkable diversity of CGE interneurons in the entorhinal cortex. Like many of the studies that preceded this, it uses a combination of RNA and antibody stains to longitudinally characterize and detail the types and ages that these populations become integrated, in this case in the entorhinal cortex. Fascinating as this detailing of cell types is, the analysis seems

rather limited both in their analysis of the origins of these cells and the types they give rise to compared to other from the Alvarez-Buylla group, in particular the recent publication from Pollen et al. The methods used are careful and well illustrated but descriptive and given no direct inkling of the function of these populations beyond describing their existence. In sum, while a careful and interesting addition to the existing literature, it does not reach a level that would make it appropriate for publication in Nature. Among the things that are too superficial to argue that this a sufficient advance are the lack of insight this work provides with regard to the origin, diversity or potential significance of these observations beyond what has been shown extensively by both the Kriegstein and Alvarez-Buylla laboratories. What is described as lineages amount to gene expression trajectories which neither identify that they **arise from a distinct progenitor pool** nor delineate how the 9 lineage type relate to **functional classes**. Given the reliance on markers but lack of functional analysis the best the authors can conclude is that further neurogenesis akin to that described by both Alvarez-Buylla, Nowakowski, Kriegstein and Pollen can now be extended to the entorhinal cortex. While this is a solid descriptive study it fall far short of an advance that would be suitable for publication in Nature.

We appreciate the reviewer's interest in the diversity of CGE-derived interneurons that we are uncovering in the postnatal human brain. While there are parallels with our previous studies in the frontal lobe, there are very important differences that we now discuss (see revised Discussion). The work from Nowakowski's, Kriegstein's, and Pollen's laboratories focus on early gestational periods and also did not study the temporal lobe. To our knowledge, this is the first identification of a large population of young migrating neurons that persists for at least 2 years of human postnatal life and is associated with the temporal lobe cortex. The stream supplies young neurons to a region of the brain that is of general interest: the most interconnected region of the neocortex that is key to memory consolidation, and orientation and linked to neurodegeneration in Alzheimer's. To address the reviewer's questions about the origin, diversity or potential significance, of these cells, our revised study now includes new data that identifies the origins, assesses the diversity, and highlights the overall significance of our findings: To summarize, we have added new microdissections of putative germinal zones (CGE, MGE, LGE) and the EC from mid-gestation (23 GW); new microdissections of the EC stream at 2 weeks of infancy; and new adolescent and adult microdissections of the EC itself (**Figs. 4, 6, Extended Data Fig. 10**). We have analyzed this new dataset to trace the EC stream neurons from their point of origin through their transcriptomic maturation as they migrate into the EC in young children. We also used this dataset to explore the diversity of cell types they generate and confirmed the presence of and mapped the ultimate location of these neurons within the EC (**Fig. 6**).

We also expanded our study to include 7 new postnatal non-human primate (Rhesus macaque) brains from birth to 1.5 years in an effort to develop a primate animal model to further investigate the importance of this process in the plasticity of entorhinal cortex circuits. Unexpectedly, we found no dense clusters and only a few migrating neurons within the same region of the macaque brain at birth, and despite searching 4 separate macaque brains at 3 months of age, were unable to find a comparable migratory process in the infant monkey. We present positive controls alongside this unexpected negative data, as well as structural evidence of the differences in the macaque that might explain this divergence between species. A similar stream might be present during gestation in macaques, but a comparable postnatal window for the maturation of interneurons in the EC does not appear to be present in all primates, although it is markedly a significant feature in humans. Below we respond point-by-point to the critiques with references to the new data added in our revised manuscript:

Critique: “limited both in their analysis of the origins of these cells and the types they give rise to”.

Response: To better understand the origins and types of neurons migrating in the EC stream, in our revised manuscript we performed a new snRNA-seq of the microdissected EC stream at birth and of the MGE, LGE, and CGE at 23GW. We also include new histological quantifications across gestational and infant ages which corroborate this new transcriptomic analysis (**Figs. 4-6 and Extended Data Figs. 7-13**, also please see our response to reviewer 1 comments #2, 4, and 9). These data confirmed our previous observations that the vast majority of the migratory neurons in the EC stream derive from the CGE. The dataset also revealed that the majority of EC stream migratory neurons are *LAMP5+* interneurons, with a smaller number of *VIP+* and *RELN+* cells. Our analysis also revealed that these neuronal classes supplied by the EC stream settle mainly on the upper layers of the EC, where the perforant path that connects to the hippocampus starts. Interestingly, *LAMP5+* interneurons have been found to be vulnerable in Alzheimer’s disease (Deng et al. 2022), and some of the first neurons to die in AD are also located in the upper layers of the EC. To expand beyond a reliance on specific markers for cellular identification, we analyzed the transcriptional signature of cells in the human subpallium and entorhinal cortex beginning at 23 gestational weeks, infancy, toddlers, adolescence, and adults up to 79 years of age. Microdissection also allowed us to link the identity of subpopulations of immature neurons in the EC stream to that of mature neurons in the entorhinal cortex, beyond the use of subtype markers, and further characterize the identity of these migrating neurons and how they are added to the entorhinal cortex at a much later time in childhood development.

Critique: “neurogenesis akin to that described by both Alvarez-Buylla, Nowakowski, Kriegstein and Pollen can now be extended to the entorhinal cortex”

Response: We describe for the first time a stream of migrating neurons in the temporal lobe. This stream persists postnatally for much longer than any other neuronal migratory process in the human brain. In our investigation, we identified the source of these neurons, the developmental processes involved in the formation of their migratory route, the specific neuronal classes they become, and the cortical layers they settle on. Critically, our investigation is also more expansive than those that looked earlier or in different regions: we include samples from mid-gestation, late-gestation, infancy, childhood, and adults to create a comprehensive transcriptomic profile of neuronal maturation in the human temporal lobe together with histological verification across these ages. Beyond these observations, which are different from previous work, our revised manuscript presents new data that show that the EC stream is enriched in a unique population of interneurons (*LAMP5*). We have also added new comparative data suggesting that the EC stream is a prominent process in humans compared to the macaque. The novelty of a major neuronal migration in the temporal lobe associated with brain regions key to higher brain function and neurodegeneration, we believe, is a significant new discovery.

We also clarify that the EC stream is not a new neurogenic region for interneurons in the stream. We clarify that if there is any residual neurogenesis producing migrating neurons in the EC stream it appears to derive from the remnants of the CGE that we observe at birth; we also document how CGE proliferation decreases rapidly during postnatal ages (see new **Fig. 5g,h**).

Critique: “no direct inkling of the function of these populations”.

Response: In our revised manuscript we provide several important steps towards this goal by first revealing the phenomenon, measuring the duration of the process, identifying the subtypes of interneurons involved, and determining the layers of the EC they populate. We believe the new data

presented has identified the origins, molecular identity, and anatomical destinations, providing a foundation for further investigation of the role(s) of the different subtypes of interneurons that are carried by the EC stream. Note that the function of postnatal neuronal recruitment in other brain regions, including where animal experimentation is available, remains speculative after decades of research. Intriguingly, several of the neuronal cell types we identified as derived from the EC stream are affected during neurodegeneration. We show how the EC stream forms as part of the ventricle collapse in the temporal lobe between GW 22 and GW27, creating an anatomical scaffolding for the extended postnatal migration of neurons into the medial temporal lobe. Its discovery highlights an important aspect of human brain development that currently cannot be mimicked *in vitro* and should be considered when using animal models that lack this structure. We would like to emphasize that this work is revealing a new process in humans that cannot be studied dynamically with any other experimental methods currently available.

Reviewer Reports on the First Revision:

Referees' comments:

Referee #1 (Remarks to the Author):

The authors have been able to address and complete all of my (fairly extensive) suggestions even more comprehensively than anticipated. Overall, they now have been able to outline cell subtypes, origins and migratory phenomena in a way that explicates a process and contributes to greater understanding to a substantial degree. I believe that this now marks a contribution that is worthy of publication in Nature.

Referee #2 (Remarks to the Author):

In their revised manuscript the authors do a fine job of outlining the difference between the present work and that of previous investigators (including themselves) who have explored the questions they raise. Among other aspects that they highlight, their drilling down on the CGE-derived aspects of their work is indeed both thoughtful and penetrating. In the revised manuscript, the authors do a particularly nice job of illustrating that the LAMP5+ cells are not the MGE-derived Lhx6 population, which given the illustration by another group is of considerable interest. Nonetheless, it would be nice if they could illustrate more deeply the identity of the CGE populations. While the VIP, REELIN, CR and LAMP5+ populations. While this rather nicely summarizes those that are CGE derived, it would be of considerable interest to better classify them (Alpha7, CCK, SNCG, VIP multipolar vs bipolar, etc). The analysis focuses upon the stream of interneurons that arise from the CGE and reside for a surprisingly long period in the temporal lobe is intriguing, as it highlights how this population might influence key aspects of development, including memory consolidation, and Alzheimer's disease. However, if the authors could better determine which of the populations are deleted, it would certainly add to their analysis. Beyond this, their comparison of humans to Rhesus Macaques is both insightful and clarifying, as it nicely illustrates differences in these two rather closely associated species. The comparison of how the species identified in humans is absent in the monkey nicely illustrates the differences in the cognitive development of these two species. Moreover, the authors do an excellent job of illustrating that the phenomenon they explore in EC is not tractable in any other species than humans. Taken together, the work presents a nice complement of the way that CGE interneurons have develop in a manner unique to humans, with a particularly exciting nod to psychiatric disease.

Referee #3 (Remarks to the Author):

The authors used histological and snRNA-seq data from valuable pre- and postnatal human samples to examine in detail postnatal immature neurons associated with the entorhinal cortex. Three-dimensional data revealed that the EC stream is a lamina structure with an abundance of immature neurons, in contrast to the tubular structure of the RMS. In the entorhinal cortex, DCX-positive cells were found in the brains of even 2-3-year-old infants, suggesting a long-term supply of neurons during the postnatal period. snRNA-seq data suggest the immaturity of cells in the EC stream and the maturation of EC cells, supporting the authors' idea that mature interneurons are supplied from the EC stream to the ECs. The results presented in this paper, which elucidate some of the mechanisms of normal human brain development, are likely to be of immediate interest not only to specialists in neurogenesis, but also to researchers in multiple specialties, including pediatrics, psychiatry, and neurology. However, I believe that this paper can be improved if the major and minor concerns I have indicated can be addressed.

Major point

1. Additional data have strengthened the idea that the EC stream is a likely source of neurons to the ECs, but there is still no evidence that mature interneurons in the ECs are supplied by the EC stream. While the authors may feel that macaques, which leave only sparse DCX-positive cells at birth and differ in the molecular identity of DCX-positive cells, are insufficient, if live macaques are available to label cells in the EC stream and to observe their migration and differentiation into mature neurons, the authors' conclusion would be more strongly supported.

2. The lamina structure appears to be two-layered in the new 3D data added by the authors, and a two-layered structure of DCX-positive cells is also observed in the extended data Fig. 3E (birth). Throughout the paper, the collapsed part of the ventricle is considered as an EC stream and a source of neurons to the postnatal ECs. The upper layer in the figure is reasonable as the collapsed part of the ventricle, but the lower layer would need some explanation.

3. I. 71-73 "but the development of the unique layer structure of the EC follows an atypical pattern of neurogenesis compared to other cortical regions in many mammals including humans"
I assume that this somewhat vague description is due to the strict word limit, but a more specific description of the differences between the EC and other cortical regions would help clarify the significance of studying the EC.

4. The authors use fetal period samples as Birth samples, but this is not clear without a careful reading of the following text:

Method I. 499-500 "For infant cases, when the brain is at full term (37 to 40 gestational weeks), we refer to this as "birth"."

I think it is understandable that the authors have mixed prenatal and postnatal samples because human samples are difficult to obtain. For these samples, I think it is acceptable to write "birth" in the figure, but at least in the text, it would be better to use more precise expressions such as "around the time of birth" instead of "at birth".

Minor point

1. Fig. 4k I think the error bar should be shown.

2. Fig. 6 Please unify the variations in notation. For example, are EC stream (14 d), Migratory Stream, EC (14d-79y), and Postnatal EC the same thing?

Author Rebuttals to First Revision:

Referees' comments:

Referee #1 (Remarks to the Author):

The authors have been able to address and complete all of my (fairly extensive) suggestions even more comprehensively than anticipated. Overall, they now have been able to outline cell subtypes, origins and migratory phenomena in a way that explicates a process and contributes to greater understanding to a substantial degree. I believe that this now marks a contribution that is worthy of publication in Nature.

We are pleased to hear that the reviewer finds our work now suitable for publication and would like to thank this reviewer for all the helpful comments and constructive assistance throughout the review process.

Referee #2 (Remarks to the Author):

In their revised manuscript the authors do a fine job of outlining the difference between the present work and that of previous investigators (including themselves) who have explored the questions they raise. Among other aspects that they highlight, their drilling down on the CGE-derived aspects of their work is indeed both thoughtful and penetrating. In the revised manuscript, the authors do a particularly nice job of illustrating that the LAMP5+ cells are not the MGE-derived Lhx6 population, which given the illustration by another group is of considerable interest. Nonetheless, it would be nice if they could illustrate more deeply the identity of the CGE populations. While the VIP, REELIN, CR and LAMP5+ populations. While this rather nicely summarizes those that are CGE derived, it would be of considerable interest to better classify them (Alpha7, CCK, SNCG, VIP multipolar vs bipolar, etc). The analysis focuses upon the stream of interneurons that arise from the CGE and reside for a surprisingly long period in the temporal lobe is intriguing, as it highlights how this population might influence key aspects of development, including memory consolidation, and Alzheimer's disease. However, if the authors could better determine which of the populations are deleted, it would certainly add to their analysis. Beyond this, their comparison of humans to Rhesus Macaques is both insightful and clarifying, as it nicely illustrates differences in these two rather closely associated species. The comparison of how the species identified in humans is absent in the monkey nicely illustrates the differences in the cognitive development of these two species. Moreover, the authors do an excellent job of illustrating that the phenomenon they explore in EC is not tractable in any other species than humans. Taken together, the work presents a nice complement of the way that CGE interneurons have developed in a manner unique to humans, with a particularly exciting nod to psychiatric disease.

We thank the reviewer for their comments regarding the revised manuscript. To provide a deeper illustration of the identity of the CGE populations we began by reanalyzing our combined transcriptomic dataset. We were able to improve our pre-processing steps for the snRNA-seq experiments, making them more consistent

across samples and leading to changes in the UMAP plots throughout the paper which further strengthened our analysis and confirmed previous conclusions.

For the immature neurons, we have now identified 5 major classifications, one LGE-derived and 2 MGE-derived and 2 CGE-derived (new **Fig. 6d**) Interestingly, we find that only one of these (CGE-2) is enriched in cells isolated from the EC stream microdissection (new **Fig. 6f**). Next, in order to provide greater depth and consistency of nomenclature we performed label transfer to our dataset from an adult human temporal lobe dataset from the Allen Brain Institute (Gabitto et al. 2023). This dataset consisting of 41,000 interneurons in the adult middle temporal gyrus, allowed us to break down the interneuron population in the postnatal EC of our dataset into more granular identities (e.g. we now identify chandelier, SNCG, and Pax6 interneurons). The CGE-derived CCK+ and Alpha7 populations are not labeled as discrete populations in these reference datasets but we included plots depicting the expression of these and other common markers of CGE-derived interneuron populations in **Extended Data Fig. 11i**.

Next, we used a spatial transcriptomic dataset from the human temporal lobe (Fang et al. 2022) to analyze the expression of genes associated with different cortical layers and we observed that the young interneurons in the EC stream had a higher expression of genes associated with upper layer LAMP5+ cells, consistent with the location of LAMP5+/COUPTFII+ neurons in the EC at birth (**Fig. 6n**). Together the revised lineage analysis of scRNAseq data reveals a population of migratory cells arriving through the EC stream that are primarily within the CGE-derived LAMP5+RELN+ lineage (**Fig. 6**).

We have also revised the discussion to consider the Reviewer's question about EC cell types that are lost during the progression of Alzheimer's Disease. A recent study examined a large cohort of samples from pre-AD patients with an accumulation of plaques in the cortex but no AD diagnosis (Gazestani et al. 2023). Interestingly, they found that cortical samples from these patients had a significant loss of a population of *NDNF+PROX1+RELN+* cells leading to an imbalance in excitation-inhibition. These cells have a very similar gene expression profile to the upper layer LAMP5+RELN+ cells arriving late through the EC stream. We now highlight this connection in the final paragraph of the discussion.

Referee #3 (Remarks to the Author):

The authors used histological and snRNA-seq data from valuable pre- and postnatal human samples to examine in detail postnatal immature neurons associated with the entorhinal cortex. Three-dimensional data revealed that the EC stream is a lamina structure with an abundance of immature neurons, in contrast to the tubular structure of the RMS. In the entorhinal cortex, DCX-positive cells were found in the brains of even 2-3-year-old infants, suggesting a long-term supply of neurons during the postnatal period. snRNA-seq data suggest the immaturity of cells in the EC stream and the maturation of EC cells, supporting the authors' idea that mature interneurons are

supplied from the EC stream to the ECs. The results presented in this paper, which elucidate some of the mechanisms of normal human brain development, are likely to be of immediate interest not only to specialists in neurogenesis, but also to researchers in multiple specialties, including pediatrics, psychiatry, and neurology. However, I believe that this paper can be improved if the major and minor concerns I have indicated can be addressed.

Major point

1. Additional data have strengthened the idea that the EC stream is a likely source of neurons to the ECs, but there is still no evidence that mature interneurons in the ECs are supplied by the EC stream. While the authors may feel that macaques, which leave only sparse DCX-positive cells at birth and differ in the molecular identity of DCX-positive cells, are insufficient, if live macaques are available to label cells in the EC stream and to observe their migration and differentiation into mature neurons, the authors' conclusion would be more strongly supported.

Unfortunately, direct lineage tracing of the young migratory cells in the EC-stream in humans, where we see the large numbers of migrating young neurons, is not feasible. As the reviewer notes, we do observe sparse individual DCX+ cells in the EC stream region in the macaque brain at birth. However, in the macaque, we did not observe these neurons extending out into the EC, as we do in humans and they do not express COUPTFII. Instead, we found that many express SP8 which is consistent with the majority of V-SVZ-derived neurons that migrate through the RMS to the olfactory bulb. In the revised **Fig. 4I** we now highlight these differences in marker expression by showing SP8 and COUPTFII expression side by side in the rare DCX+ cells in the macaque. We also show how different this is from the cells observed in the newborn human brain (which are dense clusters of COUPTFII+ cells that only infrequently express SP8; please compare **Extended Data Fig. 4a** to **Extended Data Fig. 4b**). We have also clarified this better in the results: "These results indicate that the postnatal macaque brain does not have a comparable postnatal migratory stream, but we cannot exclude the possibility that similar neurons arrive earlier during gestational ages".

In our original manuscript we used the following approaches that linked the EC stream to the EC: (1) Anatomical orientation of the stream across all sectioning levels of the EC showing the stream pointing toward the EC from the ventricle (**Fig. 1d, Extended Data Fig. 1d**); (2) Mapping the distribution and orientation of migratory cells in the stream and in the EC at birth, 7 months, 2 and 3 years (**Fig. 2a,b, Extended Data Fig. 2a,b**). (3) Orientation of the leading process of the migratory cells in the stream, exiting the stream, and within the EC itself. (**Fig. 2b, Extended Data Fig. 2**). (4) The observation of a subpopulation of the immature migratory neurons in the EC stream already expressing RNA and protein for subtypes of interneurons (LAMP5 and RELN, **Fig. 6k-m**) (5) Pseudotime analysis of the EC stream dataset placing it within the developmental trajectory of LAMP5+ CGE-derived interneurons which we observe to be present in the postnatal EC stream and EC using immunohistochemistry (**Fig. 6g,h**).

In the revised manuscript we have added new data that strengthen the evidence that interneurons in the EC are supplied by the EC stream: (1) Immunostaining for secretogin (SCGN), a protein that is important for neuronal migration and that is present in young CGE-derived migratory neurons. We now show a large population of SCGN+ neurons in the human CGE at 22 GW that extend towards the EC. In the postnatal EC stream, we found that a large population of the migratory Dcx+ cells are SCGN+ and extend into the EC between birth to 2 years of age (new **Fig. 4g,i, Extended Data Fig. 9d**). The morphology, abundance, and orientations of these SCGN+ cells strongly support the above observations that DCX+ cells in the EC stream are young interneurons migrating to the EC and neighboring cortical regions. (2) To connect the gene expression of interneurons in the EC stream to their location in the EC, we re-analyzed our data together with a spatial transcriptomic dataset of the human temporal lobe (Fang et al. 2022). We find that the immature neurons in our EC stream microdissection already express genes consistent with CGE-derived superficial LAMP5+ interneurons (new **Fig. 6i-k, Extended Data Fig. 13i**), which is consistent with the location of LAMP5+COUPTFII cells in the EC at birth (**Fig. 6n**). This further supports the link between the EC stream and interneurons in the EC and also provides a more precise identification of the types of interneurons migrating in the EC stream.

2. The lamina structure appears to be two-layered in the new 3D data added by the authors, and a two-layered structure of DCX-positive cells is also observed in the extended data Fig. 3E (birth). Throughout the paper, the collapsed part of the ventricle is considered an EC stream and a source of neurons to the postnatal ECs. The upper layer in the figure is reasonable as the collapsed part of the ventricle, but the lower layer would need some explanation.

This is a good observation that we had illustrated, but did not explain. We now explain that the EC stream contains multiple chains of DCX migrating cells separated by glial fibers creating the multi-layered structure that the reviewer noticed.

The introductory description of the stream in the results now addresses the point raised by the reviewer, defining the EC stream as an "*expansive medially-oriented network of migratory chains extending towards the entorhinal cortex*" (final sentence in the first section of the results). Within the human subventricular zone, some neurons migrate close to the ependymal layer and some migrate in clusters parallel to the VZ, but farther from the ventricle (This feature is especially visible in **Extended Data Fig. 4c inset 2**). We now clarify that the EC stream arises from the merging of two subventricular zones when the ventricle fuses together during gestation. Each of these subventricular zones spans ~2-300 microns away from the lateral and medial walls of the ventricle. After ventricular collapse, this results in a ~4-500 micron region formed by the combined subventricular zones containing migratory clusters of neurons, astrocytes, and parallel vimentin+ fibers. In serial coronal and sagittal sections, two or more chains of migrating neurons can be seen running in parallel (**Fig. 1d,e**). The chains of migratory cells the reviewer notes in **Extended Data Fig. 3e** are within the

~200-300 micron span of this collapsed ventricular region and are closely associated with the vimentin+ fibers in the collapsed ventricular region.

3. I. 71-73 “but the development of the unique layer structure of the EC follows an atypical pattern of neurogenesis compared to other cortical regions in many mammals including humans”

I assume that this somewhat vague description is due to the strict word limit, but a more specific description of the differences between the EC and other cortical regions would help clarify the significance of studying the EC.

Thank you for pointing out this need for clarification; we have now revised the introduction to clarify these points. We now explain how the EC develops very early in humans and there are some suggestions that it continues to develop postnatally, providing a rationale for studying postnatal development of the human EC. Interestingly, the cytoarchitecture of EC superficial layers appears to form in mid-gestational ages in humans (Šimić et al. 2022) and some layers develop with simultaneous birthdates in what has been termed parallel lamination (Liu et al. 2021).

4. The authors use fetal period samples as Birth samples, but this is not clear without a careful reading of the following text:

Method I. 499-500 “For infant cases, when the brain is at full term (37 to 40 gestational weeks), we refer to this as “birth”.”

I think it is understandable that the authors have mixed prenatal and postnatal samples because human samples are difficult to obtain. For these samples, I think it is acceptable to write "birth" in the figure, but at least in the text, it would be better to use more precise expressions such as "around the time of birth" instead of "at birth".

The reviewer is correct that in some places conclusions are drawn from multiple samples between 37-41 GW (considered full term). This is primarily the case for Figure 1 where the samples shown are 38GW, birth, and 10 days postnatal. We have now revised the title of this figure to refer to the “perinatal” time window. For increased clarity, we also updated all of the figures to indicate the precise gestational or postnatal ages of the samples shown. For conclusions that are drawn from only samples at birth, we have left them to say “at birth.”

Minor point

1. Fig. 4k I think the error bar should be shown.

We thank the reviewer for finding this omission and have revised this figure to include standard error bars.

2. Fig. 6 Please unify the variations in notation. For example, are EC stream (14 d), Migratory Stream, EC (14d-79y), and Postnatal EC the same thing?

We have revised this figure to standardize the notation so that the ages being referred to are now always included. For example, we now refer to the EC stream microdissected sample as the EC stream (14d). The EC microdissected samples are now referred to by age range directly e.g. EC (14d–79y).

References

Fang, Rongxin, Chenglong Xia, Jennie L. Close, Meng Zhang, Jiang He, Zhengkai Huang, Aaron R. Halpern, et al. 2022. “Conservation and Divergence of Cortical Cell Organization in Human and Mouse Revealed by MERFISH.” *Science* 377 (6601): 56–62.

Gabitto, Mariano I., Kyle J. Travaglini, Victoria M. Rachleff, Eitan S. Kaplan, Brian Long, Jeanelle Ariza, Yi Ding, et al. 2023. “Integrated Multimodal Cell Atlas of Alzheimer’s Disease.” *bioRxiv*. <https://doi.org/10.1101/2023.05.08.539485>.

Gazestani, Vahid, Tushar Kamath, Naeem M. Nadaf, S. J. Burris, Brendan Rooney, Antti Junkkari, Charles Vanderburg, et al. 2023. “Early Alzheimer’s Disease Pathology in Human Cortex Is Associated with a Transient Phase of Distinct Cell States.” *bioRxiv : The Preprint Server for Biology*, June, 2023.06.03.543569.

Liu, Yong, Tobias Bergmann, Yuki Mori, Juan Miguel Peralvo Vidal, Maria Pihl, Navneet A. Vasistha, Preben Dybdahl Thomsen, et al. 2021. “Development of the Entorhinal Cortex Occurs via Parallel Lamination during Neurogenesis.” *Frontiers in Neuroanatomy* 15 (May): 663667.

Šimić, Goran, Željka Krsnik, Vinka Knezović, Zlatko Kelović, Mathias Lysholt Mathiasen, Alisa Junaković, Milan Radoš, et al. 2022. “Prenatal Development of the Human Entorhinal Cortex.” *The Journal of Comparative Neurology* 530 (15): 2711–48.

Reviewer Reports on the Second Revision:

Referees' comments:

Referee #3 (Remarks to the Author):

The authors have successfully revised and improved the paper by carefully addressing the points I made in the previous manuscript. The authors' responses to each of my comments, including the description of EC streams, are appropriate and convincing. I think this is an important study worthy of publication in Nature.